astrophysics/galaxies/observational astronomy

galaxies: evolution, galaxies: ISM, submillimetre: galaxies, techniques: interferometric

**Author for correspondence:**
J. A. Hodge
e-mail: hodge@strw.leidenuniv.nl

# High-redshift star formation in the Atacama large millimetre/submillimetre array era

## J. A. Hodge[1] and E. da Cunha[2,3,4]

[1]Leiden Observatory, Leiden University, PO Box 9513, 2300 RA Leiden, The Netherlands
[2]International Centre for Radio Astronomy Research, University of Western Australia, 35 Stirling Highway, Crawley, Western Australia 6009, Australia
[3]Research School of Astronomy and Astrophysics, Australian National University, Canberra, Australian Capital Territory 2611, Australia
[4]ARC Centre of Excellence for All Sky Astrophysics in 3 Dimensions (ASTRO 3D)

JAH, 0000-0001-6586-8845; EdC, 0000-0001-9759-4797

The Atacama Large Millimetre/submillimetre Array (ALMA) is currently in the process of transforming our view of star-forming galaxies in the distant ($z \gtrsim 1$) universe. Before ALMA, most of what we knew about dust-obscured star formation in distant galaxies was limited to the brightest submillimetre sources—the so-called submillimetre galaxies (SMGs)—and even the information on those sources was sparse, with resolved (i.e. sub-galactic) observations of the obscured star formation and gas reservoirs typically restricted to the most extreme and/or strongly lensed sources. Starting with the beginning of early science operations in 2011, the last 9 years of ALMA observations have ushered in a new era for studies of high-redshift star formation. With its long baselines, ALMA has allowed observations of distant dust-obscured star formation with angular resolutions comparable to—or even far surpassing—the best current optical telescopes. With its bandwidth and frequency coverage, it has provided an unprecedented look at the associated molecular and atomic gas in these distant galaxies through targeted follow-up and serendipitous detections/blind line scans. Finally, with its leap in sensitivity compared to previous (sub-)millimetre arrays, it has enabled the detection of these powerful dust/gas tracers much further down the luminosity function through both statistical studies of colour/mass-selected galaxy populations and dedicated deep fields. We review the main advances ALMA has helped bring about in our understanding of the dust and gas properties of high-redshift ($z \gtrsim 1$) star-forming galaxies during these first 9 years of its science operations, and we highlight the interesting questions that may be answered by ALMA in the years to come.

# 1. Introduction

## 1.1. State of the field prior to ALMA

When newly formed stars and their surrounding HII regions exist in the presence of cosmic dust grains, a fraction of the short-wavelength emission may be absorbed by those grains and re-emitted in the far-infrared (FIR). This basic fact has long been a hindrance to the development of a complete picture of high-redshift star formation, which has been largely pioneered by studies in the rest-frame UV/optical. In particular, in the two decades since the now iconic image of the *Hubble* Deep Field (HDF; [1]) was released by the *Hubble Space Telescope* (*HST*), studies of the high-redshift galaxies detected in the HDF and its deeper successors have converged on a general picture for both when and how that star formation occurred. The majority of the Universe's stars appear to have been formed during the peak in the cosmic star formation rate (SFR) density, at redshifts between $z \sim 1 - 3$ (e.g. [2]). Moreover, a tight relation has been observed between a galaxy's star formation rate and stellar mass, and the persistent lack of scatter in the relation observed out to redshifts of at least $z \sim 6$ has been used to argue that the peak in the cosmic SFR density is primarily due not to the increased rate of mergers/interactions during this period—as was previously thought—but rather due to continuous gas accretion (e.g. [3–9]). However, it has also been known since the launch of the first infrared sky surveys, e.g. by the *Infrared Astronomical Satellite* (*IRAS*; [10]), and the *Cosmic Background Explorer* (*COBE*; [11]), that a substantial fraction of the Universe's high-redshift star formation is heavily enshrouded by dust (e.g. [12]). As the dust-reprocessed starlight emitted in the far-infrared (FIR) is redshifted to (sub-)millimetre wavelengths at high-redshift (figure 1), telescopes sensitive to this long-wavelength emission are required in order to detect the bulk of the star formation in distant galaxies. Understanding the prevalence and nature of this dusty star formation over the lifetime of the Universe has remained a challenge.

This review is about the Atacama Large Millimetre/submillimetre Array (ALMA; e.g. [15]) and the huge impact it has made—and will continue to make—toward our understanding of dust-obscured star formation in the distant ($z > 1$) Universe. The success of ALMA builds on the huge progress made by earlier long-wavelength telescopes, including (far-)infrared satellites such as the *Spitzer* [16] and *Herschel* [17] space telescopes, radio interferometers like the Karl G. Jansky Very Large Array (VLA [18,19]), single-dish submillimetre telescopes such as the James Clerk Maxwell Telescope (JCMT; [20]), the IRAM 30-metre telescope [21], the Atacama Submillimetre Telescope Experiment (ASTE; [22]), the Atacama Pathfinder EXperiment (APEX; [23]), and the South Pole Telescope (SPT; [24]), and earlier (sub-)millimetre interferometers such as the Submillimetre Array (SMA; [25]) and the Plateau de Bure interferometer (PdBI; [26] now succeeded by the NOrthern Extended Millimetre Array, NOEMA). These facilities have already revolutionized our view of high-redshift dusty star formation, from discovering submillimetre galaxies (SMGs) in the first extragalactic surveys with single-dish submillimetre telescopes, to quantifying the relative contribution of dusty star formation over much of cosmic time. Thanks to these facilities, it is now understood that, during the peak of the cosmic SFR density, the power emitted in the ultraviolet (UV) by young stars was an order of magnitude smaller than that emitted in the infrared (IR) due to dust reprocessing (e.g. [27–31]), with *Herschel* detections alone accounting for 50% of all stars ever formed [8]. Moreover, in addition to being dustier during the peak epoch of star formation, we now know that galaxies also had higher molecular gas fractions than local galaxies (e.g. [32–35]), highlighting the critical importance of studies of the cool interstellar medium (ISM).

However, despite the significant progress made in the pre-ALMA era, a large gap in our knowledge of the dust and gas reservoirs of high-redshift star-forming galaxies has persisted. This gap was largely due to the limited capabilities of pre-ALMA era facilities. In particular, only the bright so-called 'SMGs' could be detected in the distant universe by pre-ALMA era single-dish submillimetre telescopes (e.g. [36]), and at the highest redshifts ($z > 5$), only the most extreme and highly star-forming of those could be studied. Detections of the associated cool gas reservoirs of distant star-forming galaxies were similarly limited, with the majority of the detections resulting from targeted observations of the brightest SMGs and quasi-stellar object (QSO) host galaxies (e.g. [37–41]). In addition, while *Herschel* has contributed significantly to our understanding of the cosmic importance of dust-obscured star formation (e.g. [27,42]), its poor angular resolution (approx. 18″ at 250 μm and approximately 36″ at 500 μm) leads to significant source blending. Single-dish (sub-)millimetre telescopes have faced a similar challenge, with a typical resolution of the order of approximately 15″ to greater than 30″ (equivalent to greater than 100 kpc at $z \sim 2$). Far from allowing detailed studies of the dusty star formation in distant galaxies, this blending gives rise to the more fundamental challenge of reliably identifying the individual galaxies in the first place. Finally, despite

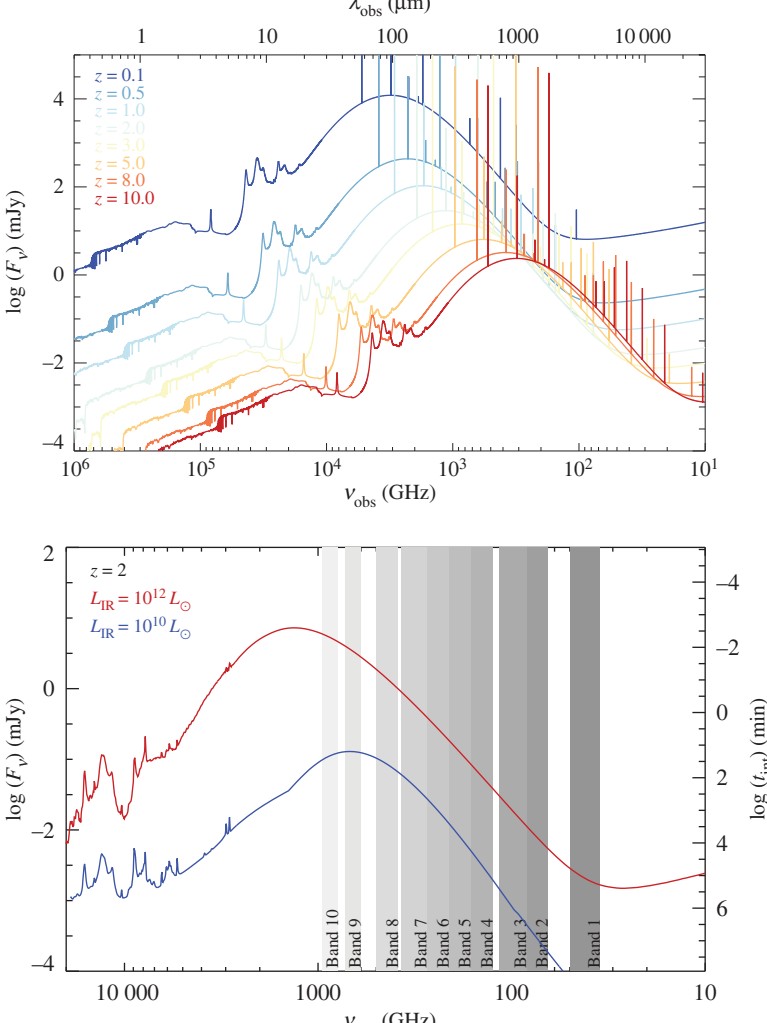

**Figure 1.** (*a*) Redshift evolution of the observed flux density of a galaxy at various wavelengths from the ultraviolet to the radio. We use the median spectral energy distribution (SED) of the ALESS submillimetre galaxies obtained by da Cunha *et al.* [13], with an infrared luminosity of $L_{IR} = 3.6 \times 10^{12} L_\odot$, and plot the brightest far-infrared/submillimetre cooling lines and CO lines for illustrative purposes. This clearly shows the effect of the negative *k*-correction at (sub-)millimetre wavelengths, where the cosmological dimming of more distant sources is (partially) compensated by the peak of the SED shifting into the wavelength range. (*b*) Galaxy dust SEDs at $z = 2$ compared with the ALMA frequency band ranges, indicated by the grey shaded regions. We plot two template SEDs from Rieke *et al.* [14], which are based on local dusty star-forming galaxies, one with $L_{IR} = 10^{10} L_\odot$, in blue, and one with $L_{IR} = 10^{12} L_\odot$, in red (note that these templates are plotted here to indicate the approximate expected (sub-)millimetre flux densities for similar dust luminosities at $z = 2$; high-$z$ galaxies may not have the same relation between infrared luminosity and dust temperature, i.e. SED peak). The right-hand axis shows the indicative integration time required to obtain a $3\sigma$ detection with ALMA in Band 6 at 230 GHz (using 50 antennas and standard precipitable water vapour conditions).

concerted efforts with interferometers such as the PdBI, SMA and VLA (e.g. [35,43–46]), resolved (sub-galactic-scale) studies of the dusty star formation and gas have been largely restricted to a handful of the very brightest (e.g. GN20; [47–49]) or most strongly magnified sources (e.g. the 'Cosmic Eyelash'; [50,51]). All of these pre-ALMA-era limitations meant that the nature of dust-obscured star formation at high-redshift—including the morphology, associated gas content, dynamics, efficiency, obscured fraction, contribution to the infrared background, or even what sources host it—remained largely unknown.

## 1.2. The unique capabilities of ALMA

The advent of ALMA has ushered in a new era for studies of high-redshift star formation. ALMA is situated on the Chajnantor plateau at over 5000 m (16 000 feet) above sea level, where atmospheric

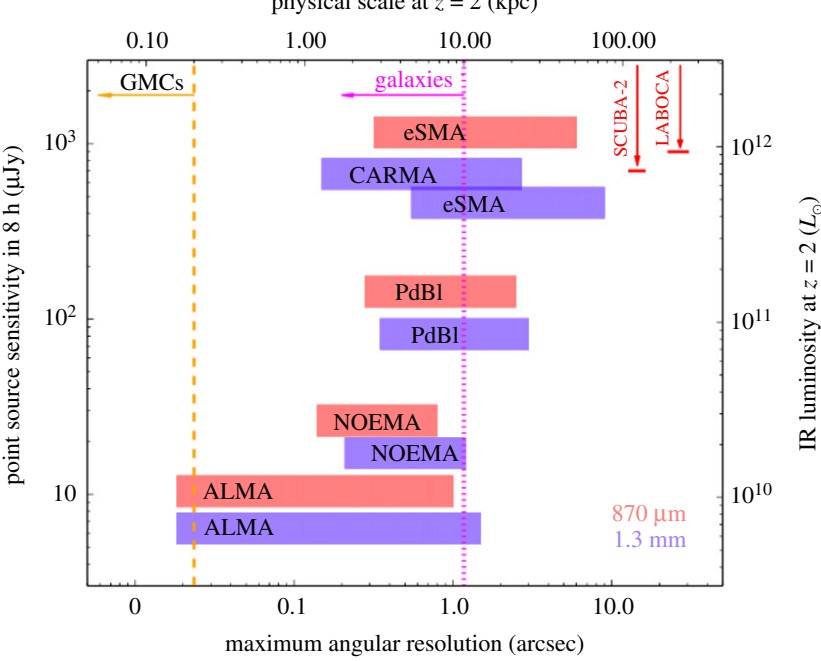

**Figure 2.** Point source sensitivity at 230 GHz (1.3 mm) achievable in 8 h on-source versus maximum angular resolution for existing and planned (sub-)millimetre interferometers. Point source sensitivity estimates were calculated assuming 3 mm of precipitable water vapour (PWV), a mean target elevation of 45°, the full available bandwidths and typical receiver temperatures as published on the websites. A PWV of less than or equal to 3 mm occurs approximately 10%, approximately 35%, approximately 65% and approximately 80% of the time for CARMA, the PdBI/NOEMA, the SMA and ALMA, respectively. The dotted/dashed lines show the maximum FIR size of local galaxies (dotted; e.g. [52]) and galactic giant molecular clouds (dashed; GMCs; e.g. [53]). Also shown are the single-dish resolutions and confusion limits at 850 μm for the SCUBA-2 camera on the JCMT and the LABOCA camera on APEX. The top and right-hand axes convert these quantities to physical scale and IR luminosity at $z = 2$ assuming the standard cosmology (see §1.3) and an Arp 220 (i.e. local ultra-luminous infrared galaxy; ULIRG) SED. For an M100 (local spiral) SED, the IR luminosities on the right-hand axis would be a factor of approximately 3 higher (because of the cooler average dust temperature). Note that the IR luminosities (right axis) implied by a given flux density are approximately constant over a large range in redshift $z > 1$ due to the negative $k$-correction (figure 1). Similarly, the physical scale on the top axis is approximately correct over $1 < z < 3$ due to the geometry of the Universe. We caution that for all interferometers (including ALMA), there is an inherent trade-off between spatial resolution and surface brightness sensitivity, which is not reflected in this figure.

conditions are exceptionally dry. The amount of precipitable water vapour (PWV) in the atmosphere is less than 1.0 mm for over 50% of the time during the best-weather months (June to November[1]). ALMA has 66 antennas in total: fifty 12 m antennas in its main reconfigurable array, plus twelve 7 m antennas in the Atacama Compact Array (ACA), and an additional four 12 m antennas in the Total Power Array (TPA). ALMA started scientific operations in 2011, with full operations started in 2013, and in the relatively short time since then, we are already witnessing its transformative power thanks to a number of key capabilities:

— *Angular resolution.* The configurations offered for ALMA's 12 m array provide angular resolutions ranging from a few arcseconds down to approximately 10 milli-arcseconds, corresponding to physical scales as small as a couple hundred parsecs for an unlensed galaxy at $z \sim 2$ (figure 2). Even at the low-resolution end, this is a huge increase in resolution over single-dish telescopes. For example, already in the first early science cycle (Cycle 0), the most compact (i.e. 'low'-resolution) configuration provided 1.5″ resolution at 345 GHz (i.e. 870 μm; Band 7), approximately 200× better in area than the LABOCA instrument on the APEX single-dish telescope at the same frequency. At the high-resolution end, it is also a significant improvement over previously existing (sub-)millimetre interferometers. For example, the maximum angular resolution of the PdBI ranged from approximately 1″ at 85 GHz to a few tenths of an arcsecond at 230 GHz. ALMA's

[1]ALMA Cycle 7 Proposer's Guide: https://protect-us.mimecast.com/s/2IQgC0RBYBiGqyL9IrGeJNV?domain=almascience.org.

resolution has increased with each new cycle and particularly following the success of the 2014 Long Baseline Campaign [54], with the full resolution already offered at 230 GHz in Cycle 5 using the approximately 16 km baseline pads (providing a resolution of 18 mas). Longer baseline expansions are already being discussed in the community as a possible future upgrade, aiming at an angular resolution of 0.001″–0.003″.[2] Even within the currently scoped project, ALMA's superb resolution allows observers to not only detect the dust-obscured star-formation and star-forming gas in individual high-redshift galaxies without blending, but also to resolve the dusty star-forming regions within individual galaxies on scales similar to—or even significantly better than—existing optical telescopes.

— *Frequency coverage*. The 10 bands nominally planned for the full ALMA offer near-continuous frequency coverage from 35 to 950 GHz; eight of these bands are already operating, with Band 1 (35–50 GHz) currently in production, and Band 2 (65–90 GHz) foreseen to start in the next couple of years. The frequency range covered by the ALMA bands probes the thermal dust spectrum in high-redshift galaxies, from the long-wavelength Rayleigh–Jeans tail to the SED peak and even shortward for the highest-redshift galaxies (figure 1). In addition to the dust, this wavelength range makes ALMA sensitive to a variety of molecular, atomic and ionization emission lines, which can be the only/best way to confirm redshifts and study the dynamics of dusty high-redshift galaxies. They also provide information on the total quantity and characteristics of the ISM in these sources. Coupled with progress in, e.g. large-scale hydrodynamic simulations (e.g. EAGLE; [55,56]), this allows theoretical predictions about the gas content of galaxies (e.g. [57–59]) to be tested.

— *Bandwidth*. The simultaneous (complementary) frequency coverage within (across) the ALMA bands allows spectral scans to identify the redshifts of dusty galaxies directly in the (sub-)millimetre. As mentioned above, this can be the only way to determine redshifts for the dustiest galaxies, as well as to confirm the redshifts of the highest-redshift sources. Combined with ALMA's sensitivity, the simultaneous bandwidth also provides the opportunity for serendipitous emission line searches for sources within the field of view.

— *Sensitivity (continuum and line)*. Another area where ALMA breaks new ground is in terms of sensitivity. ALMA has a point source sensitivity 10–100× better than previous telescopes covering the same wavelength range in the continuum, and it is 10–20× more sensitive for spectral lines. For detection experiments, this huge jump in sensitivity means that ALMA can detect galaxies much further down the luminosity function than previous (sub-)millimetre telescopes. An increase in angular resolution of a factor of $R$ requires an $R^2$ improvement in sensitivity to conserve surface brightness sensitivity, so this increased sensitivity is also necessary for (resolved) imaging studies. We note that, like all interferometers, ALMA is still limited by the unavoidable trade-off between spatial resolution and surface brightness sensitivity. ALMA offers the ACA to help improve the imaging of extended structures, but this limitation should nevertheless be kept in mind, particularly for observations with the most extended configurations.

## 1.3. This review

In this review, we will summarize some of the ways in which these unique capabilities have allowed ALMA to advance our understanding of star formation at high-redshift. Of course, it is impossible to speak about the progress of one facility in isolation. ALMA's discoveries complement the discoveries that many other facilities continue to make. Moreover, other new telescopes and instruments have allowed the pace of these discoveries to accelerate further. For example, the Submillimetre Common-User Bolometer Array 2 (SCUBA-2) on the JCMT [60] is providing wide-area surveys of high-redshift dusty star formation, with a mapping speed 100–150× faster than the previous SCUBA instrument [61]. Then there is the PdBI, which—with the addition of the seventh antenna in 2014—officially began its transformation into the NOrthern Extended millimetre Array (NOEMA; at the time of writing ten 15 m antennas are available). These telescopes have and will continue to contribute substantially to studies of distant dusty star formation in the era of ALMA.

This review will be divided into three sections based on the three methods typically used to select star-forming galaxies in ALMA's wavelength range. We begin in §2 with 'classic' SMGs: the luminous, dusty sources detected in single-dish (sub-)millimetre surveys, and thus the first dusty high-redshift galaxies to be studied in detail. Thanks largely to ALMA's sensitivity, as well as stacking studies, it is now increasingly possible to study the submillimetre emission from galaxies initially

[2]See http://alma-intweb.mtk.nao.ac.jp/diono/meetings/longBL2017/.

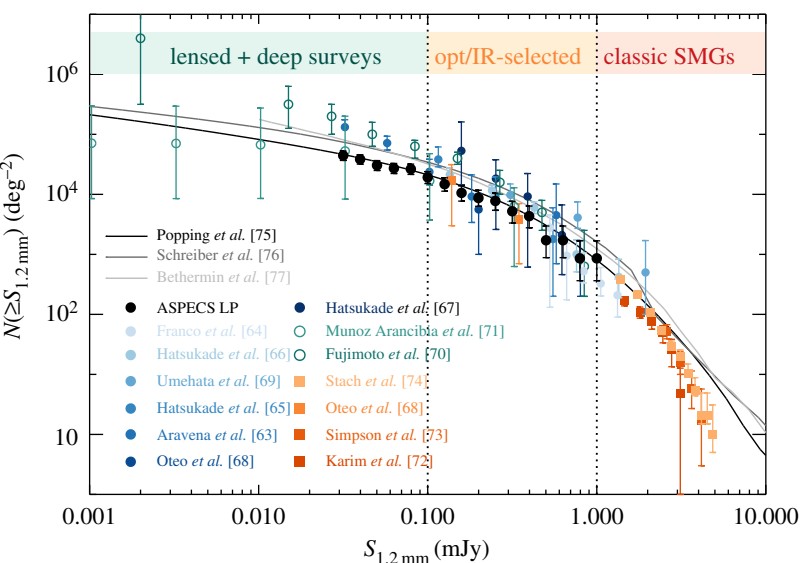

**Figure 3.** The range of flux density detections enabled by ALMA, as illustrated by the current state-of-the-art 1.2 mm number counts, following the compilation of González-López et al. [62]. The filled circles are number counts derived from deep blind fields, cluster fields and calibration fields [62–69]. The open circles extend in depth thanks to the inclusion of gravitationally lensed sources [70,71]. The orange to red filled squares correspond to ALMA follow-up of single-dish detected bright sources at 870 μm [68,72–74], with the conversion from 870 μm to 1.2 mm flux density following González-López et al. [62]. The grey lines show the number count predictions from the semi-empirical models of Popping et al. [75], Schreiber et al. [76] and Béthermin et al. [77]. We highlight three main regimes that this review focuses on: the bright end ($S_{1.2\,mm} \gtrsim 1\,mJy$), corresponds to 'classic SMGs' (§2); the flux density range $S_{1.2\,mm} \simeq 0.1-1\,mJy$ tends to be the realm of pre-selected galaxy surveys (typically in stellar mass or star formation rate; such surveys are discussed in §3); and the faint end ($S_{1.2\,mm} < 0.1\,mJy$) is now being probed for the first time thanks to deep surveys with ALMA, as discussed in §4.

selected at other wavelengths. We therefore discuss the dusty star formation in colour- and mass-selected galaxies in §3. In §4, we discuss the results from the latest blind (sub-)millimetre continuum and line surveys with ALMA, which aim to circumvent the inevitable bias that comes with pre-selection at other wavelengths. Section 5 contains some concluding remarks.

We acknowledge that this separation of different galaxies and survey types is somewhat artificial. As shown in figure 3, the 1.2 mm number counts are continuous, and the separation into different flux density regimes is historical and driven by the capabilities of available (sub-)mm facilities. The SMG realm at flux densities above 1 mJy was the first to be explored thanks to single-dish experiments, but the advent of more sensitive interferometers (first the PdBI, then ALMA) enabled surveys targeting fainter sources pre-selected in stellar mass or star formation rate, down to approximately 0.1 mJy. Now with the deepest ALMA surveys, using 150 h of deep integration in the deepest extragalactic deep field (ASPECS; e.g. [78]), or using strong gravitational lensing towards massive galaxy clusters (the Frontier Fields; e.g. [71,79]), we are probing a previously unexplored regime of faint sources, well below 0.1 mJy. As we start linking these flux density regimes with ALMA, we start connecting galaxy populations that were historically studied by different communities, e.g. SMGs and low-mass UV/ optically selected sources. In fact, the field is currently going through growing pains, as ALMA's ability to detect submillimetre emission in more 'normal' galaxies is forcing the submillimetre community and the general high-redshift community to merge, and, as we will see in what follows, the terminology is not yet completely aligned. This may seem like a simple question of semantics, but it is important to note, as our classifications have historically guided our physical interpretation. We will return to this in §2.4.

Finally, we note that it is impossible for this review to be complete with the avalanche of new results currently coming in. There are many topics related to those discussed in this review that we have decided not to cover, including (but not limited to) results on the role or host galaxies of active galactic nuclei (AGN), measurements of outflows and the large-scale environments of galaxies. For other recent reviews on the topics of dusty star-forming galaxies (i.e. SMGs), and dust and molecular gas in distant galaxies, we point the reader to Carilli & Walter [33], Casey et al. [80], Combes [81] and

Salim & Narayanan [82]; for a theoretical overview of models of early galaxy formation, see Dayal & Ferrara [83]. Here, we have simply attempted to highlight some of the main advances in the first several years of ALMA operations concerning (mostly dust-obscured) star formation at high-redshift, as well as the interesting questions for the next few years.

Where applicable we assume a concordance, flat $\Lambda$CDM cosmology of $H_0 = 71$ km s$^{-1}$ Mpc$^{-1}$, $\Omega_{\Lambda} = 0.73$ and $\Omega_M = 0.27$ [84,85]. Unless otherwise stated, AB magnitudes are adopted.

# 2. Submillimetre-selected galaxies followed up by ALMA

The line between what is considered an 'SMG' or not is blurring as ALMA probes deeper down the luminosity function, and it is the subject of continued debate. In this section, we focus primarily on the integrated properties of the original, single-dish detected sources selected at approximately 850 µm, as well as the strong lens candidates followed up by ALMA. For a comprehensive review on these and other IR-selected galaxies, which are also sometimes more generally referred to as dusty star-forming galaxies (DSFGs), we direct the reader to Casey *et al.* [80]. For a discussion of the resolved work on high-redshift galaxies (including SMGs) with ALMA in general, we refer the reader to §3.2. Here, we begin with a brief background on traditional SMGs to place the recent ALMA results into context.

## 2.1. Background

Thanks to the pioneering observations of the extragalactic background light (EBL) since the 1980s and 1990s by early infrared satellites like the *Infrared Astronomical Satellite (IRAS)* and the *Cosmic Background Explorer (COBE)*, it is well known that the cosmic infrared background (CIB) has an intensity similar to the optical background, implying that there is a comparable amount of light absorbed by dust and re-radiated in the (rest-frame) FIR as there is observable directly in the UV/optical [86,87]; see Cooray [88] for a recent review. Observations with ground-based, single-dish submillimetre telescopes (e.g. SCUBA) were the first to resolve this CIB into distinct sources, revealing a population of distant star-forming galaxies known as submillimetre-selected galaxies with 850 µm flux densities of greater than a few mJy (e.g. [89–92]); extremely infrared-bright galaxies had first been hinted at by *IRAS* observations [93]. In the subsequent years, multiwavelength campaigns, as well as deeper, large-area, blind surveys at (sub-)millimetre and IR wavelengths—including FIR efforts such as the *Herschel* Multi-tiered Extragalactic Survey (HerMES; [94]) and the *Herschel* Astrophysical Terahertz Large Area Survey (H-ATLAS; [95])—have gradually revealed the nature of these uniquely selected galaxies.

In general, these single-dish-detected SMGs appear to be massive (stellar mass approx. $10^{11}$ $M_{\odot}$), ultraluminous (approx. $10^{12}$ $L_{\odot}$) dusty galaxies with extreme SFRs (approx. $10^2$–$10^3$ $M_{\odot}$ yr$^{-1}$; [36]). Thanks to the so-called 'negative k-correction' at submillimetre wavelengths, the cosmological dimming that affects high-redshift sources is almost exactly offset by the shifting of their dust peak into the observed band, resulting in a flux density that can be close to constant across a large ($z \sim 1-10$) redshift range (figure 1). The first spectroscopic follow-up campaigns of the submillimetre-selected sources revealed a number density that peaked at $z \sim 2.5$ [96,97].

Despite hosting such copious star formation, SMGs can be very faint or even invisible in rest-frame optical/UV data—even where very deep imaging exists (e.g. [48,98,99])—due to significant dust obscuration at those wavelengths. Their associated large (rest-frame) infrared luminosities are one reason why they are often referred to in the literature as the high-redshift analogues of local ultra-luminous infrared galaxies (ULIRGs), although we shall see that there is increasing evidence that the picture is not so simple. Moreover, their number density at high-redshift is orders-of-magnitude higher than local ULIRGs (approx. $400\times$; e.g. [100]), and they appear to contribute significantly to both the volume—averaged cosmic star formation rate density at $z = 2-4$ (approx. 20%) and the stellar mass density (approx. $30-50\%$; e.g. [101]). As their peak redshift ($z \sim 2.5$) is also the peak of AGN activity (e.g. [102,103]), their enhanced star formation is thought to be tied to the evolution of QSOs (e.g. [104,105]) and ultimately to the build-up of massive elliptical galaxies (e.g. [106–109]).

While there has been substantial progress in understanding these galaxies in the approximately 20 years since they were first discovered, a large number of open questions regarding their nature remain. In particular, hierarchical galaxy formation models have found it difficult to simultaneously reproduce the number density and other observed properties (e.g. colours) of these high-redshift sources along with the local luminosity function in a $\Lambda$CDM universe (e.g. [110–113]). As a result,

various theoretical models have invoked a range of mechanisms to explain this population, including starburst-dominated major mergers (e.g. [114]), major+minor mergers with a flat (top-heavy) initial mass function (IMF) [110], a prolonged stage of mass build-up in early-Universe proto-clusters [115], the most massive extension of the normal ($z > 2$) star-forming galaxy population (e.g. [116–119]), or a combination of starbursts and isolated disc galaxies (e.g. [120]), with some models also including the effect of galaxies blended by the poor resolution of single-dish telescopes ([121–123], and see Casey *et al.* [80] for a summary of the strengths and weaknesses of the various theoretical models). Suffice it to say that the challenge these galaxies pose to modellers makes them a particularly interesting population for constraining theoretical models of galaxy formation.

The outstanding questions about the SMG population, in combination with their large submillimetre flux densities—making them (relatively) easy to observe—has also made them prime targets for observations with ALMA. In some cases, these new ALMA observations have increased the angular resolution achievable by factors of greater than 100 000 in area from the original single-dish observations, allowing not only the precise identification of previously blended galaxies, but also a detailed look at their sub-galactic ISM and dusty star formation properties. The lessons subsequently learned about the star formation process and ISM physics can inform our understanding of the star-forming ISM in the more general galaxy population, and in this way, these intrinsically bright (and/or strongly lensed) sources serve as laboratories for studying star formation at high-redshift (see also §3). In the following, we will discuss some of the key areas where ALMA has contributed—and will continue to contribute—to our understanding of this galaxy population.

## 2.2. Resolving single-dish submillimetre galaxies

### 2.2.1. Precise location and counterpart identification

One of the first results to come out of early ALMA observations was the precise location of submillimetre-emitting galaxies. In particular, SMGs are sufficiently rare (approx. 200 per $\text{deg}^2$ down to $S_{870\mu m} = 5$ mJy) that the best way to find them is through surveys using wide-field single-dish telescopes with instruments such as, for example, the Submillimetre Common-User Bolometer Array 2 (SCUBA-2, or its predecessor SCUBA), the Large APEX BOlometer CAmera (LABOCA; [124]), the Astronomical Thermal Emission Camera (AzTEC; [125]), or the Spectral and Photometric Imaging Receiver (SPIRE; [126]). Surveys using these instruments have built up large samples of hundreds of SMGs with angular resolutions of approximately 15″ to even greater than 30″. Such low resolutions mean that there may be several to tens of galaxies visible in the ancillary multi-wavelength (e.g. optical) data, depending on its depth and the exact resolution, making it difficult to identify the counterpart(s) to the submillimetre-emitters. Identifying multi-wavelength counterparts is crucial for studying the SMGs, as this is how photometric (and sometimes spectroscopic) redshifts are targeted and derived. Without redshifts, or with the wrong redshifts, it is clearly difficult to place these galaxies and their implied physical properties in the broader context of hierarchical galaxy assembly.

Prior to ALMA, this relatively straightforward observational limitation posed a significant challenge to the field. While interferometric follow-up observations at approximately arcsecond resolution were possible with the SMA and PdBI, sensitivity limitations, and thus the observing time required, limited the observations to small numbers of sources (e.g. [127–131]). Various probabilistic techniques exploiting empirical correlations with the multi-wavelength data have been explored to circumvent this challenge. For example, Ivison *et al.* [132] used cross-matching with radio and/or 24 µm catalogues to identify counterparts to SMGs, estimating the likelihood of the sources being random chance associations to the submillimetre sources with the corrected Poissonian probability (*p*-statistic; [133,134]). Biggs *et al.* [135] expanded this method to include a S/N-dependent search radius. Other identification methods take into account the very red optical-infrared colours observed for these sources (e.g. [98,136–138]). An obvious limitation to such methods is the reliance on empirical correlations with other wavelengths, which may have significant scatter and may miss the faintest/highest-redshift counterparts in wavebands (radio, IR) that do not benefit from the negative *k*-correction.

With ALMA, even the most compact configurations allow the submillimetre-emitting galaxies to be accurately located at 850 µm, with an angular resolution of ∼1″ (figure 4; [74,140–142], and note that angular resolutions were slightly coarser in some Early Science configurations). Moreover, ALMA's huge increase in sensitivity over both single-dish (sub-)millimetre telescopes and previous generation interferometers (figure 2) means that all 'classical' SMGs can be detected in only a couple of minutes per source at submillimetre frequencies, allowing large samples to be followed up. Table 1 lists some

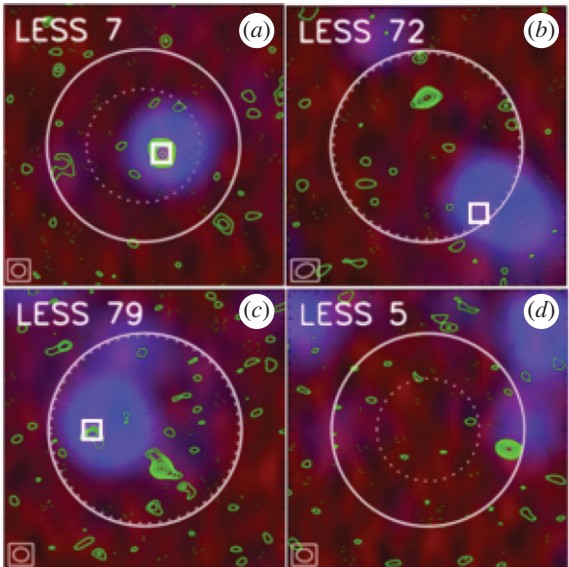

**Figure 4.** False-colour images (approx. 26″ × 26″) of four single-dish submillimetre sources from the LESS survey [139] targeted with ALMA by the ALESS survey [140], including 1.4 GHz VLA data (red), *Spitzer*/MIPS 24 μm data (blue) and ALMA 870 μm data (green contours). ALMA contours start at ±2σ and are in steps of 1σ. ALMA's synthesized beam (i.e. angular resolution) is shown in the bottom left-hand corner of each map (the typical angular resolution of these observations is 1.6″). The solid circle shows ALMA's primary beam FWHM, which is approximately equivalent to the angular resolution of the original LABOCA (single-dish) observations from Weiß *et al.* [139]. The dashed circle indicates the search radius used by Biggs *et al.* [135] to statistically identify radio and mid-infrared counterparts to the LESS sources [139], and the white squares indicate the positions of the predicted 'robust' counterparts. This figure shows examples of fields where the previously identified 'robust' counterparts were correct (*a*), incorrect (*b*), partially correct due to multiplicity (*c*; §2.2.2), and missed entirely due to the search radius used (*d*). Figure adapted from Hodge *et al.* [140].

of the largest SMG interferometric follow-up campaigns to date, where the ALMA campaigns were each completed in a matter of a few hours.

The availability of large samples of interferometrically observed SMGs, provided in a large part by ALMA, has allowed the completeness and reliability of previous methods for single-dish survey counterpart identification to be tested (figure 4).

This is important not only to understand the accuracy of past results based on such methods, but also for the continued use of such methods for wide-area single-dish surveys where interferometric follow-up may not be available, but which are still the best way to discover large samples of these bright but rare sources. Here, 'reliability' (sometimes also called 'accuracy') refers to the likelihood that a given multi-wavelength counterpart is actually a counterpart to the submillimetre emission, which can be defined as:

$$\text{reliability} = \frac{N_{\text{pred, true}}}{N_{\text{pred}}}, \tag{2.1}$$

where $N_{\text{pred, true}}$ refers to the number of predicted counterparts that were verified as true counterparts, and $N_{\text{pred}}$ refers to the number of total counterparts predicted. The 'completeness' then refers to the ability of the method to identify all true counterparts, and can be defined as:

$$\text{completeness} = \frac{N_{\text{pred, true}}}{N_{\text{true}}}, \tag{2.2}$$

where $N_{\text{pred, true}}$ again refers to the number of predicted counterparts that were verified as true counterparts, and $N_{\text{true}}$ refers to the total number of true counterparts discovered in the interferometric follow-up.

One of the main results of this interferometric follow-up has been the finding that single-dish counterpart identification methods were relatively *reliable*, but not necessarily *complete*. For example, follow-up of single-dish sources above a approximately a few mJy observed with an approximately 15–20″ beam find that the radio+mid-infrared (MIR) methods have a reliability of approximately 80% [136,140,147], but a completeness as low as approximately 50% [128,136,137,140,141,147] when only 'robust' counterparts are considered (typically defined as having a corrected Poissonian probability

**Table 1.** (Sub-)millimetre interferometrically observed SMG surveys[a].

| name | single dish sample properties | | | | interferometric follow-up | | | catalogue paper |
|---|---|---|---|---|---|---|---|---|
| | instrument/telescope | $\lambda$ | resolution | $S_\nu$ limit[c] | $N_{sources}$ | telescope/$\lambda$ | depth | |
| — | — | — | — | (mJy beam$^{-1}$) | — | — | (mJy beam$^{-1}$) | — |
| GOODS-N | SCUBA-2/JCMT | 850 μm | 14.5″ | 3.3 | 15 | SMA/860 μm | 0.7–1.5 | Barger et al. [127] |
| COSMOS | LABOCA/APEX | 870 μm | 19.2″ | 5.2 | 28 | PdBI/1.3 mm | 0.46 | Smolčić et al. [128] |
| ALESS | LABOCA/APEX | 870 μm | 19.2″ | 3.6 | 124 | ALMA/870 μm | 0.4 | Hodge et al. [140] |
| SPT | SPT/SZ | 1.4 mm | 1.05′ | 25 | 47 | ALMA/870 μm | 0.4 | Spilker et al. [143] |
| UKIDSS UDS | SCUBA-2/JCMT | 850 μm | 14.5″ | 5 | 30 | ALMA/870 μm | 0.2 | Simpson et al. [142] |
| HerMES | SPIRE/Herschel | 500 μm[b] | 36″ | 50 | 29 | ALMA/870 μm | 0.2 | Bussmann et al. [144] |
| COSMOS | AzTEC/JCMT | 1.1 mm | 18″ | 4.2 | 15 | SMA/890 μm | 1.0–1.5 | Younger et al. [130,131] |
| ″ | ″ | ″ | ″ | ″ | 15 | PdBI/1.3 mm | 0.2 | Miettinen et al. [141] |
| COSMOS | AzTEC/ASTE | 1.1 mm | 34″ | 3.5 | 129 | ALMA/1.25 mm | 0.15 | Brisbin et al. [145] |
| AS2UDS | SCUBA-2/JCMT | 850 μm | 14.5″ | 3.4 | 716 | ALMA/870 μm | 0.25 | Stach et al. [74] |
| BASIC | SCUBA-2/JCMT | 850 μm | 14.5″ | 1.6 | 53 | ALMA/870 μm | 0.095–0.32 | Cowie et al. [146] |

[a]Here we list continuum surveys of (sub-)millimetre-selected sources, some of which include strong gravitationally lensed sources as discussed in §2.3.
[b]Note the HerMES-selected sample was also observed at 250 and 350 μm. The SPIRE resolution at 250 μm is 18.1″.
[c]Limiting single dish flux density above which sources were selected for interferometric follow-up (deboosted values reported for all samples except for Simpson et al. [142]).

$p < 0.05$ in a given waveband). The completeness is higher for brighter (at approx. arcsecond-resolution) sources, as well as if only the 'dominant' (brightest) submillimetre interferometric component is considered [140,148]. The latter point has led to a lively debate in the community about the importance of those fainter submillimetre counterparts in various contexts (see e.g. §2.2.2 on 'Multiplicity'). Finally, the completeness is also higher if a fixed search radius is used instead of a S/N-dependent radius [140], and if counterparts identified as only 'tentative' (typically defined as $p < 0.1$) are considered as well, though the resultant decrease in reliability in this case is still debated [136,137,140].

These results have also led to the development and calibration of new and refined methods for single-dish source counterpart identification. For example, using an ALMA training set on a SCUBA-2 selected sample, Chen *et al.* [137] presented an Optical-IR Triple Color (OIRTC) technique that takes advantage of the fact that dusty, high-redshift galaxies like SMGs are generally red in optical-near-infrared (OIR) colours such as *i*–*K*, *J*–*K* or *K*–[4.5] (e.g. [149–152]). This results in counterparts with a similar reliability to the traditional radio/MIR *p*-value technique (approx. 80%) but with a higher completeness (69%). More recently, An *et al.* [136] used supervised machine-learning algorithms to identify SMG counterparts from optical/near-infrared-selected galaxies. They used a two-step approach combining a simple probability cut to select likely radio counterparts and then a machine-learning method applied to multi-wavelength data. This combined approach leads to a reported 85% completeness and greater than 62% precision [136]. While the reliability and completeness of such methods may be adequate for certain statistical studies, these results also highlight the continued importance of interferometric follow-up with telescopes such as ALMA, which are the only way to obtain a truly accurate view of the SMG counterparts.

### 2.2.2. Multiplicity

The speed at which ALMA can perform arcsecond-scale observations also enabled the confirmation of multiplicity in statistically significant samples [72,74,140,142]. Previous studies on smaller numbers of sources with the SMA [127,129,153] and PdBI [128], as well as even earlier in the radio [154], already indicated that some single-dish submillimetre sources could be blends of more than one galaxy. In the first years of ALMA, there has been an explosion in studies quantifying this multiplicity. The fraction of single-dish submillimetre sources[3] reported to show multiplicity varies based on the study, with reported values ranging from approximately 10 to 80% (e.g. [74,136,137,140–142,147,148,155]). An example of a single-dish source which was resolved into multiple distinct submillimetre sources with ALMA can be seen in figure 4c.

While some of the discrepancy may be due to small number statistics, much can be explained due to a number of factors which vary between studies, including resolution of the single-dish observations, submillimetre-brightness and S/N of the single-dish sources, submillimetre-brightness of the primary galaxy and depth of the follow-up interferometric observations (determining the dynamic range for detection of additional sources), size of the interferometric primary beam compared with the single-dish resolution (and whether sources are counted if the former is larger), wavelength of the follow-up observations and field-to-field variations in the global density of the extragalactic fields. For example, samples selected using 850 μm SCUBA-2 observations (14.5″ beam) find that the impact of multiplicity (defined as the number of interferometric sources which contributed to the original single-dish flux) is smaller than for, e.g. 870 μm LABOCA sources (19.2″ beam), suggesting that the higher SCUBA-2 resolution results in fewer blended sources in the original single-dish imaging [142,146,148]. There are also a number of studies reporting that the multiplicity is a function of flux density, with a higher multiplicity for brighter single-dish sources ([74,142,144], but cf. Miettinen *et al.* [141]). When these factors are controlled for, the ALMA results suggest that for $S_{850\,\mu m} > 4$ mJy single-dish sources with follow-up ALMA observations sensitive to $S_{850\,\mu m} = 1$ mJy sources across the whole ALMA beam, the true multiple fraction is likely to be higher than approximately 40% (e.g. [74]).

A continued uncertainty in the exact fraction of multiples is the existence of 'blank' maps. These are single-dish sources in which the follow-up interferometric observations fail to detect any sources. Such maps are present in large numbers in multiple surveys [74,140–142] despite the expectation that only a small fraction of the single-dish sources should be spurious (e.g. [139]). The depth reached by the ALMA observations would sometimes imply a large number ($N > 3$) of blended sources in order for them to be individually undetected (e.g. [74,140]), which would have repercussions for the submillimetre number counts. Deeper ALMA observations constraining the source multiplicity as a

---

[3]For a study of the multiplicity of *Herschel*-selected sources, e.g. Bussmann *et al.* [144].

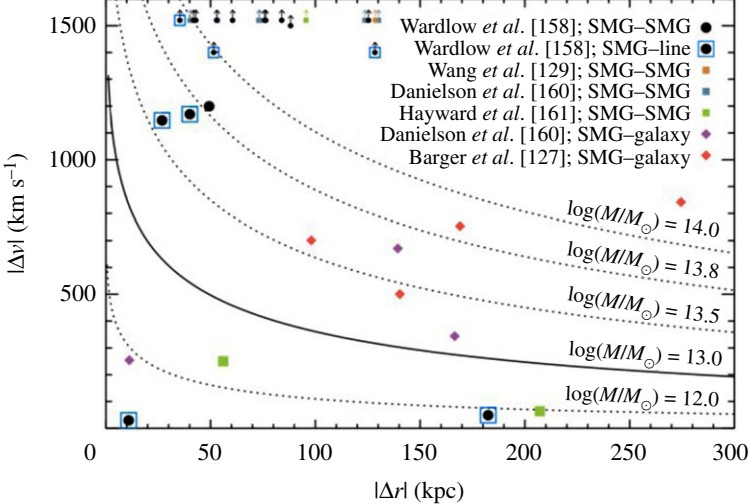

**Figure 5.** Radial and velocity separations for 870 μm-selected SMG–SMG pairs, serendipitously detected line emitters (SMG-line) and SMG-galaxy pairs from blank-field surveys [158]. The curved lines show the profiles expected from Navarro–Frenk–White (NFW) halos, where the solid curved line indicates the expected SMG halo mass based on clustering measurements (approx. $10^{13}$ $M_\odot$; cf. Garcia-Vergara et al. [159]). The majority of the galaxy pairs studied have larger velocity offsets than would be expected if they occupied the same virialized halos, and Wardlow et al. [158] further find that only $21 \pm 12\%$ of the currently studied SMGs with spectroscopically confirmed companions have spectral and spatial separations which could have resulted in interaction-induced star formation. Future work on larger and more complete samples will be needed to definitively characterize the relation of multiples and the importance of interaction-driven star formation in the SMG population. Figure from Wardlow et al. [158].

function of observed flux density will be important for constraining theoretical models for the formation of SMGs (e.g. [156], and see §2.2.3).

### 2.2.3. Relation of multiples

An interesting question raised by the ALMA observations is the relation of the galaxies in multiples that were previously blended. In particular, while some may be chance projections along the line of sight, others may be merging pairs or sources in the same halo, where an interaction between the companions may have triggered their starbursts. While the simulations make varying predictions for the relative importance of these populations—for example, Cowley et al. [156] suggest that most secondary SMGs should be line-of-sight projections with $\Delta z \sim 1$, while Hayward et al. [122] predict a more significant physically associated population—the observations are still limited. Photometric redshifts do not have the required accuracy to test these scenarios, and spectroscopic observations require more than one spectroscopic redshift per pointing. The latest ALMA results using spectroscopic (UV/optical or CO) redshifts suggest that the majority (greater than $50-75\%$) of the SMGs in blended submillimetre sources are not physically associated,[4] though these results are still plagued by small number statistics (figure 5; [158,160,161]).

In the absence of spectroscopic redshifts, some ALMA studies have used photometric redshifts to approach the question from a statistical point of view. For example, Simpson et al. [142] found that the number density of $S_{870\,\mu m} \gtrsim 2\,mJy$ SMGs in ALMA maps that target single-dish submillimetre sources was approximately 80 times higher than that derived from blank-field counts, suggesting a significant proportion of multiples are indeed physically associated, and Stach et al. [74] used a similar analysis to derive a lower limit on the fraction of physically associated pairs of *at least* 30%. An analysis of the distribution of separations between galaxies in the multiples also suggests a dependence on submillimetre source brightness, with the counterparts of brightest sources tending to lie significantly closer together ([144], though note the significant fraction of lensed sources in that sample). An excess of sources at small separations is not predicted in current theoretical models [123,156,162] and could indicate a more significant contribution from interacting/merging systems, but it could also be due to projection effects. As with the remaining uncertainties regarding the redshift distribution, the definitive

---

[4]See Gómez-Guijarro et al. [157] for a study of the relation of high-multiplicity *Herschel*-selected sources, where they reach a different conclusion.

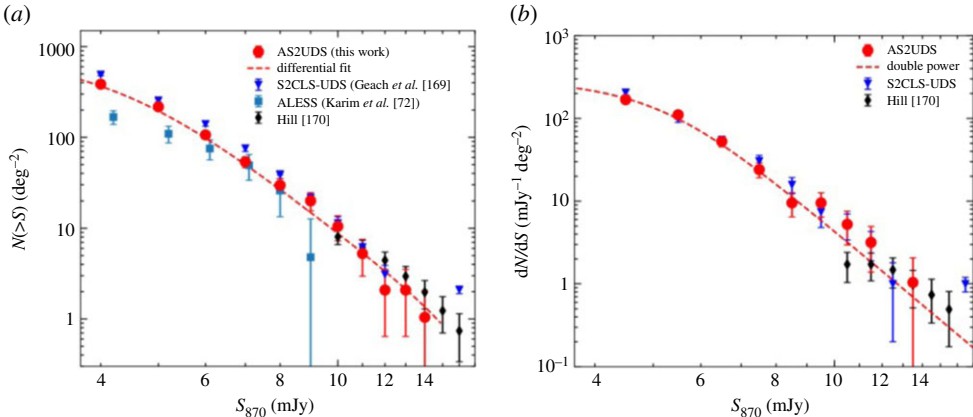

**Figure 6.** The 870 µm cumulative (*a*) and differential (*b*) number counts of approximately 700 ALMA-identified SMGs from the AS2UDS survey [74] compared with the original single-dish counts [169] as well as those from some earlier interferometric surveys. While the ALMA-derived number counts are broadly consistent with the single-dish results, they are systematically lower (37 ± 3% for this work) due to the effect of multiplicity (§2.2.2). Moreover, contrary to previous surveys over smaller areas (e.g. [72]), there is no evidence for a steep drop-off in the counts at large (approx. 9 mJy) flux densities. Figure from Stach *et al.* [74].

answer to this question will require complete samples of SMGs followed up with dust-unbiased (sub-)millimetre spectroscopy, where higher spectroscopic completeness is possible [158,163].

### 2.2.4. Number counts

One of the main reasons the topic of multiplicity in SMGs has generated so much interest is because of the implications for the submillimetre number counts, which have historically been very challenging to fit with hierarchical galaxy formation models, and are therefore one of the most important constraints for such models (e.g. [110–113,120,164–168]). Various simulations have suggested that the blending caused by multiplicity may help alleviate this tension (e.g. [123]). ALMA has contributed significantly in this area by demonstrating that, while the single-dish sources are indeed affected by multiplicity, the interferometrically derived number counts are still broadly consistent (within approx. $30-40\%$; [72,74,142]) with those inferred from earlier single-dish surveys (figure 6). These two seemingly contradictory statements can be reconciled by understanding that the primary (i.e. brightest) component detected interferometrically typically accounts for the bulk (approx. $80-90\%$) of the single-dish flux density [72,74,142].

The ALMA confirmation of the overall normalization of the submillimetre number counts is significant as it means that the tension with theoretical models remains. Various theoretical studies have thus worked on tackling this from the simulation side. In particular, Cowley *et al.* [156] presented some predictions from an updated version of the GALFORM semi-analytic galaxy formation model [110]. This model, described in detail in Lacey *et al.* [166], still requires a top-heavy IMF to match the SMG number counts, but with a less extreme slope (close to Salpeter). Some recent ALMA studies (e.g. [142]) report broad agreement between this model and the ALMA-derived number counts. Other works using both semi-analytic and semi-empirical models (e.g. [120,123,171]) have argued that IMF variation is not necessarily needed at all to match the number counts, given different assumptions about the radiative transfer calculations, merger evolution, cosmological context and other physical processes such as stellar feedback. We direct the reader to Casey *et al.* [80] for a thorough review of the strengths/limitations of the different classes of theoretical models and their implications for the SMG population.

### 2.2.5. A bright-end flux cut-off?

One area of continued debate relates to the multiplicity (and thus number counts) of the very brightest submillimetre sources. In particular, one of the first results on ALMA-derived submillimetre number counts [72] found that all of the brightest greater than 12 mJy single-dish sources were composed of multiple sources when viewed with ALMA (in marked contrast with previous SMA work by Younger *et al.* [131]), with individual 850 µm flux densities less than or equal to 9 mJy. Karim *et al.* [72] suggested that this

implies a physical limit to the SFRs of less than 1000 $M_\odot$ yr$^{-1}$, which could be due to a limited gas supply or feedback from star formation/AGN. This also suggests that the number of the brightest submillimetre sources ($S_{870\,\mu m} \gtrsim 9$ mJy) may have been overestimated in single-dish studies, and that the true space density of the most massive $z > 1$ galaxies should be small[5]: sources with gas masses greater than $5 \times 10^{10}$ $M_\odot$ would be less than $10^{-5}$ Mpc$^{-3}$. Support for an SFR cut-off also comes from the simulated infrared dusty extragalactic sky (SIDES) simulation [77], which is based on an updated version of the 2SFM (two star-formation modes) phenomenological galaxy evolution model [172], and where they are able to rule out the model without an SFR limit as already exceeding the single-dish counts of [169]. While some of the subsequent ALMA/interferometric results supported the finding that the number counts decline sharply at the brightest flux densities, implying the existence of an SFR cut-off in the range $1000-2000$ $M_\odot$ yr$^{-1}$ (e.g. [72,142,153]), the more recent AS2UDS survey of approximately 700 SMGs finds no evidence for a steep drop-off in the counts at the bright end as suggested by the first ALMA follow-up of SMGs over smaller areas ([74], figure 6). These latest results suggest that very luminous ($S_{850\,\mu m} \sim 15-20$ mJy) SMGs such as, e.g. GN20 [173,174] and HFLS3 [175], while still rare, may not be as exceptional as otherwise implied.

## 2.2.6. Redshift distribution

One of the other big implications of the robust counterpart identifications allowed by ALMA is for the redshift distribution of SMGs, $N(z)$. This has historically been measured by determining the likely optical counterpart through radio/MIR matching and then calculating photometric redshifts or obtaining spectroscopic redshifts with optical spectroscopy of those counterparts (e.g. [97,176]). Such results may thus be biased against the faintest and/or highest redshift sources—which do not benefit from the negative $k$-correction in the other wavebands—in addition to being dependent on the reliability and completeness of the probabilistic counterpart identification in the first place (§2.2.1).

The precise identifications of large samples of sources with ALMA has allowed the correct counterparts to be targeted, eliminating at least one of these unknowns. This has led to a number of photometric and/or spectroscopic studies of the redshift distribution of SMGs (e.g. [74,107,141,145,147,160,177]). These studies suggest an 850 μm redshift distribution which peaks at $z \sim 2.3-2.65$, only slightly higher than the distributions based on single-dish observations (e.g. [97,176]). However, the median redshift shifts to somewhat higher values if redshift estimates for the approximately $20-30\%$ of sources that are too faint to be seen in the optical/IR are included (approx. $2.5-2.9$; e.g. [13,107,177,178]). Evidence that these undetected sources lie at higher redshifts comes from near- and mid-IR detections with *Spitzer*/IRAC and *Herschel* (e.g. [107]), as well as their redder UV/optical colours [13].

One variable that must be taken into account when comparing different studies is the submillimetre-brightness of the sample, as some studies have suggested that brighter sources tend to reside at higher-redshift ([128,145,154], cf. [177–179]). A dependence on selection wavelength is also expected—both these effects are demonstrated in figure 7. The wavelength dependence may indeed be the main driver for the difference in median redshift observed between unlensed and lensed samples (e.g. [163,194,195], and see §2.3.1). However, the observational constraints on such models are still limited by selection effects. In particular, the (targeted) interferometric follow-up surveys are typically observed to show lower flux density limits than the parent single-dish surveys, complicating the definition of the flux limit. More importantly, even with the correct SMG counterpart(s) identified through interferometry, obtaining spectroscopic redshifts in the optical/IR is still very challenging due to the faintness/dust-obscured nature of the galaxies, resulting in completeness rates for unlensed SMG samples of less than 50% for even the most well-studied extragalactic fields (e.g. [160,196], and to further illustrate the point, note that only 44/707 sources (6%) from the AS2UDS sample of Dudzevičiūtė *et al.* [178] have spectroscopic redshifts). Such optical/IR spectroscopic studies still also miss sources in the so-called 'redshift desert' ($1.4 < z < 2$; e.g. [97]). This highlights the importance and necessity of measuring redshifts through other means, such as blind spectral scans with ALMA (e.g. [163,197]).

Despite the incompleteness in SMG redshift distributions due to the continued reliance on optical/IR redshifts for the ALMA-identified sources, the ALMA-based results suggest the presence of a high-redshift 'tail' in the redshift distribution, with approximately $20-30\%$ of 870 μm-selected SMGs lying at $z > 3-4$ [107,148,160,178]. An increasing number of SMGs have been confirmed to lie at $z > 5$ (e.g. [99,163,198–202]) and even $z > 6$ (e.g. [175,191,203–207]), demonstrating that these massive, highly

---

[5]Note that this hypothesis is regarding the intrinsically bright sources, and not the bright end sources in ultra-wide field surveys which are found to be dominated by lensed sources. The latter, however, can also help inform the debate if the lensing magnification factors are well-constrained (see §2.3.1).

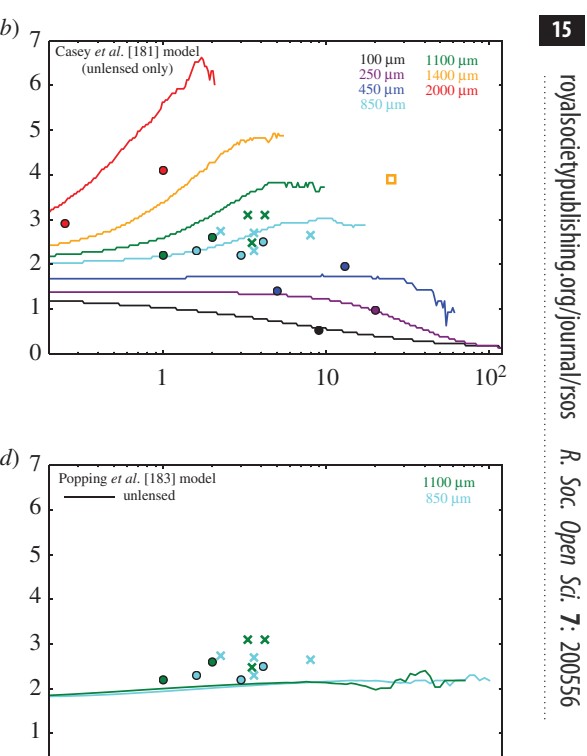

**Figure 7.** Median redshift of (sub-)millimetre selected galaxies as a function of flux density cut. The observational data are indicated by the filled circles (single dish) and crosses (interferometric follow-up) as listed in table 2. The open square indicates the interferometric follow-up of lensed SMGs from Strandet *et al.* [163]. We note that we chose not to include error bars for the observational points because the errors across the literature are not derived in the same consistent manner, and therefore are not comparable. The lines are the predictions from various semi-empirical (*a,b*) and semi-analytic (*c,d*) models. The observational constraints on such models are still sparse and limited by selection effects, but the current constraints on the median redshift from surveys with interferometric follow-up are broadly consistent with single-dish surveys within the uncertainties. (*a*) Model curves are from the empirical model of Béthermin *et al.* [172] and indicate unlensed (solid lines) and lensed (dashed lines) predictions. The predicted impact of strong lensing is evident in the figure and is due to the increased probability of lensing at high redshift. Figure adapted from Béthermin *et al.* [180]. (*b*) Model curves are from Casey *et al.* [181], using a simulation spanning 10 deg² and assuming the Zavala *et al.* [182] description of the high-redshift infrared luminosity function. The model curves cut off when there are fewer than 50 sources in the simulated volume due to increasing noise in the curves. The difference between the Casey *et al.* [181] and Béthermin *et al.* [172] models demonstrates the uncertainty that still exists in the infrared luminosity function at high-redshift. (*c*) Model curves are from the semi-analytic SHARK model of Lagos *et al.* [120]. (*d*) Model curves are from the semi-analytic model of Popping *et al.* [183]. The semi-analytic models show very different results: both models show much weaker evolution of the median redshift with flux density cut. The model of Lagos *et al.* [120] does predict some evolution with selection wavelength, while that evolution is not seen for the Popping *et al.* [183] models for the two available wavelengths. The differences are possibly attributed to different modelling of dust emission in different codes.

star-forming sources were already present when the universe was less than 1 Gyr old. Although their space densities appear to be low (e.g. [202]), their existence nevertheless challenges the hierarchical picture of galaxy growth, which would have been in its very early stages. Moreover, the large amounts of dust in these systems challenge models of chemical evolution, which need to account for the dust enrichment in these very young systems (e.g. [208]). We discuss this further for SMGs in §2.2.7, and for general star-forming galaxies in the epoch of reionization in §3.3.

## 2.2.7. Physical properties of the global SMG population

The final implication of the precise locations and counterpart identification for SMGs now possible in large numbers with ALMA is that the physical properties of these sources can be reliably studied for

**Table 2.** Observational constraints shown in figure 7.

| reference | number of sources | $\lambda_{obs}$ [a] | $S_{lim}$ [a] | $z_{median}$ [b] | follow-up |
|---|---|---|---|---|---|
| | | (µm) | (mJy) | | |
| Berta et al. [184] | 5360 | 100 | 9 | 0.52 | — |
| Béthermin et al. [185] | 2517 | 250 | 20 | 0.97 | — |
| Geach et al. [186] | 60 | 450 | 5 | 1.4 | — |
| Casey et al. [187] | 78 | 450 | 13 | 1.95 | — |
| Chapman et al. [97] | 73 | 850 | 3 | 2.2 | — |
| Wardlow et al. [176] | 72 | 850 | 4 | 2.5 | — |
| Simpson et al. [107] | 77 | 850 | 4 | 2.3 | ALMA |
| da Cunha et al. [13] | 99 | 850 | 4 | 2.7 | ALMA |
| Simpson et al. [177] | 35 | 850 | 8 | 2.65 | ALMA |
| Cowie et al. [146] | 53 | 850 | 2 | 2.74 | ALMA |
| Dudzevičiūtė et al. [178] | 707 | 850 | 3.6 | 2.61 | ALMA |
| An et al. [155] | 897 | 850 | 1.6 | 2.3 | — |
| Smolčić et al. [128] | 17 | 1100 | 4 | 3.1 | SMA |
| Michałowski et al. [188] | 95 | 1100 | 1 | 2.2 | — |
| Yun et al. [189] | 27 | 1100 | 2 | 2.6 | — |
| Miettinen et al. [141,190][c] | 37 | 1100 | 3 | 3.1 | PdBI/SMA |
| Brisbin et al. [145] | 152 | 1100 | 3 | 2.48 | ALMA |
| Strandet et al. [163], Reuter et al. [191] | 81 | 1400 | 25 | 3.9 | ALMA |
| Staguhn et al. [192] | 5 | 2000 | 0.24 | 2.91 | — |
| Magnelli et al. [193] | 3 | 2000 | 1 | 4.1 | — |

[a]Both the observed wavelength and flux density limit are given for the original single-dish survey, even in the case where the sources were identified interferometrically. The limiting flux density refers either to the flux density above which targets were selected for follow-up (if originally single-dish-selected) or the faintest sources in the sample (if not).
[b]Median redshift estimates for the samples are usually heavily reliant on photometric redshifts for individual sources—see §2.2.6.
[c]The median redshift listed is the revised value from Brisbin et al. [145].

the first time. ALMA has therefore enabled an explosion in such studies in its early years. In general, such studies confirm the previously held picture that SMGs are massive (stellar masses approx. $10^{10}-10^{11}\ M_\odot$; e.g. [13,107]) galaxies with high (approx. $10^2-10^3\ M_\odot\ yr^{-1}$; e.g. [13,209]) star formation rates, large (greater than or equal to $10^8\ M_\odot$) dust reservoirs, and a low (approx. 20%) X-ray AGN fraction (e.g. [13,107,160,179,190,209–211]). Da Cunha et al. [13] provide templates from their MAGPHYS SED fitting of the 870 µm-selected ALESS SMGs—see the median template in figure 1 (see also [160,178]).

Unsurprisingly, the average physical parameters observed for the SMGs appear to depend on selection wavelength [179]. The average characteristic dust temperatures are approximately $30-40\ K$, with some studies also reporting a dependence on redshift (e.g. [209,212]; but see Dudzevičiūtė et al. [178], who use a large sample of approximately 700 SMGs from the AS2UDS sample to show that a redshift–temperature relation does not exist at constant infrared luminosity). The SMGs are highly obscured, with average $V$-band dust attenuation values of $A_V \sim 2$ ([13]; cf. Simpson et al. [177] who extrapolate from line-of-sight dust measurements in the infrared and obtain $A_V \sim 500$). Their star formation histories/stellar ages are notoriously difficult to constrain due to this large amount of dust obscuration, though a composite spectrum of the optically detected ALESS sources suggests that they are young (100 Myr old) starbursts observed at 10 Myr [160]. Danielson et al. [160] also find evidence for velocity offsets of up to 3000 km s$^{-1}$ between nebular emission lines (i.e. H$\alpha$, [OII] $\lambda\lambda$3726, 3729, [OIII] $\lambda\lambda$4959, 5007, H$\beta$) and Ly$\alpha$ or UV-ISM absorption lines in ALESS SMGs, suggesting that many are driving winds/galaxy-scale outflows.

While SMGs selected at a particular wavelength tend to have relatively uniform infrared properties (e.g. [13,178])—due no doubt to their selection—sources with similar total FIR luminosities show a

wide variety of UV/optical/near-IR and mid-IR characteristics [160,178]. SED modelling by da Cunha *et al.* [13] showed that the optically faint SMGs tend to have similar overall properties to the optically brighter sources in their sample, but with significantly higher values of dust attenuation. This could indicate that these sources (which also seem to lie at higher average redshift) are either more compact [178], or more likely to be edge-on than the optically brighter sources.

In comparison with local ULIRGS, da Cunha *et al.* [13] find that the average properties of SMGs are generally similar. Their average intrinsic SED is also similar to local ULIRGS in the infrared range, though the stellar emission of the average SMG is brighter and bluer. This difference suggests a lower average dust attenuation despite similar dust masses (e.g. [213]), which could be due to the fact that high-redshift SMGs may be more extended than local ULIRGs and/or the dust and stellar distributions are not co-located (e.g. [214]). This interpretation would also be consistent with the lower characteristic dust temperatures found for similarly luminous 870 μm-selected sources [177,209]. These differences demonstrate that local ULIRGs are not perfect analogues of the high-redshift SMG population—a claim that is still repeated quite frequently in the literature.

*Global radio/CO properties.* Although the gas fractions implied for SMGs are large (approx. 40%; e.g. [178,179,209]), these are still derived mainly through the dust mass (assuming a constant gas-to-dust ratio; §3.1.2) for statistically significant samples. Follow-up work on CO-based gas masses still suffers from the lack of spectroscopic redshifts for many of the sources (e.g. [160]), requiring more time-intensive spectral scans (though those spectral scans can often deliver both the redshifts and CO lines at once; e.g. [215,216]). In the radio, studies of SMG counterparts report a median synchrotron spectral index of $\alpha \sim -0.8$ (e.g. [190,217,218]), consistent with the canonical synchrotron value. Those studies find a median FIR-radio correlation parameter of $q_{IR} \sim 2.2 - 2.6$ (depending on the selection), with no evidence for evolution with redshift, at slight odds with results for less-extreme star-forming galaxies (e.g. [219]) and theoretical predictions [220]. Thomson *et al.* [218] also report the first observational evidence that $\alpha$ and $q_{IR}$ evolve in a co-dependent manner with stellar age, which is the behaviour predicted for starburst galaxies [221]. More detailed studies of the resolved properties of these sources in the various tracers will be discussed in the context of §3.2.

*SMGs on the so-called 'main sequence'?* In the light of the growing statistical samples and more robustly derived physical parameters thanks to the reliable counterpart identification, there has been a significant amount of ongoing debate in the literature on the position of SMGs with respect to the so-called 'main sequence' of star-forming galaxies, i.e. the correlation between stellar mass and star formation rate observed for (mainly mass-selected) galaxy samples from low-redshifts to $z > 3$ (e.g. [4,7]). Placing individual SMGs accurately on the main sequence is challenging because of the large systematic uncertainties associated with deriving their stellar masses from very faint and often poorly sampled rest-frame SEDs. Uncertainties associated with unknown star formation histories and dust obscurations alone can change stellar masses by up to a factor of 10 (e.g. [13,178,222]). A relatively smaller uncertainty in SFR comes from potential dust heating by relatively old stellar populations, which could affect the SFRs derived directly from IR luminosities by a factor of approximately 2 (e.g. [13,115]). Uncertainties in the output by young massive stars, the IMF and possible contribution by AGN could increase uncertainties further. These errors are important to keep in mind when trying to establish if an individual SMG is offset from the main sequence by a factor of 3 or so—the typical factor often used in such studies to define 'starbursts', i.e. outliers with significantly higher SFRs for their stellar masses. Additional uncertainties come into play if the redshift of the source is not known robustly, since the normalization of the main sequence evolves with redshift (e.g. [223–226]). Finally, another important caveat of these comparisons is that the location of the main sequence *itself* at different stellar masses for a given redshift is not uniquely established (see §3.1.2). For example, some studies measure a downturn of the SFR-stellar mass relation at high masses (e.g. [225]), while others do not (e.g. [223]; figure 12). Even taking into account all of these uncertainties, and despite the popularity of the main sequence in the recent literature, the true connection between galaxies' evolutionary drivers (i.e. mergers versus secular evolution) and their position on this particular plot has yet to be robustly established either observationally or theoretically.

Nevertheless, some recent studies have attempted to take samples of SMGs for which the counterparts are robustly identified thanks to ALMA, and place them on the star-forming main sequence (figure 8; e.g. [13,148,160,179]). This is partly motivated by hydrodynamic simulations suggesting that SMGs simply make up the most massive end of the high-redshift star-forming main sequence (e.g. [115,116]). Bearing in mind the uncertainties described above, these studies find that SMGs generally show a spread in properties, with some being on the main sequence (typically at the high-mass end), and some being outliers (with higher SFRs than main sequence galaxies of the same stellar mass, in the regime often

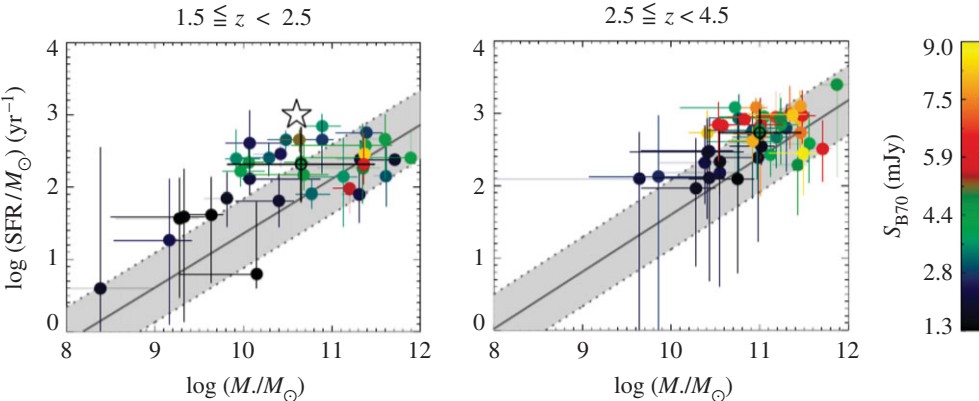

**Figure 8.** Relation of the stellar masses and star formation rates of the ALMA-identified ALESS SMGs to the star-forming 'main sequence' in two different redshift bins. At $z \sim 2$, approximately 50% of the SMGs appear to be consistent with the main sequence, and this fraction increases with redshift. Although the properties of the SMGs are now better constrained thanks to the precise counterpart identifications enabled by ALMA, it is important to realize that a number of remaining systematic uncertainties regarding the stellar masses, star formation rates, and location of the main sequence itself make the placement of any individual SMG in relation to the main sequence highly uncertain. Figure from da Cunha et al. [13].

attributed to starbursts)—again indicating a non-homogenous population. There is also some evidence that the fraction of SMGs on the main sequence increases with redshift (e.g. [13,148,178,227]; but cf. Miettinen et al. [179]). On the other hand, there is still disagreement even within SMG samples about where the galaxies lie (e.g. Danielson et al. [160] examine the subset of ALESS SMGs with spectroscopic redshifts and conclude, unlike da Cunha et al. [13], that they lie a factor of 5 above the main sequence on average). This discrepancy is probably due to a combination of the systematics discussed above and selection effects, and it emphasizes the skepticism with which current plots of SMGs in relation to the 'main sequence' should be viewed. Future near-/mid-IR observations of the obscured stellar populations with the James Webb Space Telescope (*JWST*) will hopefully shed light on the stellar masses of these galaxies, and thus their actual relation to the general population of less dust-obscured galaxies.

*Hierarchical context.* Finally, the more robust physical parameters derived for the interferometrically located SMGs has enabled their global comparison with other galaxy populations in order to try to place them in the broader context of massive galaxy evolution. In particular, an evolutionary pathway has been suggested wherein SMGs evolve into local elliptical galaxies via $z \sim 2$ compact quiescent galaxies (e.g. [108,228–231]). By making assumptions about the length of the SMG phase and the subsequent evolution of the stellar populations, Simpson et al. [107] calculated their expected *H*-band luminosity distribution and space density at $z = 0$, finding general agreement with a morphologically classified sample of local elliptical galaxies. A similar analysis of the spheroid mass and space density of SMG descendants led Simpson et al. [177] to conclude that SMGs must be the progenitors of local elliptical galaxies (as proposed much earlier by Eales et al. [90] and Lilly et al. [229]). Meanwhile, Garcia-Vergara et al. [159] used an ALMA-observed sample of SMGs to re-examine their connection to these populations via clustering. As the latest piece to the puzzle, high-resolution ALMA imaging allows this question to be addressed using the physical extent of the submillimetre emission and the size–mass relation. This will be discussed further in §3.2.

## 2.3. Strongly lensed sources

### 2.3.1. Confirmation of lenses en masse

So far in this review, we have been primarily discussing *unlensed* SMG samples. However, it has long been suspected that some of the brightest submillimetre sources detected at long wavelengths ($\lambda > 500\,\mu m$) are experiencing strong gravitational lensing by massive foreground galaxies and clusters [232]. This is due both to their high redshifts and the steepness of the intrinsic SMG number counts. The former means that SMGs have an increased probability of being in alignment with a massive foreground object, and the latter means that a cut in flux density alone should efficiently select these lensed sources once low-redshift galaxies ($z < 0.1$) and radio-bright AGN at higher-redshift are taken

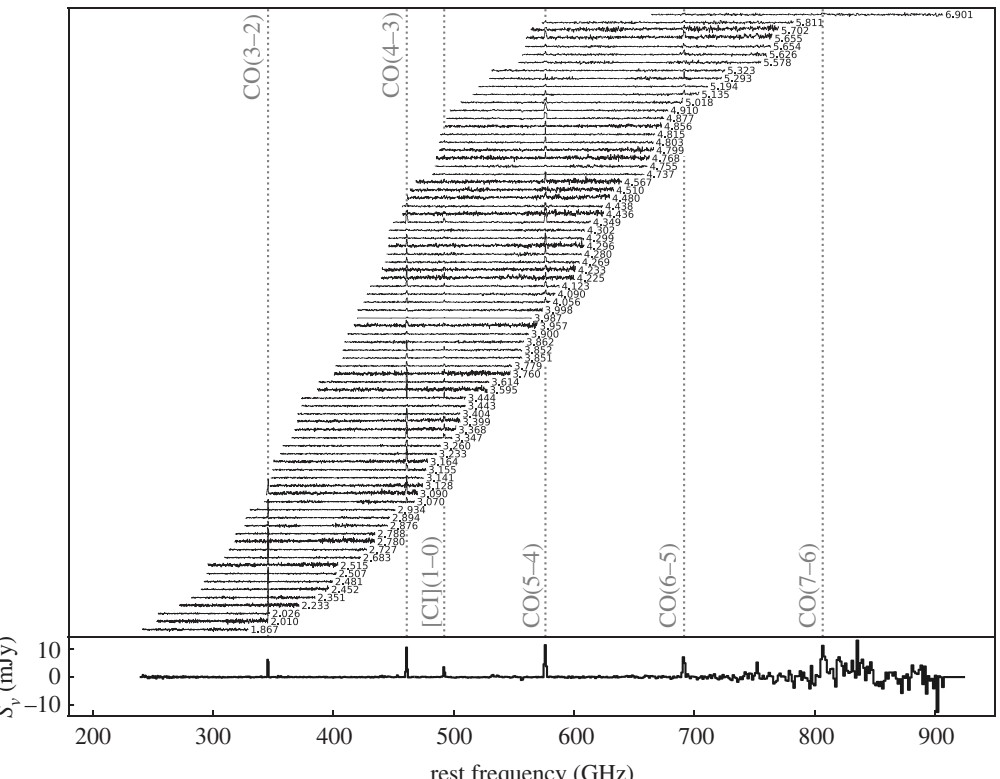

**Figure 9.** ALMA 3 mm spectra of 78 SPT sources from Reuter *et al.* [191]. Spectra are continuum subtracted and offset for clarity. Each spectrum is labelled with the derived ALMA redshift. Such work on strongly lensed sources has helped demonstrate ALMA's utility as a redshift machine for bright dusty sources at high-redshift. The bottom panel shows the stacked spectrum. Figure courtesy of C. Reuter and the SPT Collaboration.

into account (e.g. [233]). Although such bright objects are quite rare (e.g. the space density of SPT-SZ sources is approximately 1 per 30 deg$^2$; [234]), wide-area surveys with *Herschel* (e.g. H-ATLAS, HerMES), the South Pole Telescope and the *Planck* mission have returned large numbers of these extreme sources with relatively fast survey speeds [163,197,233,235–237], which, based on the luminosity function of SMGs, should then efficiently select strongly lensed sources (e.g. [232,238]).

The strongly lensed nature of a small number of these SMGs was confirmed already using imaging with pre-ALMA interferometers [233,239–243], or inferred indirectly through comparison with, e.g. the empirical luminosity–linewidth relationship [244]. However, one of the biggest results from early ALMA observations of distant galaxies was the confirmation of lensed SMGs en masse. In particular, early 0.5″-resolution ALMA imaging of SPT sources revealed ring-like structures in a high fraction of sources, showing that they have a high probability of being strongly lensed [143,237,245]. The few intrinsically very bright (unlensed) sources that do exist appear to be associated with SMG mergers (e.g. [246,247]). Spectral scans with ALMA were then used to determine CO-based redshifts for the sources, demonstrating ALMA's utility as a redshift machine for bright dusty sources ([163,197,191,237], see figure 9 for the latest compilation of SPT spectra).

High (less than 1″) resolution imaging from either optical/UV observations or submillimetre interferometry is necessary in order to make accurate lensing models of strongly lensed sources. However, (sub-)millimetre bright sources can be very dim in the optical/UV due to extinction. Consequently, the sub-arcsecond-resolution ALMA imaging of strong lens candidates has allowed lens models to be derived for dozens of sources for the first time (e.g. [143,144,245]). For the SPT sample, the median magnification factor is $\mu = 6.3$ for all sources with resolved ALMA data available, with the most extreme sources magnified by $\mu > 30$ [143]. The *Herschel* sources were already known (thanks to SMA) to have an average magnification factor of only approximately 6 [240], meaning that they are also *intrinsically* bright. These magnification factors are at odds with model predictions assuming the intrinsic number counts based on ALESS [240]. This discrepancy may be partly reconciled by more recent measurements of the 870 µm number counts over larger areas (figure 6) and/or the suggestion—based on the significantly higher resolution ALMA imaging of SDP.81—that lens models based on

lower-resolution data may underestimate the magnification factors by a factor of less than approximately 2 [248–250]; even a factor of 2 can be important given the steepness of the bright end of the number counts.

Before taking into account lensing corrections, the ALMA spectral scans of the SPT sample demonstrated that the sources lie at very high redshift on average, with a median of $z = 3.9 \pm 0.4$ [163,197]. This has prompted a lot of discussion in the literature, as it is significantly higher than the median redshift found for unlensed samples (§2.2.6). There have been various explanations proposed for this discrepancy which invoke a combination of the selection wavelength (e.g. [36], figure 7), survey depth and the redshift-dependent probability of strong lensing, where the latter may theoretically be affected further by size evolution (e.g. [197,251]). While the number of high-redshift ($z > 4$) sources with size measurements is still small, the best current studies do not find evidence for significant size evolution [107,252,253,254]. A phenomenological model by Béthermin *et al.* [194] suggested that the higher median redshift of the SPT sample could be explained by a combination of lensing probability and selection wavelength (figure 7)—a theory which is supported by the latest SPT redshift distribution results [163].

### 2.3.2. High-redshift ISM physics

Strong lensing in the (sub-)millimetre provides a unique opportunity to study the dusty star formation and star-forming ISM in distant galaxies in unprecedented detail. In particular, the physics of lensing magnification by a factor of $\mu$ provides a boost of $\mu$ in total brightness and $\sqrt{\mu}$ in (physical) angular resolution. This has allowed studies with ALMA to move beyond the typical molecular and atomic gas tracers studied in high-redshift sources (e.g. CO, [CII]) and on to other (fainter) emission/absorption lines in the (sub-)millimetre which are generally too challenging to detect/resolve in unlensed sources. The wealth of spectral features detectable in high-redshift sources with ALMA was first demonstrated by Spilker *et al.* [255] using the stacked spectrum of SPT sources, which boasts a total of 16 $S/N > 3$ spectral lines and places the first constraints on many other molecular species at high-redshift (for previous *Herschel* studies at shorter wavelengths, e.g. [256,257]).

The detection of 'non-traditional' spectral lines at high-redshift opens up an entirely new window into the ISM properties of distant star-forming galaxies. Here, we briefly summarize some of the classes of spectral lines detectable with ALMA. We have chosen to discuss these lines in the context of strongly lensed sources, as real progress on detailed studies of many of these lines will continue to be feasible in only the brightest and/or strongly lensed star-forming galaxies at high redshift, even with the full ALMA capabilities. However, we note that ALMA has also allowed some of these lines to be detected for the first time in unlensed star-forming galaxies (see §3.2.9).

— *Dense gas tracers.* While CO is typically the most easily detectable molecule in high-redshift star-forming galaxies, the relatively low critical density required to collisionally excite the lower-$J$ transitions ($n_{H_2} \sim 10^2 - 10^3$ cm$^{-3}$) implies that CO is not a reliable tracer of the dense molecular cloud cores where star formation actually occurs. Molecules with higher critical densities ($n > 10^4$ cm$^{-3}$; e.g. HCN, HNC, HCO+, CN, etc.) are thought to be much more robust tracers of the molecular gas ultimately fuelling star formation, with some studies suggesting that the ratio of HCN to SFR remains linear over more than eight decades in HCN luminosity [258–260]. Such dense gas tracers have been previously detected at high-redshift prior to ALMA, but only two objects had been detected in *multiple* transitions/species—both strongly lensed quasars: the 'Cloverleaf' quasar at $z = 2.56$ [261–266], and the APM 08279+5255 quasar at $z = 3.91$ [267–270]. Although they are typically 1–2 orders of magnitude fainter than CO lines (e.g. [258,271–273]) such lines will become increasingly important in the ALMA era thanks to ALMA's increased sensitivity and large bandwidth. In particular, several recent multi-line studies of strongly lensed star-forming galaxies with ALMA use these line ratios to constrain the typical density, temperature and excitation conditions within the star-forming ISM (e.g. [255,274]).

— $H_2O$. While technically also a dense gas tracer, water ($H_2O$) holds a special significance, as it is thought to be one of the most abundant molecules in molecular clouds (either locked up in icy dust grain mantles or in the gas phase depending on local conditions; [275]) and it is an important ISM line in dust-obscured galaxies (e.g. [276]). High excitation water lines (up to 500 K above the ground state) can be as luminous as CO lines in the same frequency range; they are radiatively excited by the local infrared radiation field (in the 50–200 µm range), and therefore they are a tracer of the local radiation field intensity and colour [276]. Water line observations of the brightest lensed SMGs started with the PdBI/NOEMA with studies by Omont *et al.* [277,278]

and Yang *et al.* [279] (see also Riechers *et al.* [175], for a multi-observatory detailed study of a maximum-starburst galaxy at $z = 6.34$, where multiple $H_2O$ lines are detected, allowing for a detailed study of its excitation mechanisms). Recently, ALMA has enabled high-resolution observations of this molecule. A high-resolution observation of thermal $H_2O$ in an extragalactic source was achieved with the $0.9''$ detection in the strongly lensed source SDP.81 during the ALMA 2014 Long Baseline Campaign [280], recently superseded with approximately $0.4''$-resolution observations of the strongly lensed merger G09v1.97 at $z = 3.63$ by Yang *et al.* [281]. Other work with ALMA is in progress to calibrate $H_2O$ as a resolved star formation tracer (e.g. [282]).

— *CO isotopologues.* CO isotopologues $^{13}CO$ and $C^{18}O$ are typically more optically thin than $^{12}CO$, making them useful as tracers of the total molecular column density. In addition, the carbon and oxygen isotopes have different formation pathways, and the ratio of these lines with $^{12}CO$ can then provide insight into high-redshift nucleosynthesis. As with the dense gas tracers, detections of multiple transitions/species in the pre-ALMA literature are sparse (e.g. [283,284]). However, based on the multiple transitions from $^{13}CO$ detected in their stacked spectrum, Spilker *et al.* [255] estimate that ALMA will be able to detect (and even resolve) these faint lines in an (admittedly bright) $L_{IR} = 5 \times 10^{13} \, L_{\odot}$ galaxy in only 30 min per line. This could open up a new window into the cosmic isotope enrichment history, including providing a dust-insensitive probe of the stellar initial mass function (IMF; as proposed by Romano *et al.* [285,286]). Indeed, Zhang *et al.* [287] find low $^{13}CO/C^{18}O$ abundance ratios for a sample of four strongly lensed SMGs at $z \simeq 2-3$ observed with ALMA and, based on the models of Romano *et al.* [285,286], argue that these ratios imply top-heavy IMFs in high-redshift SMGs. This would be consistent with the results from the earliest attempts at modelling SMGs in the cosmological context [110], though more recent models can reproduce submillimetre number counts without the need to invoke a top-heavy IMF [120]. We note, however, that linking CO isotopologue line observations to isotope abundances and thus conclusions about the stellar IMF relies on several assumptions (e.g. about line excitation, stellar yields, etc.) and more work would be helpful to confirm these results.

— *Atomic fine structure lines.* The class of atomic fine structure lines includes some of the brightest FIR emission lines in a star-forming galaxy's spectrum, many of which have also been detected in unlensed galaxies (see table 1 of [33] for a summary of IR fine structure lines). Singly ionized carbon ([CII] at 158 μm), in particular, is often the strongest line in the long-wavelength spectrum of star-forming galaxies, and it is now routinely detected and resolved with ALMA in both lensed (e.g. [288]) and unlensed (e.g. [212,289,290]) star-forming galaxies. ALMA has also allowed the first detections of [NII] and [OIII] 88 μm in unlensed high-redshift galaxies (e.g. [291–294], and see §3.4). The [OIII] 88 μm line further holds the promise that in low-metallicity galaxies ($Z < 1/3Z_{\odot}$), it can be approximately up to three times brighter than [CII] [295]. Aside from the implications for the highest-redshift (primeval) galaxies, which are discussed further in §3.2.9, strong lensing has allowed some of the first statistical studies of these important tracers. For example, Bothwell *et al.* [296] presented a study of the ground-state transition of atomic carbon ([CI]) in 13 strongly lensed SPT sources in the range $2 < z < 5$. As [CI] has been proposed as a good tracer of the cold molecular ISM (e.g. [297–301]), it has been suggested to be an excellent proxy for the (unobservable) $H_2$ mass. Bothwell *et al.* [296] used this assumption to derive [CI]-based gas masses in their sources, finding significant tension with low-*J* CO-based estimates that would suggest a denser, more carbon-rich medium in these sources than observed in local starbursts.

— *Molecular absorption lines.* Since the strength of molecular absorption lines is not diluted with distance—depending only on the brightness of the background source—such lines are very sensitive to small amounts of molecular gas along the line of sight. As such, they can be important tracers of the molecular ISM and signposts of molecular outflows. As of a decade ago, there were only five sources detected in absorption beyond the local universe, and these absorbers were all still at $z < 1$ [302]. *Herschel*/SPIRE enabled a few pre-ALMA detections of OH absorption in some of the brightest lensed high-redshift SMGs (e.g. [257,303]). Thanks to the capabilities of ALMA, molecular absorption studies are now possible at increasingly high redshifts (e.g. [304]). For example, molecular absorption has now been detected and spatially resolved via the rest-frame 119 μm ground-state doublet transition of the hydroxyl molecule, OH, within a strongly lensed starbursting galaxy at $z = 5.3$ [305]. This detection provides evidence for self-regulating feedback, with the fast molecular outflow indicated by the OH observation capable of removing a large fraction of the star-forming gas. Moreover, ALMA studies of strongly lensed sources have also enabled the detection of new molecules at high-redshift, such as the ground-state transition of the methylidyne cation, $CH^+$, which was detected in both absorption and emission in six $z \sim 2.5$

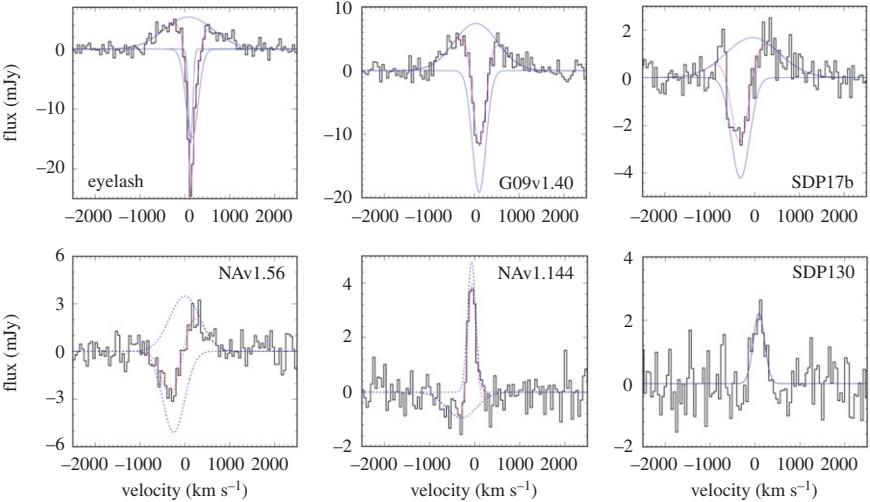

**Figure 10.** Detection of the methylidyne cation, $CH^+$, in both absorption and emission in six $z \sim 2.5$ lensed starbursts. The combination of ALMA's sensitivity and strong gravitational lensing has allowed this molecule to be detected in the high-redshift universe for the first time, highlighting the role of turbulence in the gas reservoirs of these galaxies. Figure reproduced from Falgarone *et al.* [306].

lensed starbursts (figure 10; [306]). This unique observation highlights the role of turbulence in regulating star formation and suggests that feedback, when coupled to this turbulence, extends the starburst phase rather than quenching it. ALMA has also recently detected the ground-state transitions of $OH^+$ and $H_2O^+$ in absorption towards two $z \sim 2.3$ lensed SMGs, which can be used to measure the rate of cosmic ray ionization in their extended gaseous halos, and from that infer the ionization rate in dense star-forming regions, closer to the sites of cosmic ray acceleration [307].

### 2.3.3. kpc- and pc-scale studies

In addition to the boost in brightness that allows many different gas tracers to be detected at high-redshift, strongly lensed sources experience a $\sqrt{\mu}$ boost in physical resolution. Combined with the high angular resolutions already provided by ALMA, this can result in image-plane resolutions as high as tens of parsecs. The technical feasibility of such observations was first demonstrated during the ALMA 2014 Long Baseline Campaign with the multi-band imaging of the $z = 3.4$ SMG SDP.81, which resulted in visually impressive Einstein rings at an unprecedented angular resolution of 23 milliarcseconds (figure 11; [248,249,280,308]). We note that all of the targets for the Long Baseline Campaign were chosen specifically to demonstrate the suitability of the long baseline capability [54], and that even despite the relatively compact size of SDP.81's Einstein ring ($\theta_E \sim 1.5''$), a large amount of total observing time was required (approx. $9-12$ h per band) in order to achieve good $uv$-coverage [280]. As a result, high-resolution ALMA imaging of this quality is still relatively uncommon.

From the source plane reconstructions, ALMA imaging of strongly lensed sources with sufficiently good image quality allows detailed investigations of the dusty star formation and ISM on scales that are rarely achieved outside of local galaxy studies. For example, in SDP.81, various analyses of the Long Baseline Campaign data suggest a non-uniform dust distribution with clumps on scales of approximately 200 pc situated in a more extended cold gas disc [248,249,308], and with an offset from the near-infrared emission similar to that previously seen in the $z = 4.05$ SMG GN20 [48,49]. Dye *et al.* [248] and Swinbank *et al.* [309] argue that the disc is rotationally supported, while Rybak *et al.* [308] report evidence from a kinemetry analysis for significant asymmetry at large radii, suggesting a perturbed disc with multiple velocity components. The low value derived for the Toomre stability parameter ($Q \sim 0.3$; [248,309]) suggests an unstable disc. Swinbank *et al.* [309] compare the scaling relations observed between luminosity, line-widths and sizes, finding evidence for an offset from local molecular clouds that can be attributed to an external hydrostatic pressure for the interstellar medium that is approximately $10^4 \times$ higher than the typical pressure in the Milky Way, and Rybak *et al.* [310] use photon-dominated region (PDR) models to further constrain the physical conditions of the star-forming gas. The unprecedented SDP.81 data also allowed a study of dark matter substructure in the foreground lens halo itself [311], which is a separate topic beyond the scope of this review.

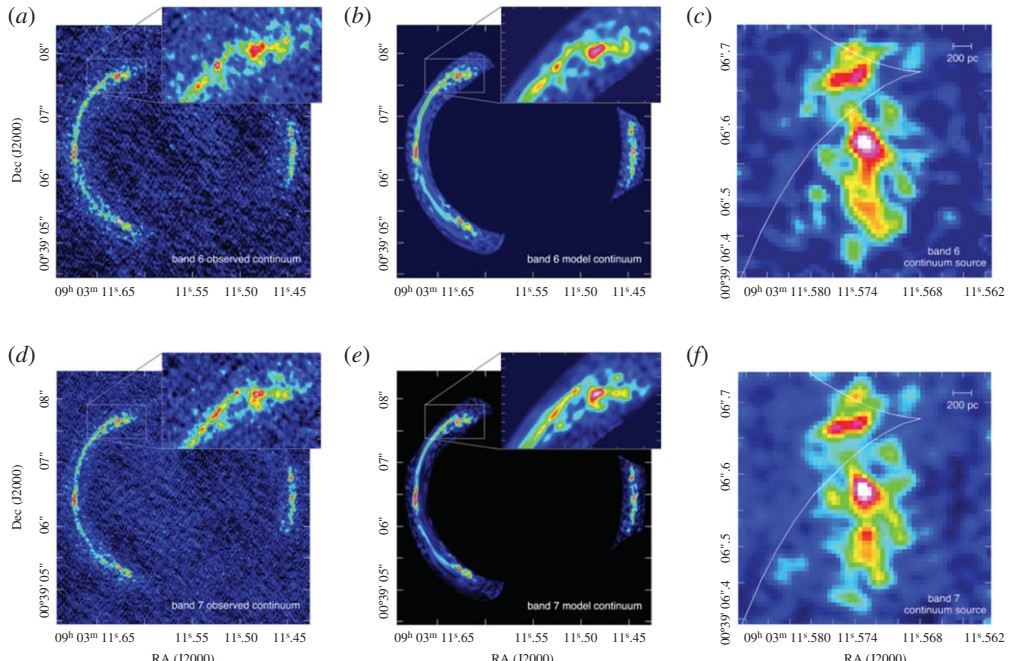

**Figure 11.** ALMA band 6/7 continuum imaging and source-plane reconstruction of the strongly lensed $z = 3.4$ SMG SDP.81 from Dye *et al.* [248]. The highest angular resolution reached is $31 \times 23$ mass for the Band 7 data, corresponding to an un-magnified spatial scale of 180 pc, and representing a factor of approximately 20–80 increase compared with previous SMA and PdBI imaging of this source. The white lines in the right-hand panels represent the lensing caustic. Figure reproduced from Dye *et al.* [248].

One area of significant interest in observations of high-redshift star formation is the use of resolved (sub-galactic) data to study the relative efficiency at which gas (traced by CO) is transformed into stars within individual galaxies (i.e. 'Kennicutt–Schmidt' relation; [312–314]). A handful of such studies have been done in very bright and/or lensed galaxies using other radio/(sub-)millimetre facilities [49,51,283,315–319], and ALMA observations of lensed sources have pushed these studies further (e.g. [320–325]), in some cases to individual star-forming 'clumps'. For example, Sharda *et al.* [324] used the high-resolution ALMA data on SDP.81, along with one of the individual resolved star-forming regions identified by Swinbank *et al.* [309], to test various star formation models, arguing that a multi-freefall (turbulence) model [326] best fits the data. They found similar results in the more recent analysis of two star-forming clumps in the bright (unlensed) AzTEC-1 SMG at $z \simeq 4.3$ [323], suggesting that the high SFR in high-redshift starbursts is sustained by an interplay between gravity and turbulence. Meanwhile, Dessauges-Zavadsky *et al.* [321] used 30 pc ALMA mapping of the CO(4–3) emission in the $z = 1.036$ 'Cosmic Snake' to identify 17 molecular clouds in this Milky Way progenitor. They measured the masses, surface densities and supersonic turbulence implied by these clouds, reporting values 10–100 times higher than present-day analogues, and bringing into question the universality of GMCs. It is important to note that the Cosmic Snake has one of the largest magnification factors known for a giant arc ($80 \pm 10$; [327]), making this sort of study rare even in the era of ALMA.

Given its brightness and rest frequency, the [CII] line can be significantly easier to detect and resolve in high-redshift galaxies with ALMA than the CO lines. This includes those magnified by strong gravitational lensing, and it means that progress has recently been made in understanding the origin of the so-called '[CII] deficit', where the $L_{[CII]}/L_{FIR}$ ratio can show a marked decrease for galaxies with a total $L_{FIR} \gtrsim 10^{11} L_\odot$ (e.g. [328–331]). In a first step, Spilker *et al.* [143] looked at the integrated properties of strongly lensed SPT sources, finding that they followed the same relation between $L_{[CII]}/L_{FIR}$ and $\Sigma_{FIR}$ as local galaxies from Díaz-Santos *et al.* [332]. Thanks to ALMA's sensitivity and angular resolution, work in this and other areas pertaining to the ISM physics and resolved properties of high-redshift galaxies has been complemented by advances in studies of unlensed galaxies and will be discussed more generally in §3.2.

### 2.3.4. Source reconstruction techniques and lensing systematics

Concurrently with the progress in strong lensing observations, the field has seen advancement in source reconstruction techniques. In particular, in addition to modelling ALMA data of lensed sources in the

image plane (e.g. [248]), various groups have developed codes to do the lens modelling directly in the *uv*-plane [240,245,249,308]. The latter has the advantage that it includes self-calibration-like antenna phase corrections as part of the model optimization, thus incorporating the full range of uncertainty present in the measurements. The exact way this reconstruction is done differs between the codes, with Hezaveh *et al.* [245] and Bussmann *et al.* [240] assuming a parametric form for the background source (multiple Gaussian or Sérsic profiles), and Rybak *et al.* [249,308] using a Bayesian pixellated reconstruction technique that extends earlier work by Vegetti & Koopmans [333] to interferometric data (see also [311,334]). The latter technique can help capture the complex surface brightness distributions revealed by high-resolution ALMA data.

Regardless of the technique, it is important to note that differences in the size and structure of a source at different wavelengths can lead to differential magnification (e.g. [335,336]). This was demonstrated, for example, by Spilker *et al.* [337], who used ALMA and Australia Telescope Compact Array (ATCA) observations to show that the difference in extent between 870 μm dust continuum emission and cold molecular gas traced by low-J CO in their sources (see also §3.2.1) causes up to 50% differences in the respective magnification factors. It is also good to keep in mind that gravitational lensing preserves surface brightness. Thus, even though (flux-limited) lensed samples are expected to be biased toward more compact sources to begin with ([251], cf. [143,288]), reaching the highest resolutions possible with ALMA still requires good surface brightness sensitivity, and thus correspondingly good *uv*-coverage. It is for this reason that much of the highest-resolution work on lensed sources still focuses on SDP.81, at least until large time allotments are granted to other sources.

## 2.4. What defines an SMG in the ALMA era?

This section would not be complete without a discussion of what constitutes an SMG in the ALMA era. On the one hand, this is simply and purely an argument of semantics. On the other hand, since many people have the tendency to associate labels with the underlying physical properties, even a semantics argument can hold importance. And the semantics in question here could use some clarification.

In particular, many of the ALMA studies presented in this section that began by targeting single-dish-selected sources have continued to refer to the new ALMA-detected sources as SMGs, even in cases where the ALMA flux limit is significantly fainter than the original single-dish detection limit (e.g. [140,145]). These 'faint SMGs', which can have 870 μm flux densities down to approximately 1 mJy, are analysed along with the rest of the population in terms of redshift distribution and detailed source properties. At the same time, ALMA studies that have initially selected their sample in other ways (via stellar mass, multi-wavelength colours, or e.g. 'compactness') may detect galaxies that are as equally submillimetre-bright as (or *brighter* than) the 'faint SMGs' from other studies, but they are not referred to as 'SMGs' given their different initial selection (e.g. [64,76,338]). There can be significant overlap between these populations, both in terms of physical parameters (high SFRs, significant dust obscuration), as well as literal overlap (for example, three of the Franco *et al.* [64] sources are also identified as ALESS SMGs in Hodge *et al.* [140]).

While the overlap itself is not a problem, the confusion comes when the use of the term 'SMG' (or lack thereof) is equated with the starburst-vs-main-sequence dichotomy. As was shown in §2.2.7, the galaxies referred to as SMGs do not necessarily lie above the main sequence, at least within the significant uncertainties inherent in such a plot. Taken at face value, a significant fraction of the SMGs are also 'main sequence' galaxies. Conversely, using the term 'main sequence' to describe a bright, massive ALMA-detected galaxy initially selected in some other way does not mean that it could not also be identified as an 'SMG' based solely on its submillimetre brightness. It is for this reason that, following a discussion of samples selected in other ways, we discuss resolved properties of ALMA-detected star-forming galaxies altogether in §3.2, regardless of their original selection.

For future reference, we propose that the term 'SMG' is used based on a purely observational definition: i.e. *an SMG is a galaxy with a high submillimetre flux density:* $S_{850\mu m} \gtrsim 1$ mJy. One should not attach any '*a priori*' physical meaning to this definition, particularly in terms of whether these sources are on the main sequence or not, as discussed above, and what that means in terms of physical processes shaping their evolution. Within this definition, SMGs may be on the main sequence or they may equally be outliers, and they may also have been previously detected at other wavelengths; if the flux density in the submillimetre is brighter than about 1 mJy, it is an SMG. This is simply a qualifier that tells us about submillimetre brightness, and it does not necessarily preclude a galaxy to be classified in other ways based on additional data (e.g. an SMG can be found to be a massive galaxy, a

merger, an AGN). Incidentally, we note that following this definition, SMGs are a rare enough population that they are not typically detected in random ALMA pointings.

# 3. ISM properties of galaxies at cosmic noon and beyond

Prior to ALMA, studies of the molecular gas content and resolved properties of $z > 1$ star-forming galaxies were typically carried out with the IRAM Plateau de Bure Interferometer (PdBI) and the Karl G. Jansky Very Large Array (VLA) on select samples targeted either as SMGs, or via colour- or mass-selection (e.g. the IRAM Plateau de Bure HIgh-z Blue Sequence Survey (PHIBSS) 1 and 2 surveys; [45,339]). ALMA is now enabling increasingly detailed studies of the dust and molecular content in the overall high-redshift galaxy population, including through large targeted surveys of galaxies over a wide redshift range, (sub-)kpc imaging of the ISM in galaxies at cosmic noon, and dust continuum and ionized gas detections well into the epoch of reionization. In this section, we review some of the most important recent results enabled by ALMA on both statistical and resolved studies of star-forming galaxies from $z \simeq 1$ to the epoch of reionization. For other recent reviews of cool gas in high-redshift galaxies, we direct the reader to Carilli & Walter [33] and Combes [81].

## 3.1. Statistical studies of the molecular gas content

Deep optical and infrared surveys in the last few decades have allowed us to measure the star formation rate and stellar masses of large samples of galaxies out to high redshifts. A major result arising from these surveys is the measurement of the assembly of galaxies across cosmic time via the evolution of the cosmic star formation rate density as a function of redshift (e.g. [2]). We know from this measurement that the star formation rate density of the Universe ramps up from the epoch of reionization (cosmic dawn) to the cosmic epoch at around $z \simeq 2$ (cosmic noon), where we see a peak in cosmic density of star formation, meaning that this was a key epoch of galaxy formation and evolution. From then, the overall cosmic star formation rate density slowly declines to $z = 0$.

In addition, as previously mentioned, observations indicate that the bulk of star-forming galaxies at a given redshift seem to follow a tight relation in the stellar mass versus star formation rate plane, in the sense that more massive galaxies are forming stars at higher rates, the so-called 'star-forming main sequence'[6] (e.g. [340–342], figure 12*a*). The normalization of this relation evolves with redshift out to at least $z \simeq 5$ (e.g. [7,223,347,348], figure 12*b*), in the sense that the overall specific star formation rate (sSFR) of galaxies increases towards higher redshifts. There is also an indication that the slope of the main sequence may vary with time [223], and as a function of stellar mass, with a possible turnover to shallower slopes at high stellar masses (e.g. [224,225,343]). At all redshifts, outliers to this tight relation are observed (typically around 2% of mass-selected populations; [6,349]); in these galaxies, often denoted 'starbursts', the observed SFR is enhanced relative to the main sequence at their stellar mass. A major goal of current studies is to link these observed behaviours of the galaxy population as a whole across cosmic time with the current picture where galaxy evolution is governed by gas consumption and stellar mass growth via star formation, and gas ejection via feedback processes, with the gas supply coming either from steady accretion from the cosmic web, major and minor mergers, or a combination of these processes [350]. It has been suggested that the tightness of the main sequence (if real; we note that the real dispersion of the relation is still under debate, as it depends significantly on selection effects and measurement methods) implies that the star formation rates of galaxies in that sequence are governed by steady gas accretion. Outliers (starbursts) could be explained by more violent stochastic processes such as major gas-rich mergers where the gas is rapidly channelled to feed a central starburst via loss of angular momentum, or they could obey a different star formation law, or present higher star formation efficiencies, or a combination of all these factors.

A key quantity that needs to be measured in order to shed light on this topic is the gas content, which enables investigations of the gas fraction and star formation efficiency (or depletion time) in galaxies as a function of redshift, stellar mass and star formation rate. Thanks to its sensitivity and frequency coverage, ALMA is the prime instrument for this, although significant work in this field was pioneered using the IRAM/PdBI (e.g. [35,39,351–354]), with parallel efforts using *Herschel* (e.g. [355–357]). Here we briefly review ongoing efforts with ALMA to obtain the gas content and scaling relations for large samples of star-forming galaxies selected from deep optical/near-infrared fields.

---

[6]Despite reservations on the usefulness of the main sequence as a means to understand the physics of galaxies (see §2.2.7 for example), this parameter space has been used as a tool to understand the statistical properties of galaxy populations.

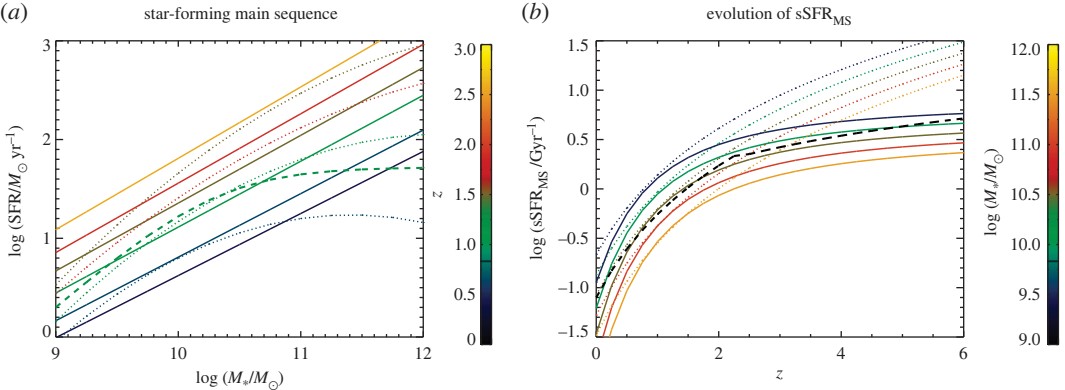

**Figure 12.** The star formation rates of 'main-sequence' galaxies as a function of stellar mass and redshift. These relations are the starting point for the scaling relations shown in figure 13. (*a*) The main sequence as a function of redshift. The solid lines show the main sequence fit by Speagle *et al.* [223], while the dotted lines show the main sequence derived by Whitaker *et al.* [225], which shows a flattening towards high stellar masses. The dashed green line shows the main sequence fit obtained by Lee *et al.* [343]. In their scaling relations work, Scoville *et al.* [344] use a combination of the Speagle *et al.* [223] and Lee *et al.* [343] main sequence fits, while Tacconi *et al.* [345] use Speagle *et al.* [223]. (*b*) The evolution of the typical specific star formation rate of a main-sequence galaxy with redshift. The solid lines show the evolution from Speagle *et al.* [223], used in the scaling relation work discussed in this section. The dotted lines show the evolution from Whitaker *et al.* [225], to highlight that studies of the main sequence have not yet converged on its normalization at high redshifts and low stellar masses (see also, appendix A of [346]). Both studies rely heavily on extrapolations at high redshift. For comparison, the dashed black line shows the evolution measured by Tasca *et al.* [347] using approximately 4500 galaxies with spectroscopic redshifts from the VIMOS Ultra-Deep Survey (VUDS), which contains spectroscopically confirmed sources out to $z \simeq 5.5$.

An alternative and complementary approach is to measure the evolution of the cosmic molecular gas content with redshift using blind surveys. An advantage of this approach is that it does not rely on pre-selecting galaxies at shorter wavelengths, and thus it may give a more unbiased view of the gas content in galaxies. We will discuss efforts carried out with ALMA towards this goal in §4. Nevertheless, targeted studies have the advantage of not needing to survey large areas of the sky, and are valuable to understand the emerging scaling relations in samples of stellar mass-selected, star-forming galaxies, provided that selection effects are properly accounted for (e.g. [345]).

### 3.1.1. Methods for measuring the molecular gas content of high-*z* galaxies

The total mass of molecular gas in galaxies is challenging to determine observationally, since the $H_2$ molecule does not have a permanent dipole moment, and quadrupole transitions require high excitation temperatures (e.g. [358]). Since most of the molecular gas is in a cold phase, this makes it very difficult to observe the bulk of $H_2$ in galaxies directly, and thus indirect tracers must be used, such as the continuum far-infrared/sub-mm emission by cold dust, submillimetre CO rotational lines or some submillimetre fine structure lines, all of which can be ideally observed by ALMA. Here we summarize the main methods used in the literature to obtain molecular gas masses of high-redshift galaxies, and briefly list their main advantages and limitations.

(i) *CO(1–0) line.* This method relies on the fact that carbon monoxide (CO) is the second most abundant molecule in cold molecular gas after $H_2$. The rotational transition CO(1–0) from the first excited state ($J = 1$ to $J = 0$, at a frequency of 115.27 GHz) is easily detectable in the submillimetre and radio (particularly at $z < 0.5$ with current ALMA capabilities; e.g. [33,359]).
   — Advantages: (Almost) direct tracer of cold molecular gas.
   — Limitations: Very faint line, so it requires long integration times. Need to assume a conversion factor (denoted $\alpha_{CO}$ or $X_{CO}$) to convert from CO luminosity to $H_2$ mass, which is uncertain and may depend on galaxy properties such as metallicity (e.g. [360]); see Bolatto *et al.* [361] for a review. Possible existence of CO-dark molecular gas at low metallicities (e.g. [362,363]). CO is easily destroyed in environments with strong cosmic ray energy densities, such as starbursts (e.g. [364,365]). Detectability may be severely affected by the cosmic microwave background (CMB) at high ($z > 2$) redshift [366].

(ii) *J > 1 CO lines.* Transitions from higher-excitation rotational states of CO ($J \to J-1$, with $J \geq 2$, at frequencies $\simeq 115.27 \times J$ GHz) are prime targets with ALMA, as they are brighter, and they can be observed at any redshift beyond $z = 1$ with currently offered frequency bands (e.g. [216,359]). CO(2–1), CO(3–2) and CO(4–3) were some of the first lines targeted for studies of the molecular gas reservoir of galaxies near the peak of cosmic star formation (at $1.5 \lesssim z \lesssim 3$), using the PdBI 1, 2 and 3 mm receiver bands (e.g. [35,39,40,351,352,354,367]). ALMA has the capability to extend these pioneering studies both in the redshift and luminosity/mass ranges probed, and in the number of objects targeted.
— Advantages: Easily observable with ALMA out to high redshifts (e.g. [359]). Can cover a wide wavelength range (e.g. [33]).
— Limitations: Need to correct for CO excitation in order to infer CO(1–0) from $J > 1$ lines (e.g. [41,368]). The limitations of using CO(1–0) described above also apply.

(iii) *Fits to the dust spectral energy distributions (SEDs) in the far-infrared.* This technique is based on deriving dust masses from fits to multi-band observations in the far-infrared/submillimetre, sampling the peak of the dust emission (e.g. [369]). The cold gas masses are then derived by assuming a gas-to-dust ratio, which may be fixed, or dependent on the gas-phase metallicity if available (e.g. [95,356,357,360]).
— Advantages: The far-infrared dust peak is bright and easily detectable at least out to $z \simeq 2.5$ with *Herschel* (e.g. [370,371]). Can use large statistical samples with multi-band measurements from available *Herschel* surveys (e.g. [357,372]).
— Limitations: Need to assume gas-to-dust ratio (which depends on metallicity; e.g. [357]). Gas-to-dust ratio dependence on metallicity may vary with redshift (e.g. [373]). Possibly biased towards warmer dust which does not include bulk of the cold gas mass (e.g. [374]). Need well-sampled infrared SEDs to obtain good constraints on dust temperature and/or dust emissivity index (e.g. [375]). The absolute opacity of dust grains (or emissivity per unit dust mass) needs to be assumed or calibrated; this quantity is model-dependent and can be uncertain by at least a factor of a few (e.g. [376–379]).

(iv) *Single-band sub-mm/mm continuum.* Empirical calibrations between single-band sub-mm/mm continuum and gas masses have been proposed by Scoville *et al.* [380] and Groves *et al.* [381]. They rely on tight empirical relations between the submillimetre flux of galaxies (in the Rayleigh–Jeans (RJ) tail of the dust emission) and gas masses measured using CO (or CO+HI in the case of [381]). The physical basis for these correlations is described in detail in Scoville *et al.* [374]. In short, the argument is that the submillimetre emission is optically thin and optimally traces the colder dust in galaxies, which traces the cold molecular gas reservoirs; the RJ continuum emission per gas mass should be fairly constant as it does not depend strongly on the dust heating in the galaxy, but rather on the total amount of dust.
— Advantages: Very efficient observationally with ALMA, as the continuum emission in a single band can be obtained in much shorter integration times (minutes) than lines or multiple bands (hours). Enables the study of large samples, over a wide redshift range (e.g. [344]), with no need for *a priori* precise redshift measurements. Less sensitive to dust SED fitting uncertainties and degeneracies than method iii.
— Limitations: Calibrated using CO line observations (see limitations of methods i and ii). Assumes single gas-to-dust ratio (solar metallicity); empirical relations break at sub-solar metallicity [381,382]. Assumes single temperature for cold dust, with no redshift evolution (possibly contradicted by Magnelli *et al.* [383] and Schreiber *et al.* [384]). Relies on extrapolations from lower rest-frame observations to the (sub-)millimetre range (usually 850 μm where the relations are calibrated), which can introduce systematic errors (see discussion in [346]). Continuum emission and CO emission may not be co-located/have the same physical extent, therefore they may not trace each other accurately (e.g. [385], see §3.2.3).

Given the obvious advantage of using continuum observations with ALMA instead of more time-consuming spectroscopic observations, this method is becoming increasingly popular to study the molecular gas content of intermediate- and high-redshift star-forming galaxies (e.g. [386–388]). However, up until very recently this method had not been directly tested on the same galaxies it has been mostly used for. The recent study of Kaasinen *et al.* [389] aimed to remedy this situation by directly comparing the gas masses measured from CO(1–0) observations of a dozen $z \sim 2$ galaxies with the Very Large Array with those inferred from the dust continuum observed with ALMA. They find that the two gas mass measurements agree within a factor of 2, and that factor of 2 uncertainty is probably due to uncertainties in dust models that are needed to extrapolate the observed ALMA dust emission to a rest-frame continuum measurement at 850 μm. A factor of 2 uncertainty compares well with uncertainties in the conversion factor from CO(1-0) to a

molecular gas mass (e.g. [361]). They conclude that the single-band method is therefore reliable to obtain the gas masses of massive, star-forming galaxies at $z \sim 2$. While these are promising results, more extensive tests on larger samples spanning wider metallicity and star formation ranges are essential (as discussed also in Liu *et al.* [346]; see also the test on low-redshift galaxies by Hughes *et al.* [390]). Of course, it is worth noting that while observationally cheaper, this dust continuum method does not provide the dynamical information that observations of CO lines do.

(v) *[CI] fine structure lines*. The fine structure lines of atomic carbon [CI] at 492 and 809 GHz were first suggested as reliable tracers of molecular gas in galaxies by Papadopoulos & Greve [299], who challenged the long-held view that [CI] is only distributed in a narrow region at the interface between [CII] and CO in far-UV illuminated molecular clouds, a view that was also starting to be challenged observationally by imaging of [CI] in molecular clouds. Papadopoulos & Greve [299] suggested that under typical ISM conditions, [CI] is ubiquitous in molecular clouds thanks to dynamic processes such as turbulent mixing, non-equilibrium chemical states and cosmic rays (see also theoretical work by, e.g. [365,391–394]). Papadopoulos & Greve [299] argue that in sites of intense star formation and low-metallicity, the production of [CII] starts diminishing the capability of [CI] to trace molecular gas (and indeed it has been suggested that at some point [CII] might become an even better tracer of the molecular gas reservoir; e.g. Madden *et al.* [363]), but nevertheless even in those cases, [CI] should still perform better than CO.

— Advantages: Observed frequencies for high-redshift ($z > 1$) galaxies ideally matched with atmospheric windows (and ALMA passbands), and thus easier to observe than low-$J$ CO transitions. The [CI] lines are optically thin in most environments. If used in conjunction with other lines such as CO lines, can be used to derive the physical properties of the gas (e.g. temperature, density) using large-velocity gradient (LVG) or photo-dissociation region (PDR) models (as done in e.g. [296,315,395–399]).

— Limitations: Theoretically, the [CI]-to-$H_2$ conversion depends on complicated physical processes and is very sensitive to modelling aspects, such as physics of cosmic rays and cloud evolutionary states (e.g. [365,391], though arguably the same can be said of our theoretical understanding of the CO conversion factor). Observationally, we still do not have a systematic calibration of [CI] as a molecular gas tracer that can be applicable to all types of galaxies, including main sequence galaxies, at various redshifts (though see some efforts by e.g. [298,300,400,401]).

We also note that [CII] has been suggested as another potential molecular gas tracer for high-redshift galaxies, especially at low metallicities (e.g. [363]). Indeed, recent theoretical ISM models find that $\gtrsim 70\%$ of [CII] emission in galaxies can come from molecular regions [402,403], and an empirical study using [NII] to differentiate the ionized from neutral regions finds that up to 80% of [CII] comes from neutral gas in local star-forming galaxies, though note the difference between neutral and molecular gas [404]; see also [405,406], for supporting results in ULIRGs. Using ALMA observations of 10 $z \sim 2$ main sequence galaxies, Zanella *et al.* [407] find that the [CII] luminosity correlates well with the molecular gas. However, this is still a controversial method because [CII] emission has been traditionally seen as a tracer of the star formation rate in galaxies (e.g. [408,409]), so whether such a correlation could simply be the result of uniform star formation efficiency is unclear. It is also important to bear in mind that studies of the [CII] deficit in star-forming galaxies show that this line depends strongly on the radiation field and metallicity in galaxies (e.g. [410,411]). It is fair to say that more work would need to be done in this area, to avoid the risk of using the conveniently bright [CII] line to measure both the star formation and the molecular gas reservoir of high-redshift galaxies.

### 3.1.2. Scaling relations between stellar mass, SFR, gas content and redshift

While the first studies of molecular gas in star-forming galaxies at the peak of cosmic star formation with IRAM/PdBI targeted CO in a few of the brightest sources (e.g. [353,412]), it has become clear in recent years that, in order to disentangle the effects of different physical parameters driving galaxy evolution and properly account for selection effects, large statistical studies, similar to those routinely carried out using deep observations in the optical and near-infrared, are needed. To understand the factors that regulate the gas reservoirs and star formation rates of star-forming galaxies as a function of redshift, recent studies are focusing on increasingly larger samples of (mostly) mass-selected galaxies that are chosen to be as representative as possible of the general star-forming population at all redshifts up to $z \simeq 3$ (so far). These are enabled by improvements in sensitivity with the PdBI and

ALMA, as well as refinements to the techniques used to derive molecular gas masses described in §3.1.1. Here we will focus mainly on the most recent results obtained since ALMA has been in operation (which includes also additional data from the PdBI).

An interesting approach is to derive scaling relations that relate the main parameters thought to affect the evolution of star-forming galaxies: cosmic time (redshift), star formation rate, stellar mass, distance from the main sequence, gas fraction and gas depletion time (e.g. [344–346,413]). These parameters enable phenomenological descriptions of gas flows and consumption in galaxies (e.g. [349,414]), as well as quantitative measurements that can be confronted with predictions from theoretical models (e.g. [58,59,415]). The goal is to understand how the gas reservoir affects the star formation and stellar mass growth as a function of redshift and, specifically, how the star formation is regulated by gas fraction and star formation efficiency. These scaling relations are used to address some of the following questions:

— What drives the evolution of the cosmic star formation rate and gas reservoirs with stellar mass and redshift? Is there a varying star formation mode, i.e. different star formation efficiencies and star formation laws? What star formation mode dominates the cosmic star formation history?
— What drives the systematic increase of specific star formation rate (for a given stellar mass) with redshift? I.e. why does the normalization of the main sequence increase?
— At each redshift, why are main sequence outliers (sometimes called 'starbursts') forming stars at much higher rates than main sequence galaxies of the same stellar mass? Is it because they have larger gas reservoirs, or are they more efficient at forming stars? What is the role of mergers?

The starting point of establishing these scaling relations is to trace the evolution of star formation rate, stellar mass and specific star formation rate (i.e. the star formation main sequence). These quantities are relatively well-measured by deep optical/near-infrared surveys (see [2], for a review). Molecular gas surveys with ALMA and the PdBI (e.g. [344,345,388]) aim to understand the peak of star formation rate at $1.5 \lesssim z \lesssim 3$ in terms of the gas reservoir and star formation evolution of galaxies that contribute the most to this peak (i.e. 'normal' galaxies). In figure 12, we plot the evolution of the typical star formation rate of main-sequence galaxies as a function of redshift and stellar mass. We highlight that while various surveys find that the evolution of the specific star formation rate of mass-selected galaxies on the main sequence (MS) at $z < 3$ seems to be well described by a power-law, $\mathrm{sSFR_{MS}} \sim (1+z)^3$ (e.g. [223,340]), the slope of the evolution at $z > 3$ is still debated. Similarly, several studies seem to point to a flattening of the MS at high ($\gtrsim 5 \times 10^{10}\ M_\odot$) stellar masses at all redshifts (e.g. [224,225,343]), although the exact turnover masses and slopes are still debated and may be strongly affected by selection effects (e.g. [416]). Recently, Katsianis *et al.* [417] showed that different methods used to estimate star formation rates in different observational studies contribute to obtaining 'main sequence' relations that do not agree with each other or with theoretical predictions. This has to be kept in mind when performing quantitative comparisons and inferences from such relations.

We use the following definitions routinely used in scaling relation studies:

— The *offset from the main sequence* is defined as:

$$\Delta \mathrm{sSFR_{MS}} = \frac{\mathrm{sSFR}}{\mathrm{sSFR_{MS}}(z, M_*)}, \tag{3.1}$$

where $\mathrm{sSFR_{MS}}(z, M_*)$ is the average specific star formation rate of a main sequence galaxy of stellar mass $M_*$ at redshift $z$.
— The *depletion time*, $t_{\mathrm{dep}}$, and the *star formation efficiency*, SFE, are defined as:

$$t_{\mathrm{dep}} = \frac{1}{\mathrm{SFE}} = \frac{M_{\mathrm{gas}}}{\mathrm{SFR}}, \tag{3.2}$$

where $M_{\mathrm{gas}}$ is the molecular gas mass (usually measured using one or more of the methods detailed in §3.1.1). This is dominated by molecular hydrogen, but it is common to correct for the helium contribution to this mass by multiplying the derived $H_2$ mass by a factor of 1.36. We note that this quantity is at times denoted differently in the literature, e.g. as $M_{\mathrm{ISM}}$ [344,374,380], or $M_{\mathrm{mol}}$ [349]. The molecular gas is considered the same as the total gas mass in the following, since the contribution by atomic hydrogen to the total baryonic mass is found to be negligible at high redshifts (e.g. [345,418]).

— The *molecular gas fraction*, $f_{gas}$, is defined as:

$$f_{gas} = \frac{M_{gas}}{M_{gas} + M_*}. \tag{3.3}$$

In the following, we focus on the largest studies of scaling relations at the time of writing, carried out by Scoville *et al.* [344], Tacconi *et al.* [345] and Liu *et al.* [346], which parametrize the evolution of the gas fraction and depletion time in galaxies as a function of cosmic age (or redshift), stellar mass and specific SFR. Scoville *et al.* [344] estimated the total ISM masses (i.e. molecular gas masses with a correction for He) using ALMA observations of the long wavelength dust continuum in a sample of 708 galaxies from $z = 0.3$ to $z = 4.5$ in the COSMOS field. Tacconi *et al.* [345] compiled a larger sample of 1444 star-forming galaxies between $z = 0$ and $z = 4$ for which molecular gas estimates were derived using three methods: direct CO measurements from IRAM (PHIBSS survey) and ALMA; dust SED modelling and 1 mm continuum (including the sample by Scoville *et al.* [344]). They analysed the systematics between these methods and find that after calibration and benchmarking they converge to consistent scaling relations. Recently, Liu *et al.* [346] performed the largest ever study of this kind in terms of both sample size and dynamical range, by combining a dataset of approximately 700 galaxies at $0.3 < z < 6$ from the A³COSMOS survey, a systematic mining of the ALMA archive in the COSMOS field [419], with an additional sample of approximately 1000 CO-observed galaxies at $0 < z < 4$. This large sample allows them to compare and calibrate different gas mass estimate conversions, as well as to explore the parameter space of star formation properties, gas content and redshift in more detail. They also propose a new functional form for the scaling relations which accounts for different evolutions of galaxies of different stellar mass, which implies down-sizing (faster evolution of more massive galaxies) and mass-quenching effects (gas consumption slows down with cosmic time for massive galaxies but speeds up for low-mass galaxies).

What becomes apparent from these studies is that the larger the samples, the more complex the scaling relations become, with more high-order dependencies between physical properties, making direct comparisons quite challenging. This also highlights the complex physical processes at play, and that while scaling relations can be useful tools in quantifying the overall evolution of the properties of galaxies, as well as how they depend with one another, the physics of galaxy formation is a complex and multi-variate problem in itself. Here we try to briefly make sense of the main results in the recent literature highlighted above.

*Evolution of the gas content.* A common conclusion from all the scaling relation studies is that the molecular gas mass at fixed stellar mass (and hence the gas fraction) of main sequence galaxies increases with redshift, and therefore at higher redshift, a galaxy of a given stellar mass simply has more fuel available to form new stars. Figure 13a,b compares the redshift evolution of $M_{gas}$ and $f_{gas}$ derived by Scoville *et al.* [344], Tacconi *et al.* [345] and Liu *et al.* [346]: while the qualitative behaviour is similar, a few quantitative differences are noticeable. At fixed redshift, the total gas mass depends more strongly on stellar mass in the Tacconi *et al.* [345] scaling relation ($M_{gas} \sim M_*^{0.65}$) than in the Scoville *et al.* [344] and Liu *et al.* [346] relations ($M_{gas} \sim M_*^{0.3}$). At fixed stellar mass, Scoville *et al.* [344] find that the gas mass evolves as $(1 + z)^{1.84}$, while Liu *et al.* [346] find a somewhat slower evolution with redshift, and Tacconi *et al.* [345] find that an additional downturn at higher redshifts fits their data better (figure 13a). The difference is probably attributable to different samples used. Despite these differences, a clear trend seems to arise: the increasing gas fractions with increasing redshifts (at fixed stellar masses) go a long way in explaining the rise of the typical star formation rates of main sequence galaxies. These higher gas fractions are attributed to more efficient accretion of gas from the cosmic web at high redshift (as described in e.g. [422]; see also the recent review by Tacconi *et al.* [423]). Galaxies that are above the main sequence seem to have slightly higher gas fractions that are not sufficient to explain their enhanced SFRs, implying that higher star formation efficiencies are needed to explain these objects (though we note that Liu *et al.* [346] find a stronger correlation where the higher above the main sequence a galaxy is, the larger its gas fraction). While all scaling relations agree that the gas fraction increases towards higher redshifts at all stellar masses, the three studies disagree somewhat on how fast the gas fractions increase for different stellar masses (figure 13b). The better agreement is found at stellar masses around $10^{11} M_\odot$ and $z < 3$, where there are more observations; however, at low masses and high-redshifts, there are significant differences that can only be addressed by obtaining more measurements for galaxies in those regions of the parameter space.

*Evolution of depletion time.* In figure 13c, we show the redshift evolution of depletion time from Scoville *et al.* [344] (solid lines), Tacconi *et al.* [345] (dotted lines) and Liu *et al.* [346] (dot-dashed lines); the lines

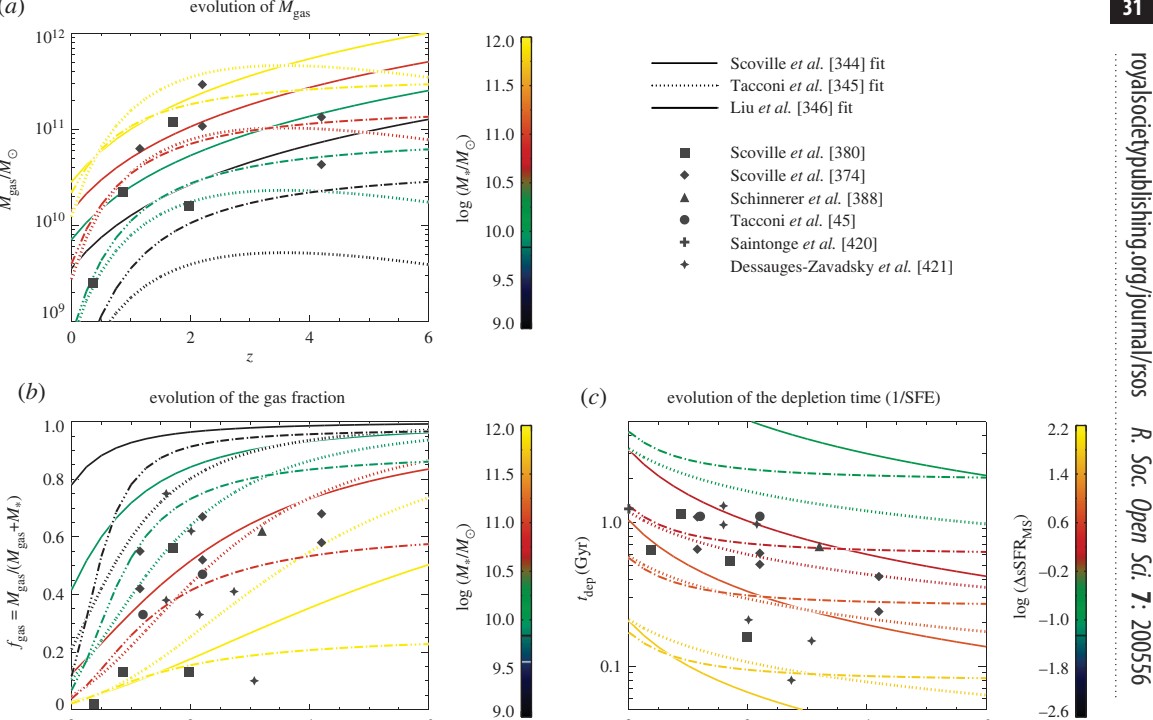

**Figure 13.** Scaling relations between star formation rate, stellar mass, gas content and redshift derived by Scoville *et al.* [344] (solid lines), Tacconi *et al.* [345] (dotted lines) and Liu *et al.* [346] (dotted-dashed lines). Here we focus in particular on the redshift evolution of the gas content (via the gas mass in (*a*) and the gas fraction in (*b*)) and depletion time (*c*), and colour-code the lines according to the main secondary property they depend on (stellar mass or distance from the main sequence). The small symbols show the measurements compiled and 'benchmarked' by Tacconi *et al.* [345]: circles show stacked measurements (mostly using dust continuum) and crosses show individual measurements. Larger symbols show additional notable individual results from the literature: Scoville *et al.* [380]: stack measurements based on ALMA continuum of 107 stellar-mass selected COSMOS galaxies at $0.2 < z < 2.5$ with $M_* \sim 10^{11} M_\odot$; Scoville *et al.* [374]: stack measurements based on ALMA continuum of 145 star-forming galaxies at $\langle z \rangle = 1.1$, 2.2 and 4.4, with $M_* \gtrsim 2 \times 10^{10} M_\odot$, with sources both on and above the main sequence; Schinnerer *et al.* [388]: individual ALMA continuum measurements of 45 main sequence galaxies at $z \sim 3.2$ in the COSMOS field with $M_* \sim 5 \times 10^{10} M_\odot$; Tacconi *et al.* [45]: IRAM PHIBSS CO(3-2) detections of 52 main sequence galaxies at $z \simeq 1.2$ and $z \simeq 2.2$ and $M_* \gtrsim 2.5 \times 10^{10} M_\odot$; Saintonge *et al.* [420]: 222 CO(1-0) measurements of $z \sim 0$ galaxies with $M_* \geq 2.5 \times 10^{10} M_\odot$ from the IRAM COLD GASS survey; Dessauges-Zavadsky *et al.* [421]: IRAM CO measurements of five $z \sim 1.5 - 3$ lensed galaxies with low stellar masses ($M_* < 2.5 \times 10^{10} M_\odot$) and low star formation rates (SFR $< 40 M_\odot$ yr$^{-1}$).

are shown for a fiducial stellar mass of $5 \times 10^{10} M_\odot$, and colour-coded according to offset from the star-forming main sequence. The depletion times depend weakly on stellar mass in the scaling relations of Scoville *et al.* [344] and Tacconi *et al.* [345], meaning perhaps that at each redshift all main sequence galaxies seem to have a similar star formation mode. However, Liu *et al.* [346] predict a stronger evolution, in the sense that in high-mass galaxies the depletion time increases 20-fold from early cosmic times to present, while low-mass galaxies show faster depletion times at later cosmic times. This could be indicative of downsizing, where more massive galaxies evolve at earlier times (see discussion in [346]). All scaling relations predict a slow decrease of the depletion time (or increase of the star formation efficiency) with redshift, though it is important to note that there are some significant offsets between the different derivations at $z = 0$, and the depletion time decreases faster with redshift for Scoville *et al.* [344] [$t_{\rm dep} \sim (1 + z)^{-1.04}$] than for Tacconi *et al.* [345] [$t_{\rm dep} \sim (1 + z)^{-0.62}$] and Liu *et al.* [346] ($t_{\rm dep}$ almost constant with redshift for a fixed stellar mass). Tacconi *et al.* [345] and Liu *et al.* [346] attribute these differences at least in part to the different datasets used by the two studies to anchor the relation at $z = 0$, but Tacconi *et al.* [345] also note that their method obtains steeper slopes when only dust continuum measurements are used (i.e. excluding CO), so some of the difference could come from different measurement methods as well. Earlier studies with limited ALMA samples seemed to show that the increase in specific SFR of the main sequence with redshift

was due solely to the increase in gas fraction of galaxies, and not to a change in star formation efficiency/ depletion time (e.g. [374,380,388]; see also [45,413]). It is important to note that [374,380] relied mostly on a stacking analysis of the continuum emission for relatively small (less than 100) samples. Small statistics are also a problem for Schinnerer *et al.* [388]. The evolution of $t_{\mathrm{dep}}$ with redshift is crucial to our understanding of the small-scale star formation processes in galaxies and how they evolve. If the depletion time of galaxies in the main sequence essentially does not evolve with redshift (as also found previously by Schinnerer *et al.* [388] and Genzel *et al.* [413]), then this would imply that the rapid increase of cosmic SFR density towards $z \simeq 2$ is caused by a larger availability of molecular gas (thanks to, for example, increased accretion through gas flows and mergers), rather than a fundamental change in the small-scale physics of star formation in galaxies. On the contrary, the Scoville *et al.* [344] results support the idea that a change in the star formation efficiency at high redshift is also required. With the current samples, which scenario is more likely is still hard to establish; more direct ALMA (and NOEMA) measurements of the gas content of galaxies in samples spanning a wide range in redshift, star formation rate and stellar mass, using both targeted and blind surveys, will be needed to address these discrepancies. Regardless of the behaviour in the main sequence, Scoville *et al.* [344], Tacconi *et al.* [345] and Liu *et al.* [346] all find that galaxies above the main sequence at a given redshift seem to be forming stars at higher efficiencies than main sequence galaxies at the same redshift ($t_{\mathrm{dep}} \sim \Delta\mathrm{sSFR}_{\mathrm{MS}}^{-0.70}$, $t_{\mathrm{dep}} \sim \Delta\mathrm{sSFR}_{\mathrm{MS}}^{-0.44}$ and $t_{\mathrm{dep}} \sim \Delta\mathrm{sSFR}_{\mathrm{MS}}^{-0.57}$, respectively). The favoured interpretation is that these outliers (starbursts) are forming stars more efficiently, presumably as a result of major gas-rich mergers (e.g. [344]).

## 3.2. Resolved studies

While statistical studies of SMGs (§2) and the global FIR galaxy population (§3.1) with ALMA's most compact configurations have already dramatically affected our understanding of high-redshift galaxy assembly, with its more extended configurations, ALMA has been delving into almost completely uncharted territory. The sub-arcsecond resolution configurations make it possible to resolve individual high-redshift sources, allowing sub-galactic studies of the dust-obscured star formation and ISM in star-forming galaxies on scales down to less than or equal to 1 kpc, even for unlensed sources. Only a handful of the very brightest (e.g. [37,46,48,49]) and/or most strongly lensed star-forming galaxies (e.g. [50,51]) had previously been studied on these scales. This has led to an avalanche of new results on the resolved dust/gas properties of distant ($z > 1$) galaxies. We note that due to surface brightness sensitivity limitations, much of the most detailed/highest-resolution work with ALMA has necessarily still focused on submillimetre-bright sources (i.e. $S_{850\,\mu\mathrm{m}} \gtrsim 1\,\mathrm{mJy}$), regardless of how those sources were initially selected (see §2.4). Here we review some of the main applications and results that this leap in observational capabilities has enabled.

### 3.2.1. Source sizes/profiles in rest-frame FIR continuum emission

One of the first results from ALMA on the resolved properties of $z \simeq 1$ star-forming galaxies has been on the spatial extent of the (rest-frame) FIR continuum emission, which was previously largely unknown. Specifically, high-resolution (less than or equal to 0.2") ALMA observations have revealed compact (approx. 1–5 kpc FWHM) dusty cores in submillimetre continuum imaging of $z \sim 2$ galaxies (e.g. [252,214,338,424–430]), substantiating earlier claims from lower-resolution data (e.g. [46]) and sparsely sampled *uv*-data on the high-redshift tail of SMGs [431]. Interestingly, this observation—which has been proffered as evidence for bulge growth and morphological transformation (§3.2.2)—appears not to depend strongly on either merger state [424] or relation to the 'main sequence', with similarly compact 'cores' reported in everything from 'main sequence galaxies' [338][7] to the brightest SMGs [252,214]. Only a handful of the most extreme early-stage mergers have been observed to show clear evidence for distinct merging components in the FIR, and even then, the individual merging galaxies show evidence for compact FIR emission (e.g. [202]). Nevertheless, these observations are consistent with previous suggestions from other tracers (e.g. radio synchrotron emission, the [CII]/FIR ratio; [154,290,329]) that the FIR regions in luminous high-redshift sources are more extended than the even more compact FIR regions frequently observed in local ULIRGs (e.g. [432]).

---

[7]We note, however, that according to our proposed definition in §2.4, this source is an SMG, because of its bright submillimetre flux, regardless of how it was originally selected.

While the overall trend is for compact FIR emission, and while the sizes of the brightest FIR sources appear to be roughly consistent with expectations from the (optically thick) Stefan–Boltzmann law relating size, luminosity and dust temperature (e.g. [177,214,253]), a few studies probing galaxies further down the luminosity function report slightly more extended emission (e.g. [428,433]). This is in contrast to the size-luminosity relation measured for all Band 6/7 resolved sources from the ALMA Archive [424], where the authors found evidence for larger FIR sizes at high luminosities [$R_e$(FIR) $\propto L_{\mathrm{FIR}}^{\alpha}$, with $\alpha = 0.28 \pm 0.07$], in agreement with UV measurements of star-forming galaxies (e.g. [434]). The normalization of this relation is also found to evolve with redshift, suggesting that (like in the UV), sources are smaller at higher-redshift. We note that the archival study assumes a single dust temperature for all galaxies and, at any rate, shows a very large scatter among individual measurements, but it demonstrates the ever-growing size and potential of the ALMA Archive, enabling statistical studies of resolved properties of high-redshift galaxies.

Taking the extent of the FIR emission as a proxy for the extent of the dusty star formation, one of the immediate implications of the measured FIR sizes is for the global SFR surface densities ($\Sigma_{\mathrm{SFR}}$) of high-redshift sources. (We note that the assumption is generally made that the FIR emission is heated primarily by star formation, with negligible AGN contribution; this could be wrong especially for the most massive sources.) This is of particular interest for the brightest FIR sources, where the high SFRs could potentially lead to values of $\Sigma_{\mathrm{SFR}}$ exceeding the Eddington limit for a radiation pressure supported starburst (approx. 1000 $M_\odot$ yr$^{-1}$ kpc$^{-2}$; [435], though note that the precise value depends on the physical conditions of the source, including optical depth). While the ALMA results and earlier efforts suggest that some of the brightest and most extreme sources (including quasar hosts) may approach this limit (e.g. [99,252,175,202,426,436,437]), the statistically significant samples of deblended and resolved sources provided by ALMA suggest that such cases are indeed rare, with median values that are typically sub-Eddington even for the FIR-brightest sources (e.g. 100 $M_\odot$ yr$^{-1}$ kpc$^{-2}$; [252], figure 14). Making the simplistic assumption that variations in the single-band submillimetre flux density correlate with variations in the local star formation rate, resolved (sub-galactic) observations suggest that the star formation remains sub-Eddington on approximately 500 pc scales (e.g. [439]), though even higher-resolution (approx. 150 pc) observations find evidence for more extreme (greater than 1000 $M_\odot$ yr$^{-1}$ kpc$^{-2}$) SFR surface densities ([427], though they caution that an AGN contribution cannot be ruled out). At the same time, ALMA has allowed global values of $\Sigma_{\mathrm{SFR}}$ to be measured for sources much further down the luminosity function, reaching values as low as $<1\ M_\odot$ yr$^{-1}$ kpc$^{-2}$ (e.g. [428]).

In addition to constraining FIR sizes for high-redshift sources, in cases with enough S/N per beam (or with stacking), ALMA has allowed the *profile* of the FIR emission to be fit. As with the measurements of source sizes, the current ALMA results suggest uniformity in the profiles, with Sérsic fits returning Sérsic indices near unity (e.g. [214,253,338,429,440,441]). These results suggest that the FIR profiles of high-redshift sources are consistent with exponential discs (Sérsic index $n = 1$) over a large range in source properties. Such low Sérsic indices, even for the most FIR-bright sources, suggest that even the most massive sources observed with ALMA are still in the process of building their bulges (§3.2.2).

Resolving the FIR emission also allows the possibility of constraining the (global) optical depth of the sources. This is possible as resolved observations provide a measurement of the brightness temperature ($T_B$), which is the equivalent temperature that a blackbody would have in order to be as bright. In this way, Simpson *et al.* [177] constrain the typical optical depth within the half-light radius for their SMGs of $\tau = 1$ at $\lambda_0 \geq 75\ \mu$m. Compared with local ULIRGs (e.g. [52]), this limit suggests that high-redshift SMGs remain optically thick to *longer* wavelengths than similarly luminous local sources. Such analyses—now made possible by ALMA—also serve as a reminder to treat the stellar masses of such dusty galaxies (§2.2.7) with considerable caution.

### 3.2.2. Comparison with rest-frame optical emission/stellar mass

For the ALMA continuum sources initially selected as single-dish submillimetre sources, the angular resolution of ALMA has allowed studies not only of the detailed submillimetre morphologies, but also the first detailed rest-frame optical/UV morphologies via reliable counterpart identification. Many of the FIR-bright sources show irregular rest-frame optical/UV morphologies (e.g. [442,443]), with little correlation between the detailed (approx. kpc-scale) ALMA and *HST* morphologies (e.g. [214]). Some ALMA-identified continuum sources are not detected at all in deep *HST* imaging (e.g. approx. 20% of the SMGs in $H_{160}$-band imaging with a median sensitivity of 27.8 mag [442]), including those in 'blind' ALMA surveys [64]. These '*HST*-dark' sources are not a new phenomenon,

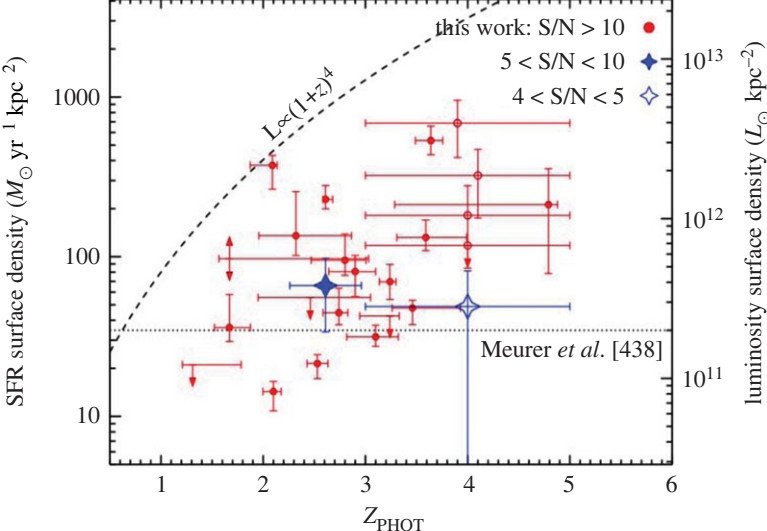

**Figure 14.** Star formation rate density versus redshift for the FIR-bright SMG sample from Simpson *et al.* [252]. The dashed line indicates luminosity evolution $L_{IR} \propto (1+z)^4$, and the dotted line shows the 90th percentile of the luminosity surface density for a sample of UV-selected sources [438]. Taking the extent of the FIR emission as a proxy for the extent of the dusty star formation, the newly measured FIR sizes for large samples of high-redshift sources have allowed the global star formation rate surface densities to be constrained. The resultant values are largely below the Eddington limit for a radiation pressure supported starburst (approx. 1000 $M_\odot$ yr$^{-1}$ kpc$^{-2}$; [435]), though note that the precise value depends on the physical conditions of the source, including optical depth). Figure from Simpson *et al.* [252].

having been known to exist for some time based on pre-ALMA-era interferometry (e.g. [98,99,150], see also §2.4), although some of the newly discovered examples can be up to an order of magnitude fainter in the (sub-)millimetre (e.g. [444]). Other ALMA-identified continuum sources can show significant offsets between the ALMA centroid and the bulk of the rest-frame optical/UV emission, even after astrometric corrections have been applied. This is true not only in FIR-bright continuum sources (e.g. [141,440,442]), but also between FIR lines and optical/UV emission in lower-luminosity $z > 5$ galaxies (e.g. [63,291,445,446]), where Carniani *et al.* [291] argue that the latter does not correlate with SFR. Such offsets could indicate either complex morphologies (e.g. distinct physical components such as major/minor mergers or accretion events) or differential dust obscuration. In either scenario, this observation may have implications for commonly used SED fitting routines that implicitly assume the dust is co-located with the optical/near-IR continuum emission in order to perform energy balance (e.g. [13,447–449]).

For the FIR-bright sources (both in 'classical' SMG samples and otherwise), multiple studies report that the newly resolved FIR continuum emission is more compact on average than the rest-frame optical/UV imaging (e.g. [252,214,338,385,424,425,429,450], figure 15). For these sources, this size discrepancy between the existing stellar populations and the active, dusty star-forming regions has been interpreted as evidence for ongoing bulge formation. Other studies have reported that this difference may not exist for FIR-fainter galaxies, which may therefore be in a state that precedes bulge formation [428]. Note that some of these studies focus on the existing rest-frame optical imaging directly, while others attempt to derive the underlying stellar mass distributions, which are typically found to be more compact than the optical imaging alone (e.g. [338,425,450]). In this way, Barro *et al.* [338] and Lang *et al.* [450] found that the stellar mass profiles of their galaxies were more extended than the ALMA-traced FIR emission, still consistent with the interpretation of bulge growth, while Nelson *et al.* [425] found that the underlying stellar mass distribution was actually more compact than the FIR emission in their target, which was classed as a $z = 1.25$ 'Andromeda progenitor'. These studies typically rely on an empirical correlation between the stellar mass-to-light ratio and a two-band optical colour, and are thus limited by the optical imaging and high central column densities of dust, emphasizing the importance of future near-IR imaging campaigns with *JWST*.

In the sources where evidence for bulge growth has been reported, the intense central star formation implied by the relatively compact FIR emission has been further used to argue for rapid morphological transformation (timescales of less than approx. a few hundred Myr), which can help place the ALMA-

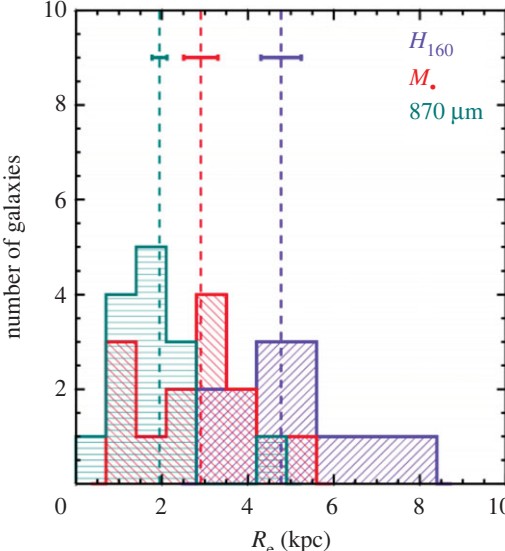

**Figure 15.** Histograms of the effective radii for the ALMA 870 μm continuum emission, stellar mass profiles, and $H_{band}$-light for the SMGs studied by Lang *et al.* [450]. The stellar mass distributions were inferred from spatial mass-to-light ratio corrections based on rest-frame optical colours. The compact FIR continuum sizes measured for FIR-bright sources compared with their implied stellar mass distributions suggests that these galaxies are experiencing intense periods of morphological transformation and bulge growth. Figure from Lang *et al.* [450].

detected galaxy populations in the broader cosmological context. In particular, while SMGs have previously been linked to local elliptical galaxies via $z \sim 2$ compact quiescent galaxies (e.g. [107,108,228–230]), including based on their interferometrically confirmed global physical properties (§2.2.7), the constraints that now exist on their FIR sizes and light profiles have helped further investigations of this connection. Indeed, Chen *et al.* [442] argue that the difference in average physical extent and Sérsic index between SMGs and $z \sim 2$ quiescent galaxies requires significant structural evolution before the star formation is quenched, which Simpson *et al.* [252] show is possible for their SMG sample (on average) based on the current bursts of star formation. Hodge *et al.* [214] further argue that the expected sizes, stellar masses and gas surface densities of the $z \sim 0$ SMG descendants are consistent with the most compact, massive early-type galaxies observed locally. Miettinen *et al.* [179], meanwhile, find that while the evolution of $z > 3$ SMGs into $z = 2$ compact quiescent galaxies is plausible, their $z < 3$ SMGs (which are more massive than some other SMG samples) would not fit into a scenario where they evolve into lower-mass compact quiescent galaxies, highlighting the fact that not all SMG samples are equal. Finally, Barro *et al.* [338] examine the potential connection between $z = 2$ compact quiescent galaxies and massive $z = 2.5$ dusty star-forming galaxies which are specifically selected to be compact in the rest-frame optical (e.g. [451]), arguing that the structural evolution implied by the ALMA-observed nuclear starbursts supports a dissipation-driven formation scenario.

### 3.2.3. Comparison with other tracers

In addition to revealing the sizes and profiles of the FIR continuum in high-redshift galaxies, the advent of ALMA has also led to advances in resolved studies of their molecular and atomic gas. While some such high-resolution studies had been carried out previously using pre-ALMA-era radio and (sub-)millimetre interferometers (e.g. [35,37,48,412,452]), resolved CO studies are particularly time-intensive due to the brightness of the lines and the more limited bandwidths over which they are observable (compared with the dust continuum), and this remains true even with ALMA. We also caution that conclusions drawn from resolved CO studies probably depend on the rotational *J*-transition considered, with higher-*J* lines tracing denser and more highly excited gas that may have a significantly different spatial extent (e.g. [337,368,453,454]).

With this caveat in mind, one of the general findings for FIR-bright sources has been the difference in effective radius between the dust continuum and the cool gas traced by $J \leq 3$ CO (e.g. [385,430,440], figure 16). In particular, these studies find that the cool gas is more extended than the FIR continuum, as was previously suggested by some of pre-ALMA results (e.g. [368,412]). Naively, such a result could

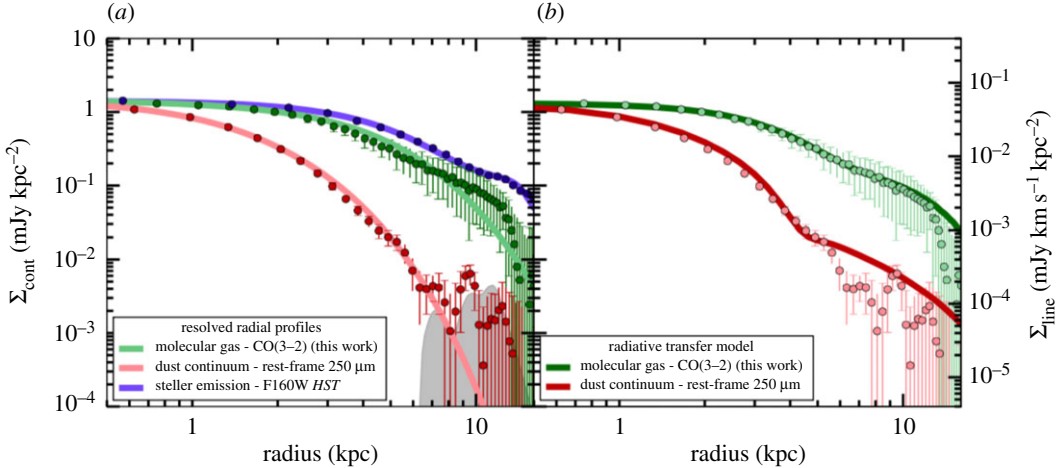

**Figure 16.** Stacked radial profiles for the cool molecular gas (traced by ALMA CO(3–2) emission), dust continuum (traced by rest-frame approx. 250 μm ALMA emission) and stellar emission (traced by $H_{160}$-band *HST* emission) in SMGs [440]. (*a*) One-component exponential fits convolved with the respective beam sizes, and demonstrating that the cool molecular gas and stellar emission are clearly more extended than the rest-frame 250 μm dust continuum. (*b*) Best-fit model from joint radiative transfer modelling to the dust and CO emission, demonstrating that the observed size difference can be explained through radially decreasing temperature and column density distributions. Figure from Calistro Rivera *et al.* [440].

be taken to imply that the dust is more concentrated than the molecular gas, which would then suggest a varying dust-to-gas ratio across the sources, as has been observed in some local spiral galaxies (e.g. [455–457]). However, joint radiative transfer modelling of the dust continuum and CO demonstrates that this effect can also be achieved through radial variations in the dust temperature and optical depth [206]. Following the same radiative transfer calculations (originally presented in [270]), Calistro Rivera *et al.* [440] show that such a model successfully reproduces the apparent size difference observed between CO(3–2) and dust continuum emission in stacked radial profiles of SMGs (figure 16). The importance of dust temperature gradients was also recognized by Cochrane *et al.* [458], whose radiative transfer modelling of galaxies from the FIRE-2 simulations demonstrated that, due to dust heating, the spatial extent of the observed dust continuum emission is sensitive to the scale of recent star formation. On the other hand, the multi-band continuum imaging of the resolved $z = 3$ source SDP.81 shows no evidence for a varying dust temperature across the source [310]. Nevertheless, caution should be exercised when using the FIR continuum to trace the cool gas (§3.1.1) in a resolved sense without taking into account potential variations in dust temperature and gas column density.

The relatively compact size of the FIR continuum emission in FIR-bright sources also appears to hold with respect to the [CII] 158 μm emission (e.g. [310,329,426,459,460]), where the latter is now routinely detected (including serendipitously; e.g. [290,426]) and resolved with ALMA. As an extremely bright FIR line, [CII] has delivered on its promise of being a workhorse line in the era of ALMA, including for lower star formation rate galaxies (e.g. [461]) and at the highest redshifts (§3.4) where low-*J* CO emission is affected by the CMB (e.g. [366]). The physical origin of the [CII] is more difficult to constrain, in general, as it may arise from multiple different phases of the ISM—from photodissociation regions (e.g. [462]) to cold atomic gas (e.g. [363])—as well as being enhanced by shocks (e.g. [463]). This may explain why recent ALMA studies have reported evidence for both extended, low-surface-brightness emission (e.g. [253,329,460]) as well as compact cores less than or equal to 1 kpc in radius [410], suggesting a different surface brightness distribution than either the FIR continuum or low-J CO emission.

Finally, while the tight and almost universal radio-FIR correlation (e.g. [219]) suggests that the FIR continuum and radio synchrotron emission from galaxies are closely linked on global scales, ALMA's superb angular resolution has allowed this correlation to be tested on both unresolved (e.g. [428], see also §2.2.7) as well as resolved scales. The latter report that the FIR continuum sizes measured are smaller, on average, than the radio continuum sizes for FIR-bright sources (e.g. [252,464]). Simpson *et al.* [252] suggest that the discrepant sizes may be due to cosmic ray diffusion, although Miettinen *et al.* [464] argue that the short cooling time of cosmic ray electrons rules out this explanation. Another possibility is that mergers have perturbed the magnetic fields, stretching them out to larger spatial scales (e.g. [465]). This possibility was considered unlikely by Miettinen *et al.* [464] due to the observed agreement between the radio and mid/high-*J* CO sizes, though they cautioned that their

analysis relied on measurements from *different* SMG samples. Alternatively, the discrepancy could again be due to a radially varying dust temperature (or a two-component ISM; [464]), where the spatially extended gas component is traced by the low/mid-$J$ CO and radio continuum emission. Recently, Thomson *et al.* [466] confirmed that the VLA radio sizes of 41 SMGs for the S2CLS survey are about a factor of 2 larger than the cool dust emission traced by ALMA at 870 μm. Thanks to multi-frequency radio data at 610 MHz, 1.4 GHz and 6 GHz, they were able to obtain radio spectral shapes for their sources, which they explain using a combination of weak magnetic field strength and young starburst ages. Their modelling also supports the idea that the mismatch between radio and far-infrared sizes may indicate production of low-energy secondary cosmic ray electrons in the extended gas disc, due to the interaction of cosmic rays produced in the central starburst with baryons in the circumnuclear region. Note that Rujopakarn *et al.* [428] do not find any evidence for a size difference between the ALMA and VLA sizes of their FIR-fainter sources, consistent with the agreement they reported between the FIR continuum and rest-frame optical/UV sizes. This could be consistent with a picture where the star formation is occurring over a larger portion of the disc in such sources, but further work is needed to determine the actual distribution of the star formation itself in the various populations (through resolved, multi-frequency ALMA observations), as well as the relevant galaxy parameters (beyond selection wavelength) on which these trends depend.

### 3.2.4. The star formation law

Taking the observed extents of the FIR continuum and CO emission to trace the star formation and/or molecular gas extents, some studies have attempted resolved (i.e. sub-galactic) analyses of the SFR surface density versus the molecular gas surface density. The relationship between these quantities describes the relative efficiency with which gas is transformed into stars in different environments, and is thus used to study the star formation law (i.e. 'Kennicutt–Schmidt' relation; [312–314]). In the pre-ALMA era, most high-redshift studies had been limited to unresolved studies—in many cases, with the same global size assumed when calculating both the total SFR and gas surface density—with resolved studies limited to a handful of the most extreme SMGs or strongly lensed galaxies (e.g. [44,49,317,318]).

The advent of ALMA has allowed such resolved studies on an increasing variety of sources (see also §2.3.3 for studies of individual star-forming clumps using strong gravitational lensing). For example, Chen *et al.* [385] studied an unlensed $z = 2.2$ SMG in resolved CO(3–2) emission, finding that the central region has a gas consumption timescale that agrees with local ULIRGs and SMGs, while the gas consumption timescales seen in the outskirts are more consistent with local and $z \sim 2$ star-forming galaxies. Meanwhile, Cibinel *et al.* [433] presented resolved CO(5–4) imaging of a $z = 1.5$ 'main sequence' galaxy, arguing that the more centrally concentrated CO(5–4) emission observed (compared with other star formation tracers) could again be evidence for a radially varying star formation efficiency. While such a result may be expected based on resolved studies of local galaxies (e.g. [467]), the high-redshift studies are still plagued by uncertainties in e.g. the CO-to-$H_2$ conversion factor, CO excitation ratio, lack of high-resolution low-$J$ observations and use of single-band submillimetre continuum emission to trace the resolved SFR surface density (in addition to small number statistics; see §3.1.1). Ultimately making progress in this area will require systematic studies of larger samples where these factors can be better constrained, including through dynamical constraints on the CO-to-$H_2$ conversion factor (§3.2.6), observations of lower-$J$ CO lines, and multi-band continuum studies to better constrain the distribution of SFR. Such studies are possible with ALMA (typically using higher-$J$ CO lines) but require more observing time than has typically been allocated thus far. Resolving the lower-$J$ CO lines in larger samples of high-redshift sources will require the ALMA Band 1/2 receivers and the proposed next-generation VLA (ngVLA).

### 3.2.5. The [CII]/FIR deficit

Thanks to the high angular resolution achievable by ALMA in both the [CII] line and FIR continuum emission of high-redshift galaxies, progress has recently been made in studies of the '[CII] deficit', where the (global) $L_{[CII]}/L_{FIR}$ ratio can show a marked decrease for galaxies with a total $L_{FIR} \gtrsim 10^{11}$ $L_{\odot}$ (e.g. [328,330,331]). Using the size measurements obtained with ALMA for their strongly lensed SPT sources, Spilker *et al.* [143] showed that the [CII]/FIR luminosity ratio is a strong function of FIR surface density, extending the result found by Díaz-Santos *et al.* [332] for low-redshift galaxies by another two orders of magnitude (figure 17).

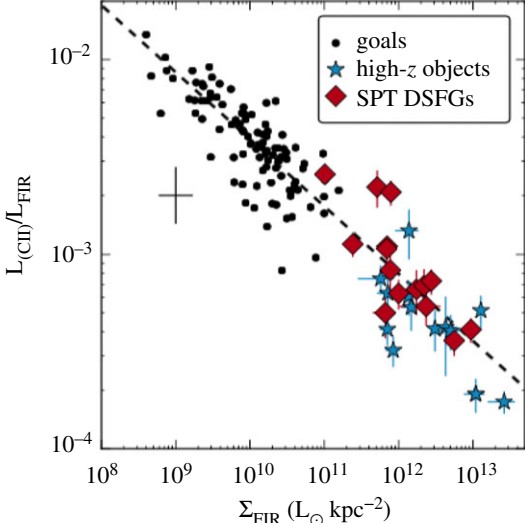

**Figure 17.** The [CII]/FIR luminosity ratio as a function of FIR surface density for local and high-redshift galaxies (the '[CII] deficit' refers to the observation that this ratio is lower for more luminous sources). Spilker *et al.* [143] used the size measurements obtained with ALMA for their strongly lensed SPT sources to extend the local relation found by Díaz-Santos *et al.* [332] (dashed line) by another two orders of magnitude. Subsequent studies have used the superb sensitivity and angular resolution of ALMA to extend the investigation of the deficit to unlensed sources and (sub-)kpc scales. Figure from Spilker *et al.* [143].

Subsequent studies have expanded the investigation to $z > 5$ galaxies (e.g. [468]) as well as to kpc and even sub-kpc scales. For example, Lamarche *et al.* [469] and Litke *et al.* [470] use the persistence of the deficit in sub-galactic measurements of strongly lensed galaxies to conclude that if there is a physical scale where the deficit emerges, it must be sub-kpc. This suggests a local origin for the deficit, as argued previously for nearby galaxies by e.g. Smith *et al.* [411]. A similar conclusion was reached by Gullberg *et al.* [329] and Rybak *et al.* [410], who used the capabilities of ALMA to extend such resolved studies to unlensed SMGs. Rybak *et al.* [310,410] further argue that the slope of the deficit in the $L_{[CII]}/L_{FIR}$-versus-$\Sigma_{SFR}$ plane is consistent with thermal saturation of the [CII] line at high gas temperatures. This explanation was proposed previously by Muñoz & Oh [471], but was not found to hold for the source studied by Litke *et al.* [470]. It would also be inconsistent with the low gas temperatures found in local galaxies [405]. Further work is needed to determine whether this explanation holds for the high-redshift galaxy population in general.

### 3.2.6. Dynamical studies

When a line such as CO (or [CII]) is resolved with sufficient signal-to-noise per beam, this also allows the kinematic properties to be investigated through the fitting of dynamical models. Such studies were again carried out already prior to ALMA (e.g. [35,48,452,472,473]). However, they have been increasing in frequency thanks to the relative speed at which ALMA can resolve these emission lines—even into the epoch of reionization (§3.4)—and there are now too many to list comprehensively here. Due to its brightness, the [CII] line can be imaged particularly quickly, sometimes serendipitously (e.g. [290,426]) or at exquisite (less than or equal to 1 kpc) resolution (figure 18; e.g. [329,410,474]).

Most such studies find signatures of disc-like rotation, and they then attempt to quantify the rotation dominance using various dynamical modelling tools (e.g. DYSMAL, GALPAK3D, $^{3D}$BAROLO; [475–477]). As one of the goals of these studies is often to search for evidence of a merger origin, it is important to note that the presence of significant disc rotation alone is not a sufficient condition to rule out a merger scenario, as gas-rich mergers at high-redshift are thought to quickly reform rotating gas discs after final coalescence [478–480], and late-stage mergers can be mistaken for rotation depending on data quality (e.g. [470]). This has also been demonstrated observationally using an ALMA (+CARMA/SMA/PdBI) CO imaging study of optically selected merger remnants in the local universe, where some of the CO discs were even found to approach the size of the Milky Way disc [481]. Those authors suggest that deep, rest-frame *K*-band imaging at high resolution is necessary to understand the true nature of high-redshift sources, emphasizing the important role to be played by *JWST*.

One application of the CO dynamical modelling increasingly made possible with ALMA is the ability to dynamically constrain the CO-to-H$_2$ conversion factor, $\alpha_{CO}$ [361], which is notoriously uncertain for high-

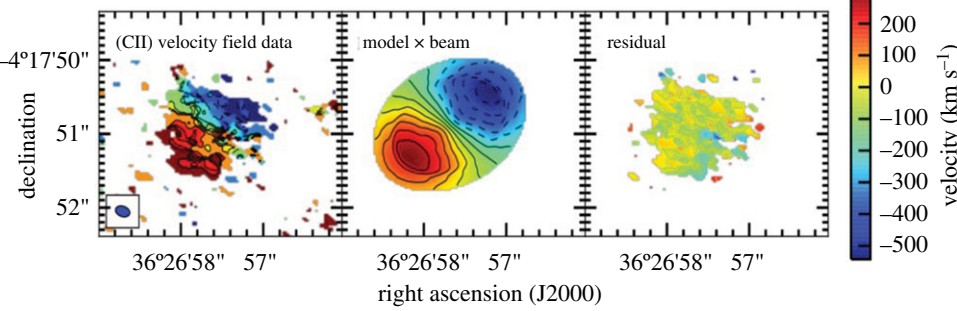

**Figure 18.** The brightness of the [CII] line has allowed it to be imaged and spatially resolved particularly quickly in high-redshift galaxies with ALMA, facilitating dynamical modelling. Here the [CII] line is resolved at approximately 1 kpc resolution in a $z \sim 3$ galaxy and compared with a model of a rotating disc. Figure from Leung *et al.* [474].

redshift galaxies. This usually entails subtracting the stellar mass and likely dark matter fraction from the dynamical mass (or neglecting their potentially significant contributions to derive an upper limit), and then taking the ratio of the remaining mass and CO luminosity, assuming the remaining mass is molecular (e.g. [202,430]). Calistro Rivera *et al.* [440] employed a similar technique, but they used a Bayesian approach to explore the covariance between $\alpha_{CO}$ and the stellar mass-to-light ratio, which is often highly uncertain for dusty, strongly star-forming galaxies. Note that despite ALMA's relative speed compared with other facilities, resolving the low-*J* transitions of CO with sufficient resolution and S/N to carry out such analyses still requires non-negligible time investment, even for CO-bright sources like SMGs. Nevertheless, such studies remain one of the best ways to constrain the molecular gas mass in high-redshift galaxies—as well as the physical conditions that may be driving changes in $\alpha_{CO}$ (e.g. [482–484])—and the application of the Bayesian technique laid out by Calistro Rivera *et al.* [440] to larger samples of galaxies with higher-quality data has the potential to accurately constrain multiple key galaxy parameters simultaneously.

### 3.2.7. Spatially resolved gas excitation and dust mapping

While there have been a handful of multi-frequency studies using ALMA to investigate the CO spectral line energy distribution (SLED) (e.g. [474]) and dust SED (e.g. [460]; da Cunha *et al.* [375]) in high-redshift star-forming galaxies, such studies are still quite limited, even in the global sense. These multi-band investigations will be key for constraining the gas excitation conditions and dust properties of various populations. In particular, dust temperatures are often assumed for high-redshift galaxies based on single-band measurements, despite the fact that an incorrectly assumed dust temperature can change the derived FIR luminosity (and thus implied SFR) by an order of magnitude or more (figure 19). Clearly, multi-band global studies are the first necessary step.

For CO- and FIR-bright sources, ALMA further has the ability to easily resolve multi-band measurements on approximately kpc or even sub-kpc (for the dust continuum) scales. For the dust continuum, such studies will be important for determining how the dust SED changes *within* individual galaxies, which can cause the resolved star formation rate to differ from that implied using the typical method of simply scaling the global dust SED based on a resolved single-band continuum measurement (§3.2.1). For CO, resolved multi-line studies could help shed light on the dominant excitation sources (e.g. SF versus AGN) as well as test physical prescriptions between CO excitation and e.g. $\Sigma_{SFR}$ (e.g. [485]), as Sharon *et al.* [319] attempt on a strongly lensed source using SMA and VLA data. Note that depending on the redshift of the source(s), multi-line CO studies still typically require lower-frequency observations than are possible with the current ALMA bands in order to anchor the CO SLED at low-*J* transitions. This is therefore an area that is ripe for future work, not just with the current ALMA, but also with the future Band 1/2 receivers, the VLA, the proposed ngVLA, and the SKA [486].

### 3.2.8. Detailed morphological studies

In the brightest high-redshift sources, the resolution achieved by ALMA has allowed studies of their resolved (kpc, or even sub-kpc) structure. This has enabled searches for e.g. the approximate kpc-scale 'clumps' first reported in observations of the rest-frame optical/UV emission of $z \geq 1$ galaxies (e.g. [487–490]) and then in H$\alpha$ line emission (e.g. [491–493]), CO (e.g. [35,48]) and even (in rare cases) the dust continuum and gas emission in strongly lensed sources [51,249]. These massive star-forming

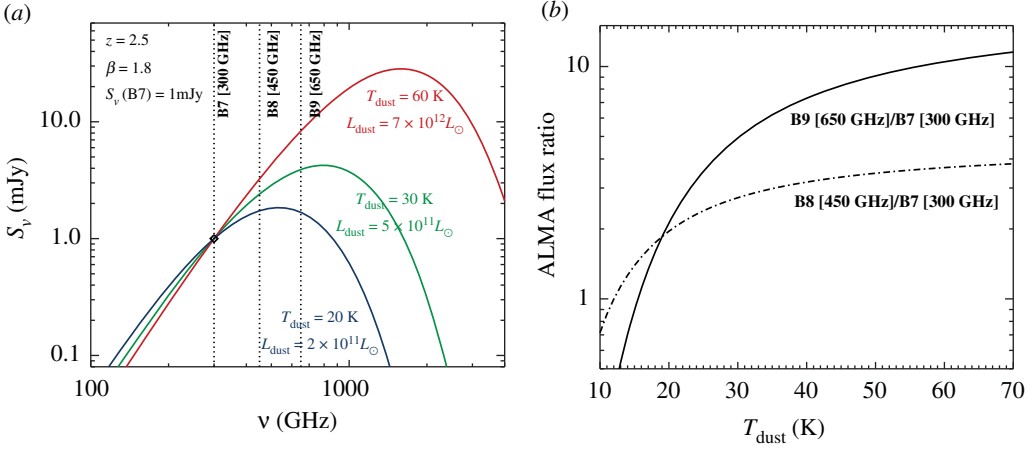

**Figure 19.** (*a*) Model dust SEDs showing a single modified black body spectrum at three different dust temperatures (at $z = 2.5$ and assuming a fixed dust emissivity index, $\beta$). An incorrectly assumed dust temperature for a galaxy (anchored by a single-band dust continuum measurement in Band 7) can change the derived FIR luminosity—and thus implied SFR—by an order of magnitude or more. Obtaining even one additional dust continuum measurement at a different frequency can help constrain the true dust SED, as shown in panel (*b*). While a longer lever-arm in frequency (e.g. including Band 9, solid line) would provide a stronger constraint at fixed S/N, atmospheric and configuration constraints must also be taken into account. The dashed line shows that with high enough S/N measurements, even a neighbouring band (Band 8) can help tightly constrain the dust temperature, and thus the implied dust luminosity and SFR. ALMA enables such studies in high-redshift galaxies on both global and resolved scales.

regions have long been discussed as a ubiquitous feature not only in merging/interacting systems, but also in the gas-rich turbulent discs that are more common at high redshift (e.g. [247,494–496]). However, the nature and importance of these candidate GMCs in the line and/or continuum emission is still debated, in part because they appear less prominent or even invisible in the derived stellar mass maps [497].

ALMA has now allowed searches for such clumps in the line and dust continuum emission of high-redshift galaxies in unprecedented detail. For example, Dessauges-Zavadsky *et al.* [321] reported the discovery of 17 GMCs in the CO(4–3) emission of the $z = 1.036$ Milky Way progenitor the 'Cosmic Snake'. This is an exceptionally strongly lensed galaxy, providing a source-plane resolution as high as 30 pc, and allowing the GMCs to be studied at a resolution comparable to CO observations of nearby galaxies. Beyond such rare cases, ALMA's resolving power has also allowed candidate clumps to be identified in an increasing number of unlensed sources. For example, Iono *et al.* [498] reported two approximately 200 pc clumps in the 860 μm dust continuum imaging of the SMGs AzTEC4 and AzTEC8, as well as approximately $40 > 3\sigma$ clumps in AzTEC1. We note that the latter were apparently embedded in a smooth, more extended (3–4 kpc) emission region, which Hodge *et al.* [214] and Gullberg *et al.* [329] demonstrate may appear clumpy due to the noise inherent in interferometric maps, and must therefore be treated with caution. Subsequent 550 pc-resolution work by Tadaki *et al.* [325] has confirmed that the two brightest off-centre clumps in AzTEC1 are detected in both dust continuum emission and CO(4–3)—a rare example of clumps detected in multiple tracers—where the CO kinematics also suggest that the underlying rotationally supported disc is gravitationally unstable. Meanwhile, studies of other unlensed SMGs in the dust continuum have also confirmed sub-kpc-scale clump-like emission [427,439], while the evidence in IR-fainter sources is still lacking [499]. This could indicate either a different mode of star formation, or insufficient surface brightness sensitivity.

A continued challenge with understanding the properties and importance of these sub-galactic structures has been the lack of correlation between the ALMA 'clumps' and those observed in the rest-frame optical/UV. In particular, no correlation has yet been observed between sub-galactic clumpy structure observed in the UV and that observed in the dust continuum (figure 20; [439]), nor have CO clumps been detected at the position of (off-centre) UV (stellar) clumps (e.g. [321,433]).[8] The lack of co-spatial dust and UV continuum emission suggests that commonly used global SED fitting routines

[8]There have been several studies reporting [CII] 'clumps' aligned with UV clumps in $z > 5$ galaxies (e.g. [500–502]), but note that the term 'clump' is used there to refer to distinct components in what are probably merging systems as opposed to substructure within an extended disc.

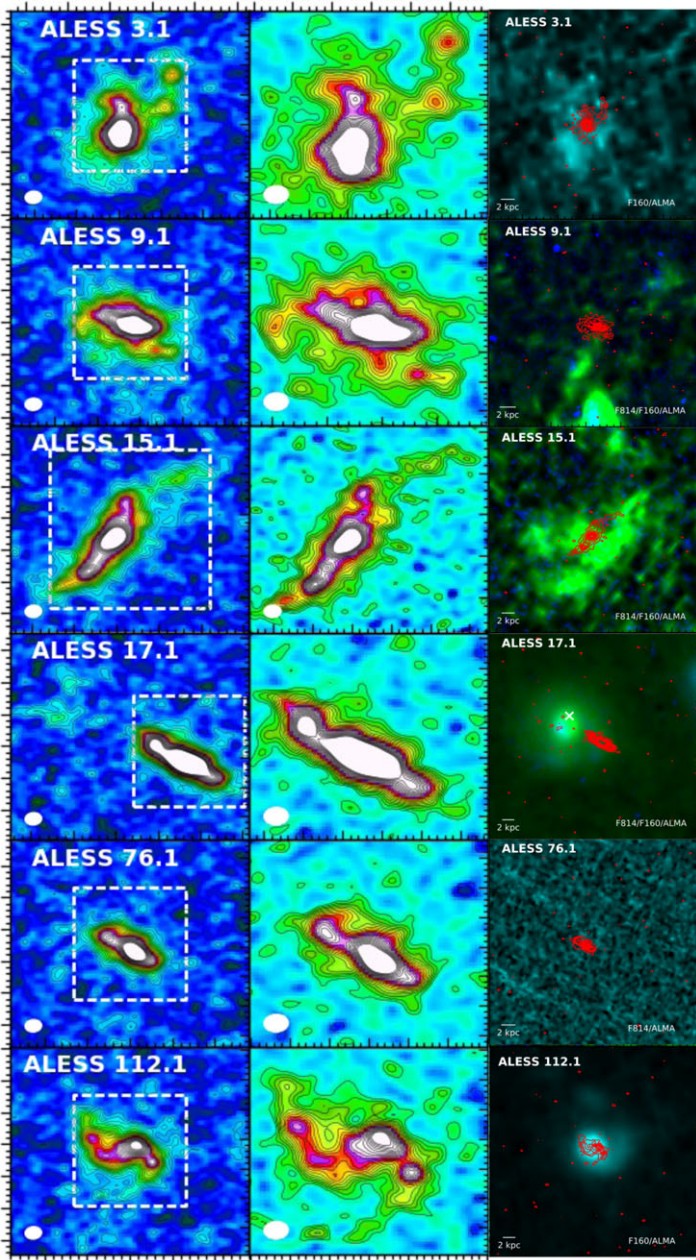

**Figure 20.** ALMA 870 μm dust continuum in high-redshift SMGs from Hodge *et al.* [439] shown as 1.3″ × 1.3″ panels with natural weighting (left column); with robust weighting and zoomed-in to the white dashed boxes (middle column); and as red contours on 4″ × 4″ *HST* false-colour images. The resolution achieved in the middle column corresponds to approximately 500 pc at the redshifts of these sources, allowing a detailed view of these dust-obscured galaxies. The robust dusty substructure observed in these sources with ALMA is uncorrelated with the unobscured stellar populations traced by the *HST* imaging. Figure adapted from Hodge *et al.* [439].

that assume the dust and observed optical/near-IR emission are co-located are too simplistic. Meanwhile, detecting clumps in CO emission can be even more time-consuming for all but the most strongly lensed sources, and is fraught with uncertainties such as the excitation correction and CO-to-$H_2$ conversion factor, and thus the current limits implying, e.g. high star formation efficiencies for the UV clumps [433] are still not particularly constraining. Dessauges-Zavadsky *et al.* [321] report comparable mass distributions for their (CO-identified) GMCs and stellar clumps as those seen in (respectively) the gravitationally bound gas clouds and stellar clumps produced by simulations of fragmenting gas-rich discs [503]; however, the GMCs and stellar clumps are still not co-located. Due to the observational expense of such endeavours even for the most gas-rich galaxies, characterizing the molecular gas (or even dust) properties of the UV clumps (if real) in even fainter galaxies will remain challenging.

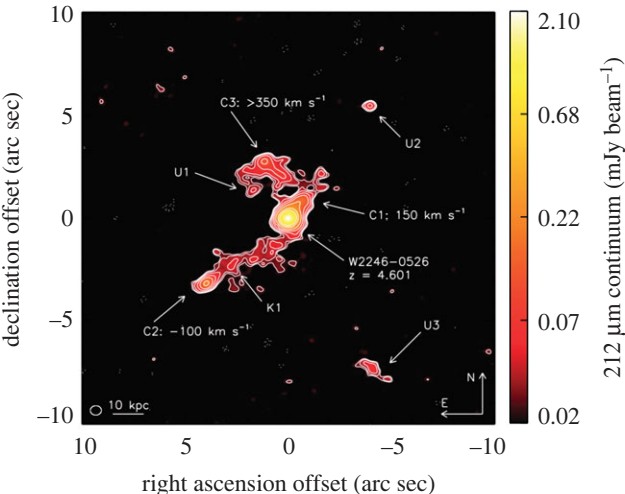

**Figure 21.** ALMA 212 μm dust continuum map ($20'' \times 20''$) of the $z = 4.6$ merging system from Díaz-Santos *et al.* [505]. The labels indicate the three companion galaxies (C1, C2, C3) as well as a number of sources with unknown redshifts. The ALMA imaging reveals a stream of dusty material between W2246-0526 and C2 as well as bridges with C1 and C3, allowing an unprecedented view of this multiple merger event. Figure from Díaz-Santos *et al.* [505].

For the high-redshift galaxies that do have detected substructure with ALMA, the interpretation of the substructure is not limited to clumps. Based on the global morphologies of the SMG substructure detected in the dust continuum by Hodge *et al.* [214] and then Hodge *et al.* [439], they argued that the ALMA observations could be revealing evidence for bars, rings and spiral arms. Using a geometric analysis of an independent SMG sample, Gullberg *et al.* [253] also argued for the existence of bars. While these claims still require kinematic confirmation, observing such non-axisymmetric structures in SMGs would be consistent with the view that these sources are affected by interactions and could help explain the very high star formation rates implied by their long wavelength SEDs (e.g. [80,209,504]).

In another study, Litke *et al.* [470] observed the strongly lensed $z = 5.7$ galaxy SPT0346-52 with ALMA in [CII] and identified two spatially (approx. 1 kpc) and kinematically (approx. 500 km s$^{-1}$) separated components connected by a gas 'bridge', which they argue suggests a major merger. Other observations with ALMA of known mergers/interacting systems have also revealed potential evidence for 'bridges' on larger scales. For example, Carilli *et al.* [461] report evidence for extended [CII] emission between the quasar and SMG in the $z = 4.7$ gas-rich merger system BRI 1202-0725, which they tentatively interpret as a 'bridge', and Oteo *et al.* [426] detect elongated CO(5–4) emission in the $z = 4.425$ pair of interacting starbursts SGP38326. Finally, in an impressive example of ALMA's capabilities, Díaz-Santos *et al.* [505] observed the $z = 4.6$ multiple merger event (and dust-obscured quasar) W2246-0526 in dust continuum and found three galaxy companions connected by streams of dust like tidal tails (figure 21). Studies such as these illustrate the incredible power of ALMA for detailed morphological studies of galaxies in the distant universe.

### 3.2.9. Multi-line studies and other gas tracers

When multiple gas tracers are detected, a comparison of the observed line ratios with theoretical models that take into account the chemistry, radiative transfer and thermal balance of the ISM can provide valuable information on its physical and chemical properties. This includes a comparison with photo-dissociation region (PDR) models (e.g. [506–508]) and X-ray-dominated region (XDR) models (e.g. [509,510]), which ALMA observations have now expanded beyond the typical global studies of submillimetre-selected sources. For instance, Popping *et al.* [511] used ALMA observations of [CI](1–0), CO(3–2), CO(4–3) and the FIR continuum in a $z = 2.2$ 'compact star-forming galaxy' (cSFG) to put constraints on its gas density and UV radiation field strength, deriving starburst-like ISM properties despite its location on the 'main sequence'. Meanwhile, Rybak *et al.* [310] used the angular resolution provided by ALMA in multiple tracers in combination with strong lensing to map these parameters within the $z = 3$ source SDP.81 on approximately 200 pc scales.

In addition to detections in strongly lensed galaxies, ALMA has also allowed the detection of an increasing number of less common molecular and atomic gas tracers (§2.3.2) in high-redshift galaxies

in general. In particular, there are an increasing number of detections of fine-structure lines beyond [CII] 158 µm (see also §3.4). This observational progress has been accompanied by progress in high-resolution radiative transfer modelling of FIR line emission (e.g. [512–515]). These diagnostic lines become particularly important at very high ($z > 4$) redshifts, where most of the commonly used optical/UV nebular lines shift into the mid-infrared and become inaccessible to current instrumentation. The fine structure lines are also less affected by dust extinction, and—due to their brightness—may even be resolved by ALMA within individual unlensed galaxies. For example, Lu *et al.* [516] used ALMA to resolve the [NII] 205 µm emission in the $z = 4.7$ interacting system BRI 1202-0725, following an earlier detection of [NII] in that system with the IRAM interferometer [517]. They then used the ratio of [NII] to CO(7–6) to constrain the dust temperature, using the steep dependence of that ratio on the rest-frame FIR colour $S_\nu\,(60\,\mu m)/S_\nu\,(100\,\mu m)$ (e.g. [518]). The [NII]/[CII] ratio can also help constrain the gas-phase metallicity in HII regions [257,293], though other studies use this ratio to constrain the fraction of [CII] attributable to PDRs, as [CII] comes from both the ionized and neutral medium, while [NII] comes only from the ionized medium. Tadaki *et al.* [460] used this method, along with their resolved observations of [CII] and [NII] in the unlensed $z = 4.3$ SMG AzTEC1, to estimate the fraction of [CII] coming from PDRs in the central 1–3 kpc region. They then used the ratio of [OIII] 88 µm to [NII] to constrain its gas-phase metallicity, finding a value consistent with the extrapolation of the $z = 3$–4 mass-metallicity relation (e.g. [519]). They also attempted a first look at a radial metallicity gradient using resolved ratios, but find no evidence for a positive gradient with the present data. Studies such as these demonstrate the growing utility of the fine structure lines in the ALMA era. In the future, sensitive, high-resolution observations of these line ratios will help disentangle the contributions from the different ISM phases, which may differ from those in local galaxies, particularly at the highest redshifts (e.g. [294]).

## 3.3. The dusty ISM at early epochs

To obtain a complete view of star formation of galaxies, we must account for the fraction of starlight from newborn stars that is obscured by dust (see also [82], for a recent review on the dust attenuation law in galaxies). We know that the fraction of obscured star formation in typical star-forming galaxies is significant out to at least $z \sim 2.5$ [520]. However, due to observational limitations—namely the fact that the *Herschel* space telescope has large PSFs and becomes severely confusion-limited at high redshift (e.g. [521])—directly observing the dust emission from galaxies beyond the peak of cosmic star formation and into the epoch of reionization before ALMA was extremely challenging. The notable exceptions are the rare, bright SMGs (§2) easily detected out to high redshifts with both submillimetre bolometers and *Herschel* (e.g. [80]). But apart from those cases, which may not be representative of the high-redshift galaxy population, the dust content of more typical, low-mass galaxies and the contribution of dust-obscured star formation to the cosmic SFR density at $z > 3$ remained largely unknown until the advent of ALMA. In this section, we briefly review the early ALMA results on dust emission in the very highest-redshift ($z > 5$) galaxies.

### 3.3.1. ALMA observations of Lyman-break galaxies: are the infrared excesses low?

The leap in sensitivity and angular resolution enabled by ALMA means that we can now carry out the deepest continuum observations ever achieved at (sub-)millimetre wavelengths, and attempt to detect (rest-frame far-IR) dust emission in low-mass, optical/near-IR-selected galaxies that are thought to be the dominant contributors to the cosmic SFR density at $z > 5$, notably Lyman-break galaxies (LBGs; e.g. [522,523]).

Since direct observations of the dust emission in high-redshift LBGs were completely unavailable before ALMA, the most widely used method to correct for dust attenuation in rest-frame UV/optical observations of these galaxies and obtain dust-corrected UV luminosities (and hence total SFRs) has been to infer the infrared excess of galaxies (IRX = $L_{IR}/L_{UV}$) from their ultraviolet spectral slope ($\beta$, defined from $f_\lambda \propto \lambda^\beta$, where $f_\lambda$ is the galaxy UV spectrum), the so-called 'IRX-$\beta$ relation'. The clear advantage of this method is that dust attenuation can be directly inferred from the observed UV slope, which is easily accessible with *HST* (e.g. [522,524,525]). This method relies on the tight correlation between IRX and $\beta$ found for local starburst galaxies [526], which are thought to be analogous (at least to some extent) to young galaxies in the high-redshift Universe. The tight relation found for these sources is explained by the fact that they have similar intrinsic UV slopes (young stellar populations), and their location in the IRX-$\beta$ plot is set only by their total dust attenuation: the more dust they have, the redder their observed UV slopes, and

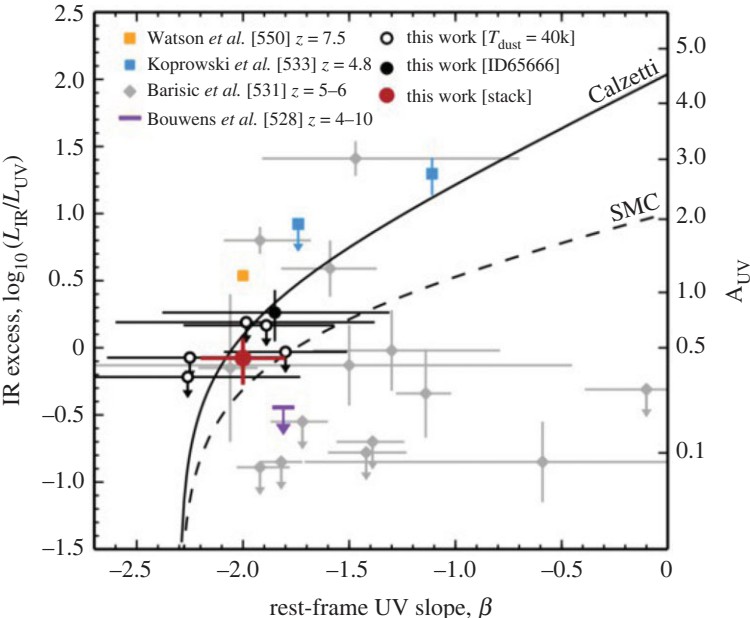

**Figure 22.** The IR excess as a function of UV slope for samples of galaxies observed with ALMA at $z > 4$. The solid line shows the relation expected for a screen of MW-like dust, while the dashed line shows the relation for SMC-like dust. The rest-frame far-IR observations now possible with ALMA in low-mass, optical/near-IR-selected galaxies have allowed the IRX-$\beta$ calibration (and its use for dust attenuation corrections) to be tested for high-redshift galaxies. Figure from Bowler *et al.* [530].

the higher their infrared excess. Galaxies populate this relation in the way that would be expected if we take a screen geometry of Milky Way-like dust [527].

The accuracy with which this IRX-$\beta$ calibration can correct for dust attenuation in high-redshift galaxies has been recently called into question by the first deep ALMA observations of high-redshift LBGs. In particular, observations by Capak *et al.* [445] and Bouwens *et al.* [528] using ALMA Band 6 (approx. 1 mm) find that the infrared luminosities of $z \gtrsim 4$ LBGs measured using ALMA are significantly lower than what would be predicted from their UV slopes using the local Meurer relation. Indeed, they seem to be more consistent with an SMC-like dust extinction curve ([529], figure 22). These results are tantalizing because they may indicate a rapid evolution of the dust content and/or dust properties of star-forming galaxies in the first billion years of cosmic history, and they have generated a good amount of discussion in the community and in the recent literature.

For example, Bowler *et al.* [530] obtained Band 6 (1.3 mm) ALMA observations of six $z \simeq 7$ LBGs; they detect only one of the sources, but using that detection and upper limits, and assuming a dust temperature $T_{\mathrm{dust}} = 40$–$50$ K, they conclude that the infrared excess in their sources is consistent with a Calzetti-like attenuation law, contrary to the findings of Capak *et al.* [445], Bouwens *et al.* [528], and also Barisic *et al.* [531], who reanalysed the sample of Capak *et al.* [445] (figure 22). Part of the disagreement might be due to a different selection of targets (e.g. Capak *et al.* [445] also included narrow-band selected LAEs in their sample). Their results appear to disagree strongly with the Bouwens *et al.* [528] result, and Bowler *et al.* [530] argue that the disagreement might be due, at least in part, to the fact that stacks on $\beta$ bins tend be biased towards low IRX values, as described by McLure *et al.* [532]. We delve into the measurement uncertainties plaguing the IRX-$\beta$ diagram below.

*The impact of measurement uncertainties.* Measurement uncertainties and biases may indeed be quite significant in understanding this problem. McLure *et al.* [532] offer what they call a not definitive but plausible explanation for some of the results that seem to fall below the SMC curve: that this is due to uncertainties in measuring the UV slope (see also [533]). They argue that a combination of $\beta$ measurement uncertainties, with the shape of the mass function, plus the steepness of the IRX-$\beta$ relation at blue UV slopes, means that a given $\beta$ bin may be easily contaminated by bluer galaxies, which can lead to a lower stacked IR luminosity and hence lower derived IRX for that bin. Popping *et al.* [399] also explore the effect of poor photometric sampling of the rest-frame UV spectra on the measurements of the UV slope using their models and find that this can cause significant artificial scatter in the IRX-$\beta$ plane; they conclude that to measure $\beta$ reliably we need a filter combination that at least probes the rest-frame FUV (approx. 1250 Å) and rest-frame NUV (approx. 3000 Å) wavelengths. The importance of accurate UV slope measurements is also highlighted by the analysis

presented in Barisic *et al.* [531], where the UV slopes of the $z \sim 5.5$ LBGs from the Capak *et al.* [445] sample were re-measured using *HST*/WFC3 imaging. They measure systematically bluer UV slopes than those found by Capak *et al.* [445] using lower-resolution, ground-based data, which brings some of the sources closer to the canonical local starburst IRX-$\beta$ relation, but several sources are still more consistent with an SMC-like dust curve, or fall below it. Barisic *et al.* [531] stack rest-UV Keck spectra of those sources and find that they show weak UV absorption features which could be indicative of low metal and dust content in these galaxies, which would presumably explain why their IR excesses are low. Finally, Saturni *et al.* [534] offered another possible source of contamination of the measured UV slopes: weak unresolved AGN that would affect the distribution of UV continuum slopes without altering the IR excess. They find that AGN with bolometric luminosities from $10^{43}$ to $10^{48}$ erg s$^{-1}$ populate the same region of the IRX-$\beta$ diagram as high-redshift LBGs observed with ALMA. However, more observations would be needed to confirm the AGN nature of these sources.

Another large source of uncertainty in placing observed high-redshift galaxies on the IRX-$\beta$ plot is the measurement of the total infrared luminosity from a single ALMA continuum measurement. The deep ALMA observations of high-redshift LBGs have been carried out in Band 6, at around 1 mm, which samples the rest-frame dust emission at approximately 160 μm (some of these observations have targeted the [CII] line at 158 μm; e.g. [445]). To derive the total IR luminosity, typically the assumption is made that the dust emits as an optically thin, single-temperature modified black body, $S_\nu \sim \nu^{\beta_{em}+2} B_\nu(T_{dust})$, where $\beta_{em}$ is the dust emissivity index, and $T_{dust}$ is the dust temperature. The total IR luminosity is then taken to be the integral of this function, normalized to the observed flux in Band 6 (or any other ALMA band). Since there is usually only one data point available, a choice must be made for the parameters $\beta_{em}$ and $T_{dust}$, which can result in large systematic uncertainties of the inferred $L_{IR}$, which is particularly sensitive to the choice of dust temperature (e.g. figure 19). A natural choice adopted by Capak *et al.* [445] and Bouwens *et al.* [528] was to adopt dust temperatures similar to the typical dust temperatures of local galaxies with SFRs close to those of LBGs, i.e. $T_{dust} \sim 25-45$ K. However, as discussed by e.g. Bouwens *et al.* [528] and Faisst *et al.* [535], assuming a hotter dust temperature ($T_{dust} \sim 50-70$ K) would increase the $L_{IR}$ inferred from the same observed millimetre flux by factors of at least a few, and up to an order of magnitude, which could place the $z > 5$ LBGs closer to the local IRX-$\beta$ relation. *Herschel* observations of other samples of galaxies at lower redshifts ($z < 4$) support a trend of increasing dust temperatures (due to stronger radiation fields) with redshift for star-forming galaxies (e.g. [180,182,384]), which could potentially continue until the epoch of reionization (we note, however, that the stellar mass ranges probed are different and there might be selection effects at play, as shown by e.g. [178]). Moreover, Faisst *et al.* [535] find luminosity-weighted temperatures for three $z \sim 0.3$ analogues of high-$z$ LBGs of about 80 K. Higher dust temperatures at high redshift find some additional support from high-resolution radiative transfer simulations (e.g. [167,536–539]).

At the same time, Casey *et al.* [540] argue that not including a mid-infrared component in the dust spectral energy distributions, which contributes around 10–30% of the total IR luminosity, may severely underestimate the IRX, and that including such a component may reconcile observations with the local IRX-$\beta$ relation without the need to resort to very high dust temperatures. They also make the important point that the widely used 'local calibration' of Meurer *et al.* [526] is offset towards bluer colours due to differences in aperture sizes of the UV and IR measurements, and that using the aperture-corrected calibration obtained by Takeuchi *et al.* [541] for the same sample of local starbursts results on approximately 0.3 dex lower IR luminosities. Both factors could go a long way in reconciling the observed ALMA observations of LBGs with the standard Calzetti Law, but tensions still exist.

This is clearly an open issue that will need additional deep multi-band ALMA observations of the dust emission of LBGs at high-redshift, including at higher frequencies, to sample the dust emission peak. Larger samples spanning a wide range of UV slopes and possibly stellar masses are also highly desirable to not only establish if there is a correlation between the UV slopes and the IR excess at high redshift, but also to determine its scatter (we know that there is large scatter in the IRX-$\beta$ relation when more diverse samples are included at both low and intermediate-redshift, and that there is a stellar-mass dependence on the infrared excesses; e.g. [80,203,542–544]), and ultimately its physical drivers.

*Theoretical interpretations.* These puzzling new ALMA results on the IRX-$\beta$ relation have led to a revisiting of the physical drivers of this relation by several theoretical studies. Popping *et al.* [399] used idealized simulations of a screen of dust in front of a stellar population, and explored changing the properties of the dust screen and the stellar population in a controlled setting. Narayanan *et al.* [539] used modern cosmological simulations with radiative transfer to explore the various parameters affecting the positions of galaxies in the IRX-$\beta$ plane. Both studies demonstrate that we can expect a tight relation between the UV slope and the IR excess of galaxies that follows approximately the

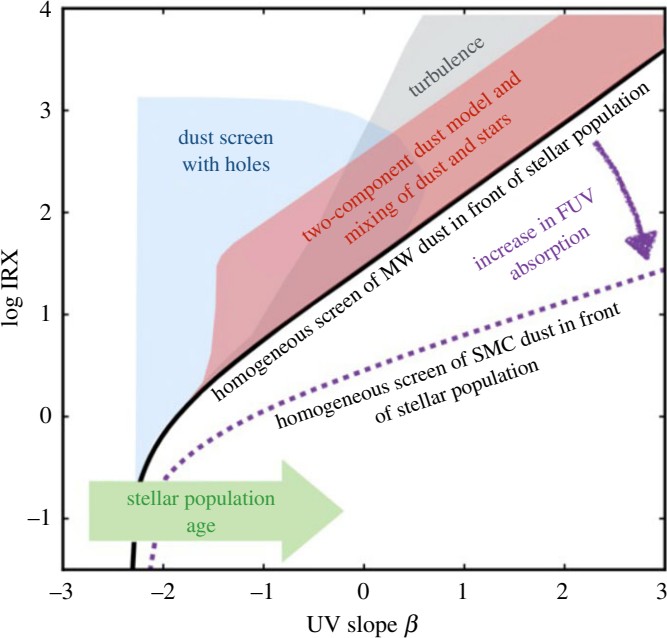

**Figure 23.** The various physical processes affecting the location of galaxies in the IRX-$\beta$ plot according to the theoretical models of Popping *et al.* [399]. Figure from Popping *et al.* [399].

empirical Meurer *et al.* [526] relation if we consider a young stellar population (with intrinsically blue UV slope) behind a uniform screen of dust that has an extinction curve like the Milky Way. Scatter and deviations from this relation can be explained by (see also figure 23):

— older stellar populations that drive galaxies to redder UV slopes at fixed infrared excess (due to the fact that the UV slopes of older stellar populations are intrinsically redder; see also, e.g. [545,546]);
— complex stars/dust geometries that drive galaxies towards bluer UV slopes due to optically thin lines of sight (e.g. Koprowski *et al.* [227], who find the IRX varies by more than a factor of 3 across a spatially resolved galaxy at $z = 3$ due to the complex morphologies of UV and IR-emitting regions);
— shallower extinction curves (such as the SMC extinction curve) that lead to lower infrared excess at fixed $\beta$ due to lower near-infrared to far-infrared extinction ratios, i.e. lower total energy absorbed by dust.

We know that, even for local galaxies, the IRX-$\beta$ relation becomes much less tight—with galaxies populating the various regions of the IRX-$\beta$ plane—when different selections are applied. Therefore, at least some of these effects are important even at low redshift.

In another study, Mancini *et al.* [547] post-process hydrodynamical models of galaxy formation with chemical evolution and dust formation/destruction. Interestingly, they find that in their models, young, low-mass galaxies, where dust grains are mostly from stellar sources, fall below the Meurer *et al.* [526] relation in the IRX-$\beta$ diagram, indicating an intrinsically steeper dust extinction curve. Meanwhile, more massive galaxies with efficient ISM dust growth introduce scatter to the relation and shift towards the Meurer *et al.* [526] line at $z \lesssim 6$. This demonstrates how dust growth processes in early galaxies might be affecting their large-scale observables.

A different theoretical explanation comes from Ferrara *et al.* [548], who proposed that the IR emission deficit at high redshifts could be explained by larger molecular gas fractions of high-$z$ galaxies. In these galaxies, a large fraction of the dust mass would be embedded in dense gas and remain cold, therefore not contributing to increasing the infrared luminosity. Somewhat counterintuitively, this model leads to the suggestion that this far-IR deficit might provide a new way of finding galaxies with large molecular gas fractions at high-redshift, which remains to be confirmed by actual CO observations.

### 3.3.2. Robust $z \simeq 8$ continuum detections with ALMA

In contrast with the apparently lower-than-expected IR luminosities in the studies described above, there have been a few notable examples of very robust detections of the dust continuum of primordial galaxies

**Table 3.** The highest-redshift dust continuum detections of LBGs with ALMA to date.

| source | $z$ | SFR/ $M_\odot$ yr$^{-1}$ | $M_*/M_\odot$ | $M_{dust}/M_\odot$ | reference |
|---|---|---|---|---|---|
| A1689-zD1 | 7.5 | 9 | $2 \times 10^9$ | $4 \times 10^7$ | Watson *et al.* [550] |
| MACS0416_Y1 | 8.31 | 60 | $2 \times 10^8$ | $4 \times 10^7$ | Tamura *et al.* [552] |
| A2744_YD4 | 8.38 | 20 | $2 \times 10^9$ | $6 \times 10^6$ | Laporte *et al.* [551] |

well into the epoch of reionization with ALMA. In these cases, the detected galaxies seem to follow a Milky Way-like relation in the IRX-$\beta$ plot, which is perhaps indicative of a range of galaxy dust properties at high redshift similar to that seen at low redshift (e.g. [530,549]). The highest-redshift continuum detections of star-forming galaxies with ALMA at the time of writing are A1689-zD1 at $z = 7.5$ [550], A2744_YD4 at $z = 8.38$ [551] and MACS0416_Y1 at $z = 8.31$ [552]. All of these galaxies are LBGs identified using the drop-out technique that are strongly gravitationally lensed by massive foreground clusters (see also [553], for a strong dust continuum detection in a $z = 7.5$ quasar host). These sources are characterized by surprisingly high dust masses of approximately $10^7 \, M_\odot$ (table 3). Their measured dust-to-stellar mass ratios are as high as, or even in excess of, those measured in present-day galaxies (e.g. [213]), which presents a challenge to chemical enrichment and dust formation models.

Dust grains form mainly via condensation of heavy elements in dense and cool regions such as supernovae remnants and the envelopes of evolved stars, namely asymptotic giant branch (AGB) stars, and they can further grow via accretion in dense molecular clouds (e.g. [554–558]). Supernova dust starts contributing as soon as the first type II SNe explode in galaxies, on short timescales of around 10 Myr; AGB stars start contributing at about 1 Gyr. At $z \simeq 8$, the Universe is less than 1 Gyr old, and hence, according to current models, AGB stars cannot be a major contributor to the large dust masses measured with ALMA. Modelling shows that in order to explain those dust masses, high SNe rates are needed (thanks to high star formation rates and/or more top-heavy IMFs), combined with high SNe dust yields (or low destruction rates), but given current estimates of SNe yields, fast and efficient dust growth in the dense interstellar medium is also required (e.g. [208,377,559,560]). Chemical enrichment and dust growth models have shown that dust growth in the dense ISM is a major contributor to the dust mass of the Milky Way [554,558]. However, in the epoch of reionization, this poses a problem, because models show that ISM dust growth only starts being efficient after a critical metallicity has been reached in the ISM [557,561]; we note that models like these are still relatively uncertain because of large uncertainties in sticking coefficients, growing mechanisms and destruction rates. Therefore, a rapid metal enrichment in the ISM of these $z \simeq 8$ galaxies would be required. Furthermore, Ferrara *et al.* [562] make the point that at high redshifts, ISM dust growth may be problematic due to the higher ISM temperatures and densities; they argue that grain growth can occur in the cooler dense molecular clouds where they are more sheltered, but the icy mantles do not survive in the diffuse ISM.

Along with the still large uncertainties in dust formation and growth modelling, there are still many observational uncertainties, including the star formation histories and chemical enrichments of galaxies at the epoch of reionization. Current *HST* and *Spitzer* data only probe the rest-frame UV of these galaxies, and therefore their stellar masses and past star formation histories are still quite uncertain. Tamura *et al.* [552] argue that their observations are consistent with the existence of an underlying older (300 Myr) stellar population in the galaxy that does not contribute to its UV SED but could imply a higher stellar mass (and hence lower dust-to-stellar mass ratio). Such a stellar population would have provided early chemical enrichment of the ISM and dust growth. This hypothesis needs to be tested with rest-frame optical/near-infrared observations that will soon be enabled with *JWST*, which will also enable more accurate measurements of the gas-phase metallicity, a crucial ingredient in grain growth. Observations with *JWST* also have the promising potential of constraining dust attenuation more precisely at those redshifts, which could be used to test models that predict the grain optical properties and size distributions in the context of dust formation models (e.g. [547,563,564]).

At the same time, current dust mass measurements are still highly uncertain, as they are often based on only one or two ALMA flux measurements. Assuming the simple case of cool isothermal, optically thin dust contributing the majority of the dust mass, most single-band measurements still need to include at least three parameters: the dust temperature, the dust emissivity spectral index and the dust emissivity normalization. Typically, the dust emissivity properties at high-redshifts are assumed to be similar to

those measured in the local Universe, simply because we lack empirical measurements. However, both theoretical models and laboratory studies indicate that different types of dust grains could have widely different emissivity properties (e.g. [565]). For example, different chemical compositions and structures of young dust grains can lead to different opacities per unit mass (e.g. [378], and Appendix B of [566]). It is not far-fetched to consider that dust grains in primordial galaxies could be significantly different in their physical properties than dust grains in the present-day Milky Way, given the potentially different physical conditions in the ISM, metals available, and dominant formation mechanisms. The dust emissivity indices and temperatures are still uncertain and have not been directly measured for high-redshift sources of this kind. This requires multi-frequency ALMA observations (e.g. da Cunha *et al.* [375]). It is crucial to obtain better constraints on these properties, as they can introduce significant systematic uncertainties in the derived dust masses. As an example, assuming $\beta_{em} = 1.5$ instead of $\beta_{em} = 2$ can lead to dust masses between 5 and 10 times higher depending on the temperature; assuming $T_d = 80$ K instead of $T_d = 30$ K can lead to dust masses between 100 and 300 times higher depending on the emissivity index. Additionally, CMB effects become crucial at those redshifts: da Cunha *et al.* [366] show that ignoring the effect of the CMB on dust heating and (sub-)mm observations can lead to severe overestimation of the dust emissivity index and underestimation of the dust mass at $z > 5$. Finally, even the optically thin assumption can introduce severe systematic biases, with models that consider more general opacity scenarios retrieving typically higher dust temperatures and lower dust masses than optically thin models (e.g. [375,567]).

## 3.4. ALMA spectroscopy at the high-redshift frontier

Thanks to its sensitivity and frequency range, ALMA has been considered a promising 'redshift machine' for the very distant Universe (though the modest bandwidth and large overheads still imply a significant time investment for all but the brightest sources). One of the most promising lines to target is the [CII] fine structure line at 158 µm. [CII] is one of the main ISM cooling lines and the brightest far-infrared line in most star-forming galaxies, carrying typically around 1% of their total infrared luminosity (e.g. [568]). [CII] has the advantage of being observable even towards neutral sightlines in the epoch of reionization (contrary to Ly$\alpha$). Moreover, it has also been considered a promising tracer of the star formation rate of galaxies, and because it is typically a bright line, it can additionally be used to trace their gas dynamics.

Early studies with ALMA targeted the [CII] line in known $z > 6$ bright Lyman-$\alpha$ emitters (LAEs). These galaxies are known to have high star formation rates, and their redshifts are known thanks to the Ly$\alpha$ line, making them prime targets. However, the first studies with ALMA surprisingly failed to detect the [CII] line (e.g. [569–571]). This implied that these sources might not follow the local correlation between [CII] luminosity and star formation rate (or infrared luminosity); i.e. these bright LAEs seem to have a [CII] deficit (see also, e.g. [436,572,573], §3.2.5). Indeed, the large statistical study of approximately 1000 LAEs of Harikane *et al.* [574], which includes ALMA [CII] measurements for a subsample of 34 sources, shows that there is an anti-correlation between the [CII]-to-SFR ratio and the Ly$\alpha$ equivalent width (see also Pentericci *et al.* [575], who successfully detected [CII] emission in $z \sim 7$ LAEs with fainter Ly$\alpha$ emission). This is probably a consequence of low metallicities, high ionization parameters, and strong radiation fields in high-redshift galaxies with very prominent Ly$\alpha$ emission and/or the [CII] emission coming from very high-density photodissociation regions in these galaxies (see discussion in Harikane *et al.* [574]; see also modelling efforts by Vallini *et al.* [515] and Lagache *et al.* [576]).

Other studies have focused on searching for [CII] in more 'normal' star-forming galaxies where no bright Ly$\alpha$ emission is detected, selected with the Lyman-break technique (i.e. Lyman-break galaxies). Capak *et al.* [445] and Willott *et al.* [446] targeted small samples of LBGs with prior spectroscopic redshifts and successfully detected the [CII] line in those sources, in contrast to LAE studies. They found that their LBGs had bright [CII] emission but low IR luminosities (most were undetected in the continuum; see §3.3), implying [CII]-to-IR ratios similar or even higher than found for local galaxies. This could presumably be because the ISM conditions in those sources are more similar to local galaxies; however, it is still puzzling that they seem to have lower IR luminosities than expected, as discussed in §3.3. Nevertheless, even though the physical origin of the [CII] in these galaxies is still a matter of debate, its brightness in LBGs means that it can be used to pinpoint the redshift and measure the dynamics of these distant sources with ALMA. Smit *et al.* [289] demonstrated this by using ALMA to measure the redshift of two LBGs at $z \simeq 6.8$ that had been selected on the basis of their photometric redshifts alone and had no previous spectroscopic redshifts from other instruments (figure 24). The strong [CII] detections (and the less than approx. arcsec angular resolution enabled by ALMA) allowed them to make the first dynamical maps of 'normal' galaxies at the epoch of

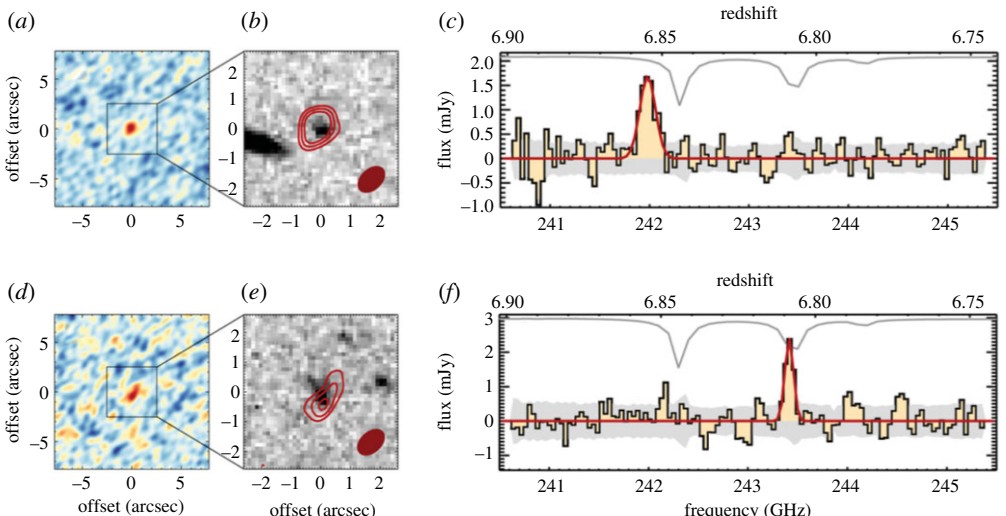

**Figure 24.** ALMA provides spectroscopic confirmation of the redshift of two Lyman-break galaxies with photometric redshifts in the range $6.6 < z < 6.9$ via detection of the [CII] line. Panels (a,d) show the ALMA Band 6 observations averaged, panels (b,e) show the ALMA line-averaged contours overlaid on deep *HST* optical imaging at 1.6 μm, and panels (c,f) show the ALMA spectra of the [CII] lines. Figure reproduced from Smit *et al.* [289].

reionization, which seem to indicate rotation in these sources (although at the current angular resolution, mergers cannot be excluded). Higher resolution follow-up with ALMA will provide valuable additional information in the near future; see also the 0.3″ resolution study of a Lyman-break galaxy at $z = 7.15$ by Hashimoto *et al.* [549], that finds dynamical evidence for a major merger-induced starburst.

Another promising line in studies of very high redshift objects with ALMA is the [OIII] fine structure line at 88 μm, which is observable in Band 7 at $8 \lesssim z \lesssim 11$. The [OIII] 88 μm line is predicted to be very bright in young galaxies, easily outshining [CII] in sources with intense radiation fields and low metallicities (see [513,577–580], for theoretical predictions); high [OIII]-to-[CII] ratios are also observed in local low-metallicity dwarf galaxies (e.g. [295,581]). This makes the [OIII] line more easily detectable than [CII] in sources such as LAEs. The first detection of [OIII] at high redshift with ALMA was of a LAE at $z = 7.212$ by Inoue *et al.* [292]. Since then, various other detections have been made, demonstrating that [OIII] is often more easily detectable than [CII] in $z > 7$ sources (e.g. [549,551,552,582]; see also, e.g. [583] for an ALMA study of the [OIII] emission in a high-redshift quasar host galaxy). These sources with high [OIII]-to-[CII] luminosity ratios are thought to have very little neutral gas ([OIII] arises mainly from HII regions while [CII] arises mainly from the neutral ISM/photodissociation regions), and ionizing photons are able to escape their ISM, making them potentially important sources of cosmic reionization [292]. Multi-tracer studies can dissect the multi-phase ISM as well as material inflows/outflows of such sources by analysing the spatial distribution and velocity offsets of Lyα, [OIII] and [CII] emission (when detected) with resolved observations (e.g. [204,291,459,549,572,584,585]).

The use of [CII] and [OIII] lines to not only confirm sources at the epoch of reionization but also study their detailed physical properties is only starting. The importance of this topic has been recently recognized by the ALMA community with a Cycle 7 Large Programme, REBELS (An ALMA Large Programme to Discover the Most Luminous [CII]+[OIII] Galaxies in the Epoch of Reionization; PI: Bouwens), which is building a large statistical sample of 40 UV-bright star-forming galaxies with photometric redshifts $6.5 < z < 9.5$. Preliminary results at the time of writing are showing the power of ALMA spectral scans to efficiently obtain spectroscopic redshifts for these distant sources; detailed studies of these emission lines will also bring new insight into their physical properties and kinematic structure, as demonstrated in Smit *et al.* [289]. Importantly, REBELS is also detecting the dust continuum at the epoch of reionization, which will be crucial to understand the ISM evolution and dust growth in the early Universe. Such a sample holds promising targets for future follow-up with the *JWST*.

We have truly entered the era of ALMA spectroscopic studies of galaxies at frontier distances. The current redshift record holder is the detection of the [OIII] line in MACS1149-JD1, a gravitationally lensed, $10^9\,M_\odot$ stellar mass galaxy at $z = 9.1$ ([582], figure 25).

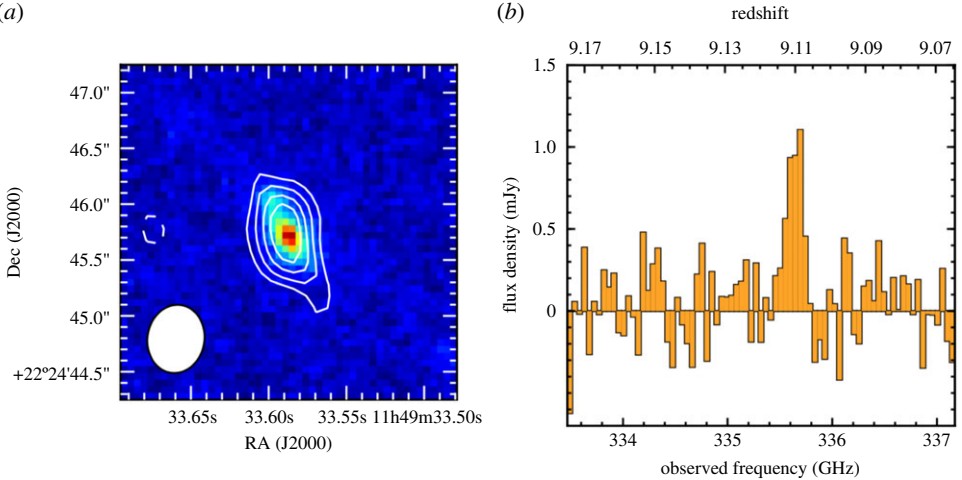

**Figure 25.** Current redshift record-holder with ALMA: [OIII] line detection at $z = 9.1096$. (a) *HST* image (F160W) with ALMA [OIII] contours. (b) ALMA spectrum, showing the 7-$\sigma$ [OIII] 88 μm line detection in Band 7. Figure reproduced from Hashimoto *et al.* [582].

## 3.5. The effect of the cosmic microwave background in high-redshift observations

Observational studies of the cool interstellar medium at high redshifts have the potential of being severely impacted by the cosmic microwave background (CMB), whose temperature approaches or even exceeds that of the ISM components being studied at those redshifts. This concern specifically affects observations of the (cold) dust continuum, CO lines and potentially also [CII] lines.

Da Cunha *et al.* [366] summarize the effect of the CMB in high-redshift (sub-)millimetre observations of the dust continuum and also CO lines (though we note the latter have also been treated before, with the effect often being taken into account in line modelling studies; e.g. [586–589]). Essentially, the CMB affects the observed (sub-)millimetre dust continuum and the line emission in two ways: first, it provides an additional source of (both dust and gas) heating, and second, it is a non-negligible background against which the line and continuum emission are measured. Da Cunha *et al.* [366] quantify how these two competing processes affect ALMA (or any (sub-)millimetre) measured fluxes and provide correction factors to compute what fraction of the intrinsic dust (and line) emission can be detected against the CMB as a function of frequency, redshift and temperature. They also discuss how the physical interpretation of ALMA observations is affected: for example, the inferred dust and molecular gas masses can be severely underestimated, while dust emissivity indices and temperatures can be overestimated if the impact of the CMB is not properly taken into account. The effect on the inferred dust emissivity indices is discussed in Jin *et al.* [201] for a sample of four galaxies at $z = 3.62$–5.85 in the COSMOS field. They find that, when ignoring the effect of the CMB on the dust SEDs, the Rayleigh–Jeans slopes are unusually steep (with inferred dust emissivity indices $\beta_{em} \sim 2.4$–3.7), while they become consistent with 'normal' (i.e. typical measurements, mostly obtained for low-redshift galaxies, $\beta_{em} \simeq 2$), when the effect of the CMB is taken into account. They argue that this is the first direct evidence of the impact of the CMB on galaxy observables at high redshifts. Indeed the CMB effect cannot be ignored especially as we move to sampling higher-redshift galaxies in the RJ regime; however, the possibility that the emissivity indices are different at high-redshift (which could be the case if dust grains have different properties) cannot be ruled out with existing data.

Zhang *et al.* [590] extended the analysis of the effect of the CMB on (sub-)millimetre observations to spatially resolved observations. They point out that, in galaxies with dust (and gas) temperature gradients, the different contrast between the galaxy emission and the CMB in different regions (due to different temperatures), can significantly affect resolved imaging and dynamical studies. For example, in galaxies where the dust temperature decreases with radius, the cool dust in the outer regions might not be visible against the CMB background, and therefore the size of the dust-emitting region might be underestimated (this is true for galaxies at any redshift, but particularly relevant for ALMA observations at high redshifts).

The CMB also provides extra heating and background for cool gas emission lines such as CO and [CII], though the situation is further complicated by the need to know the excitation temperatures ($T_{exc}$) of the different lines, because the contrast of a given line against the CMB background is set by the difference between its $T_{exc}$ and the CMB temperature at that redshift (e.g. [366]). For CO, in the

local thermodynamic equilibrium (LTE) case (where we can assume that the excitation temperature of all lines is the same and equal to the kinetic temperature), the fraction of intrinsic line (velocity-integrated) flux that is recoverable against the background decreases monotonically with increasing redshift and decreasing $J$; since the magnitude of the effect is not exactly the same for all $J$ transitions, the shape of the spectral line energy distribution can be distorted (e.g. [366,591]). For non-LTE cases, the $T_{exc}$ and optical depth of each transition must be computed using detailed models (e.g. LVG modelling; [592]), and while the behaviour with redshift is still similar, in the sense that all CO lines become harder to observe at higher redshifts, the way different transitions are affected depends strongly on the physical properties of the gas (see examples in [366]).

Similarly for [CII] lines, we must model their excitation in detail, though this is further complicated by the fact that [CII] originates from ISM phases with vastly different ISM conditions, from ionized gas to diffuse neutral gas to PDRs (e.g. [593]). Vallini *et al.* [515] perform detailed calculations of the [CII] excitation (or 'spin') temperature in the context of high-resolution, radiative transfer cosmological simulations at $z \simeq 7$. They obtain $T_{exc} \sim 30$–$120\,K$ in PDRs (with SFR $\sim 0.1$–$100\,M_\odot\,yr^{-1}$), and $T_{exc} \sim 22$–$23\,K$ in the cold neutral medium (CNM). They conclude that the CMB has a negligible effect on the [CII] emission from galaxies at $z \lesssim 4.5$, but e.g. at $z \sim 7$, the emission from the CNM is strongly attenuated due to the strong CMB background, which has a temperature close to $T_{exc}$ in that component. Therefore, even at high redshifts, the [CII] emission from star-forming regions is mostly robust against CMB effects, but we might lose the ability to detect an extended cooler gas component using that line. Lagache *et al.* [576] use a semi-analytic model to predict the [CII] luminosity functions from $z = 4$ to $z = 8$ from galaxy-wide properties, and they find that the CMB systematically reduces their normalizations by approximately 25–35% (though note that they only include the PDR component).

# 4. Blind surveys with ALMA

## 4.1. Motivation and summary of existing surveys

Pre-selection of galaxy samples from deep surveys at optical/near-infrared wavelengths as discussed in the previous section has the disadvantage that we might be biasing our view of galaxy evolution by using only specific subsets of the more general population, or even missing a potentially important population of gas-rich galaxies that are not included in those samples for being too optically faint (e.g. [99,594]). If such a population is important, this could make it very challenging to, for example, reliably trace the evolution of dust and CO luminosity functions, understand the cosmic evolution of the gas content of galaxies, and test the predictions of galaxy formation models (e.g. [595–597]).

This motivates the execution of blind surveys with ALMA which aim to detect the continuum and CO/[CII] lines of galaxies in an unbiased and complete way, and without suffering from the effects of source confusion of other instruments. Given the relatively small fields-of-view achievable with one single pointing (the ALMA primary beam FWHM ranges from about 9 arcsec in Band 9 to about 1 arcmin in Band 3), it is challenging to execute blind surveys over large, cosmologically important areas. Even so, given how important/necessary blind surveys are, significant time has been invested in executing a few of these surveys since the start of ALMA operations (table 4), which we summarize in this section.

These (continuum and line) deep surveys have so far focused on:

— Pushing down the continuum detection limits in the (sub-)millimetre in order to constrain the faint end of the number counts, to resolve and characterize the sources responsible for the extragalactic background light (EBL), and to measure their contribution to the cosmic star formation history.
— Measuring the dust and molecular gas content of high-redshift galaxies in an unbiased way (i.e. without prior pre-selection at lower wavelengths).
— Measuring the $H_2$ mass function at various redshifts (through measurements of CO luminosity functions), and tracing the evolution of the cosmic density of $H_2$.
— Characterizing the ISM of galaxies near the epoch of reionization through searches for dust continuum and [CII] emission at $z > 4$ by leveraging the very high sensitivity achieved with ALMA and/or high magnifications enabled by strong gravitational lensing towards galaxy clusters.

We focus on recent results from the main deep blind surveys executed to date, which are summarized in table 4, and which we briefly describe here:

**Table 4.** Summary of blind extragalactic surveys executed so far with ALMA.

| survey | description | area (arcmin$^2$) | 1-$\sigma$ depth | resolution (arcsec) |
|---|---|---|---|---|
| ASPECS Pilot | full frequency scans | 1.0 | $L'_{CO} \sim 2 \times 10^9$ K km s$^{-1}$ pc$^2$ | |
| Walter et al. [216] | B3 (3 mm): 1 pointing | | 3.8 μJy beam$^{-1}$ | 2.8 |
| | B6 (1.2 mm): 7 pointings | | 12.7 μJy beam$^{-1}$ | 1.3 |
| ASPECS Large Programme | full frequency scans | 4.6 | $L'_{CO} \sim 2 \times 10^9$ K km s$^{-1}$ pc$^2$ | |
| Decarli et al. [78] | B3 (3 mm): 17 pointings | | 3.8 μJy beam$^{-1}$ | 1.8 |
| González-López et al. [598] | B6 (1.2 mm): 85 pointings | | 9.3 μJy beam$^{-1}$ | 1.5 |
| HUDF Continuum Image | B6 (1 mm): 45 pointings | 4.5 | 35 μJy beam$^{-1}$ | 0.7 |
| Dunlop et al. [599] | | | | |
| ALMA-SXDF | B6 (1.1 mm): 19 pointings | 1.5 | 55 μJy beam$^{-1}$ | 0.5 |
| Kohno et al. [600] | | | | |
| ALMA Frontier Fields | B6 (1.1 mm): 3 × 126 | | | |
| González-López et al. [79] | pointings | | | |
| | Abel 2744 | 4.6 | 55 μJy beam$^{-1}$ | 0.6 |
| | MACSJ0416 | 4.6 | 59 μJy beam$^{-1}$ | 1.2 |
| | MACSJ1144 | 4.6 | 71 μJy beam$^{-1}$ | 1.1 |
| SSA22/ADF22A | B6 (1.1 mm): | | | |
| Umehata et al. [69] | 103 pointings | | | |
| | FULL/LOWRES | 7.0 | 75 μJy beam$^{-1}$ | 1.0 |
| | DEEP/HIRES | 5.8 | 60 μJy beam$^{-1}$ | 0.7 |
| SSA22/ADF22B | B6 (1.1 mm): | 13 | 73 μJy beam$^{-1}$ | 0.53 |
| Umehata et al. [601] | 133 pointings | | | |
| GOODS-ALMA | B6 (1.13 mm): | 69 | | |
| Franco et al. [64] | 846 pointings | | | |
| | native | | 110 μJy beam$^{-1}$ | 0.24 |
| | tapered | | 182 μJy beam$^{-1}$ | 0.6 |
| ASAGAO | B6 (1.2 mm): 9 × 90 | 26 | 61 μJy beam$^{-1}$ | 0.5 |
| Hatsukade et al. [66] | pointings | | | |

*ASPECS.* The ALMA SPECtroscopic Survey (ASPECS) started with a pilot programme that targeted a $\simeq 1$ arcmin$^2$ region in the *Hubble* Ultra-Deep Field (UDF), which was then extended as a Large Programme to mosaic an area about five times larger ($\simeq 4.6$ arcmin$^2$ covering most of the *Hubble* eXtreme Deep Field; [602]) to similar depths, and also coinciding with deep MUSE ancillary data in the UDF [603]. ASPECS consists of full frequency scans in ALMA Bands 3 and 6, continuously covering the frequency ranges from 84 to 115 GHz and from 212 to 272 GHz at approximately uniform CO line sensitivity $L'_{CO} \sim 2 \times 10^9$ K km s$^{-1}$ pc$^2$; see Walter *et al.* [216] for a full pilot survey description; see also Decarli *et al.* [78], González-López *et al.* [62,598] for descriptions of the large programme. The frequency ranges and depth were chosen to maximize the redshift coverage of the survey with various CO lines and sample the knee of the predicted CO luminosity functions, as well as possibly detecting [CII] at the highest redshifts ($6 < z < 8$). This is the deepest blind field performed to date with ALMA, with continuum noise levels achieved of 3.8 μJy beam$^{-1}$ in Band 3 ($\simeq 3$ mm) and 9.3 μJy beam$^{-1}$ in Band 6 ($\simeq 1.2$ mm).

*HUDF Continuum Image.* This survey [599] performed a 45-pointing mosaic with Band 6 of the *Hubble* Ultra Deep Field (HUDF) over $\simeq 4.5$ arcmin$^2$. They obtain a contiguous and homogeneous 1.3 mm image reaching a depth of 35 μJy beam$^{-1}$ (largely superseded by the ASPECS large programme now). Like ASPECS, the field was chosen to maximize the overlap with the exquisite deep coverage available at

other wavelengths (specifically with *HST* and *Spitzer*), in order to confirm and characterize the blind continuum detections, as well as to enable deep stacking of optical/near-infrared-selected samples. The choice of band and sensitivity was designed to maximize detections of $z > 3$ dusty galaxies while keeping the survey feasible (more details in [599]).

*ALMA-SXDF.* This survey [600,604] covers a 1.5 arcmin$^2$ rectangular region in the Subaru/XMM-Newton Deep Survey Field [605], where deep ancillary multi-wavelength observations are available from the X-rays to the radio. The targeted region is chosen to include a bright 1.1 mm source detected with AzTEC, and 12 Hα-bright star-forming galaxies at $z \simeq 2.5$ detected using narrow-band imaging.

*ALMA Frontier Fields Survey.* This survey, described in González-López *et al.* [79], targeted three strong-lensing galaxy clusters from the *Hubble* Frontier Fields [606]: Abel 2744, MACSJ0416 and MACSJ1149. The goal is to combine the sensitivity of ALMA with the strong gravitational lensing magnifications towards these clusters to reach the faintest dusty star-forming galaxies, and thus probe the low-luminosity regime as well as to detect very high-redshift sources (e.g. [551]). Each cluster was covered using a 126-pointing mosaic in Band 6 (1.1 mm), reaching sensitivities of 55, 59 and 71 $\mu$Jy beam$^{-1}$, respectively (with the Abel 2744 field having the deepest and most uniform data).

*ALMA Deep Field in SSA22 (ADF22).* This survey first mosaicked a $\sim 2 \times 3$ arcmin$^2$ area with Band 6 at 1.1 mm (ADF22A region, [69]), targeting the core region of a protocluster at $z = 3.09$, which had been previously identified via overdensities of Lyman-break galaxies [607] and Lyman-α emitters [608]. One of the main goals was to detect and characterize dusty star formation in the protocluster. This survey detected 18 SMGs at greater than $5\sigma$, 10 of which are spectroscopically confirmed at the redshift of the protocluster. A follow-up mapped a contiguous area, ADF22B to a similar depth, bringing the combined area of the SSA22 ALMA coverage to 20 arcmin$^2$ (71 co-moving Mpc$^2$ at the protocluster redshift; [601]). This combined ADF22 area contains a total of 35 SMGs at $>5\sigma$, with star formation rates approximately 100–1000 $M_\odot$ yr$^{-1}$. This is a clear overdensity of millimetre sources in the protocluster core (by a factor of 3–5 compared with blank-field number counts), suggesting that intense dusty star formation may be enhanced by the large-scale environment, as also found in other studies (e.g. [609]).

*GOODS-ALMA.* This survey [64] targeted the largest contiguous area surveyed by ALMA so far, a 69 arcmin area in GOODS-South using 846 pointings in Band 6 (1.13 mm). The observations were taken at 0.24 arcsec resolution to a mean depth of 110 $\mu$Jy beam$^{-1}$; however, the main source extraction and science analysis in Franco *et al.* [64] are done using a map tapered to 0.60 arcsec (to reduce the number of independent beams) with an RMS sensitivity of 182 $\mu$Jy beam$^{-1}$. The targeted area was chosen to match the deepest *H*-band imaging of the GOODS-South field, enabling identification of the counterparts. One finding from this survey is that about 20 percent of the ALMA detections are *HST*-dark galaxies, which could be at $z > 4$ (see also §3.2.2 and [610,611]).

*ALMA 26 arcmin$^2$ survey of GOODS-S at 1 mm (ASAGAO).* This survey [66] targeted again the GOODS-South field in Band 6 (1.2 mm). An area of 26 arcmin$^2$ was covered in using nine tiles of approximately 90 pointings each, and a resolution of about 0.5 arsec and a depth of 61 $\mu$Jy beam$^{-1}$ were reached. Hatsukade *et al.* [66] also combined their observations with previous deep observations of GOODS-South at similar resolutions from the HUDF [599] and GOODS-ALMA [64] to obtain a deeper 1.2 mm map that reaches a sensitivity of approximately 26 $\mu$Jy beam$^{-1}$ in the central area (essentially the area covered by the Dunlop *et al.* [599] observations).

*Archival surveys.* Along with these targeted fields, another productive approach has been to mine the public ALMA Science Archive for existing observations to obtain deep measurements over large combined areas in sometimes random areas of the sky, which helps overcome cosmic variance. The ALMACAL survey [612] exploits observations of ALMA calibration fields in various frequency bands and array configurations. Using observations of 69 calibrators, they reached depths of approximately 25 $\mu$Jy beam$^{-1}$ at sub-arcsec resolution, and detected 8 and 11 faint dusty star-forming galaxies at greater than or equal to $5\sigma$ in Bands 6 and 7, respectively (another interesting application of the ALMACAL survey is the work by Klitsch *et al.* [304], who measured upper limits on the cosmic molecular gas density using CO absorption towards distant quasars). Others search for all deep ALMA pointings in certain bands available from the archive. Fujimoto *et al.* [70] combined 120 pointings in Band 6 to study faint dusty star-forming galaxies and push the 1.2 mm number counts down to 0.02 mJy partly thanks to gravitational lensing. Similarly, Zavala *et al.* [182] used over 130 individual ALMA continuum pointings at 3 mm (Band 3) towards three extragalactic legacy fields, achieving an effective survey area of 200 arcmin$^2$; their derived 3 mm number counts imply that the contribution of dusty star-forming galaxies to the cosmic star formation rate density at $z > 4$ is non-negligible. More recently, Liu *et al.* [346] presented the automated mining of the ALMA archive in the COSMOS field (A$^3$COSMOS), which

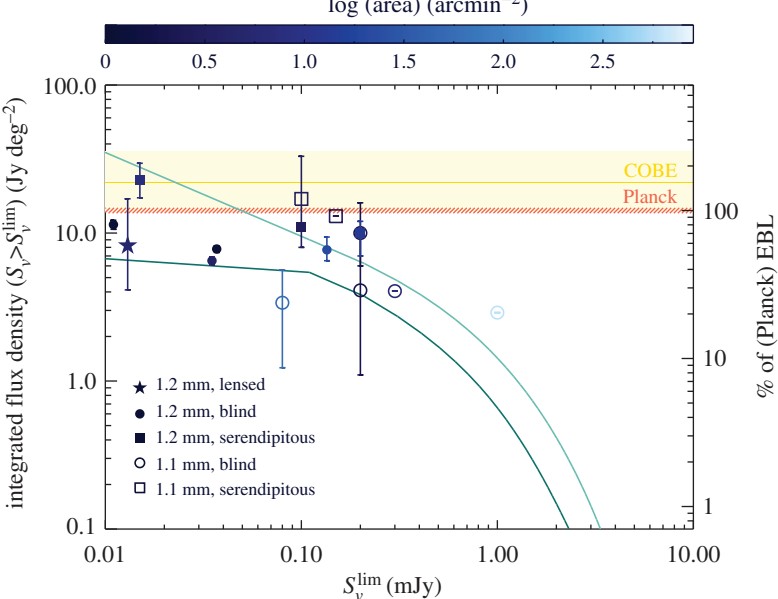

**Figure 26.** Integrated flux density of sources above various flux density limits $S_\nu^{lim}$, derived by recent ALMA surveys at 1.1 mm (open symbols) and 1.2 mm (filled symbols). The circles show the results from blind field surveys [62,64,66,69,617,619,620], and the squares are the results from serendipitous detections in fields around targeted sources [67,70,621,622] and in ALMA calibration fields [68]. The star shows the results from the ALMA survey of the *Hubble* Frontier Fields, i.e. using gravitational lensing [71]. Each point is colour-coded according to the survey area. For reference, the horizontal regions show the range of values of the EBL at 1.2 mm from *COBE* (in yellow), and *Planck* (in red). We compute the fraction of EBL recovered by the different surveys (right-hand *y*-axis) using the *Planck* value of $14.2 \pm 0.6$ Jy deg$^{-2}$ [619]. The light teal line shows the integration of the best-fit Schechter function to the number counts from these surveys obtained by Hatsukade *et al.* [66]; the dark teal line shows integration of the triple power-law fit to the number counts by González-López *et al.* [62].

includes a number of tools to automatically and continuously mine the science archive for continuum imaging observations and perform automated source extraction and counterpart association. Using these tools, they obtain approximately 1000 (sub-)mm detections from over 1500 individual ALMA pointings in the COSMOS field. Given the wealth of available multi-wavelength information in the COSMOS field, they obtain the redshifts and SEDs of the majority of these sources from matching with multi-wavelength counterparts, and they use this large sample to study the evolution of the stellar and gas content of galaxies with cosmic time (§3.1.2, [346,419]).

The clear advantage of this approach is that blind deep fields over large areas can be obtained from publicly available data, enhancing the scientific benefit of already-executed ALMA observations. These studies achieve effective areas orders-of-magnitude larger than those achieved by contiguous fields of similar depths that represent very significant observatory time investments. However, it must be noted that combining observations of various depths taken in a variety of configurations to study, e.g. number counts, where quantifying completeness is important, is non-trivial: the total survey area depends on the RMS achieved, the synthesized beam (i.e. the surface brightness sensitivity), and the primary beam attenuation. In addition, the selection function of PI-led programmes is difficult to quantify (see discussion in Liu *et al.* [346]).

## 4.2. Faint (sub-)millimetre sources and their contribution to the extragalactic background light

One of the many goals of the ALMA deep fields described above is to characterize the sources that make up the (sub-)mm extragalactic background light (EBL), measured by the *Cosmic Background Explorer (COBE)* satellite [86,87], and more recently by the *Planck* satellite [613]. This involves detecting the faint sources, i.e. the 'normal' star-forming galaxies that have lower infrared luminosities than the bright SMGs. These faint sources make up the bulk of the cosmic star formation rate density and therefore are thought to be more representative of star-forming galaxies at high-redshifts. Even before ALMA, various studies found that bright SMGs only contribute a relatively small fraction to the total EBL (e.g. [139,151,614,615]). Specifically, SMGs brighter than approximately 1 mJy constitute ≲ 20% of

the (sub-)millimetre EBL (e.g. [209,616–618], figure 26). However, single-dish studies were limited in sensitivity and also potentially biased due to source blending (e.g. [623]). ALMA surveys can capitalize on the unprecedented sensitivity and spatial resolution of ALMA to detect and resolve dust emission in faint dusty star-forming galaxies directly. To push down even more in luminosity, studies are taking advantage of the rich ancillary data available in those fields to obtain the (sub-)mm flux densities of samples of galaxies stacked in, e.g. optical colour, stellar mass and SFR (e.g. [599,619,624]).

Current surveys are deep enough to resolve most of—if not all—the sources of extragalactic background light at approximately 1 mm. The exact fraction of the EBL recovered in various studies depends on the assumed value for the total EBL at approximately 1 mm, which can vary widely (e.g. discussions in [62,619,621]). For example, at 1.2 mm, Fujimoto et al. [70] obtain a total EBL value of $22^{+14}_{-8}$ Jy deg$^2$ from the fit to COBE observations by Fixsen et al. [86], while Aravena et al. [619] adopt an EBL of $14.2 \pm 0.6$ Jy deg$^2$ from the recent Planck observations [613], which they argue are more accurate because the COBE spectrum is highly uncertain at frequencies below 350 GHz due to Galactic contamination. Cosmic variance due to the small area of the ALMA deep observations so far (table 4) is also a source of uncertainty (e.g. [625]). In figure 26, we compile the results on the integrated flux densities at 1.1–1.2 mm as a function of limiting flux density for all of the ALMA deep field studies so far. We also include the results from serendipitous detections of faint sources obtained by searching for sources in the fields around main targets in archival ALMA data [67,70,621,622] and in the calibration fields [68]. Figure 26 shows that there is still quite a significant spread in total integrated flux density obtained by different surveys at similar flux density limits (possibly due at least to some extent to cosmic variance, given how small the deep fields are, and also to the fact that some studies are based on lensing, which could carry some uncertainties); this is where calibration fields have an advantage, with a large combined area spread out in random locations on the sky, along with thousands of hours of total integration time. The uncertainty in the real value of the EBL adds further to uncertainties in determining how deep ALMA surveys have to go to resolve all the sources of the EBL. Furthermore, at the faint end, a significant number of recent measurements rely on serendipitously detected sources in ALMA fields targeting different sources, which could be biased if there is significant clustering around the main targets. At $S_\nu \lesssim 0.03$ mJy, it is interesting to note the difference between the serendipitous archival results of Fujimoto et al. [70] and the ASPECS results of Aravena et al. [619] and González-López et al. [62], which could be due to such overdensities in the archival fields (see also the Frontier Field results of Muñoz Arancibia et al. [71], though their errors are larger). Given the small area of the ASPECS Pilot field, differences could have been attributable to cosmic variance. However, the discrepancy remains when using the five times larger area of the ASPECS Large Programme field; González-López et al. [62] argue (based on modelling by Popping et al. [75]), that cosmic variance alone is not enough to reconcile their results with previous studies.

If we extrapolate the integral of the Schechter-function fit to a compilation of 1.2 mm number counts by Hatsukade et al. [66], which include most of the studies plotted in figure 26, the EBL measured by Planck is fully recovered by going down to approximately 0.04 mJy. However, the faint end number counts are mainly driven by the relatively steep faint-end number counts from the serendipitous-detection studies of Fujimoto et al. [70] and Carniani et al. [621], and in contrast with the faint-end stacking results from Aravena et al. [619] (note also that Carniani et al. [621] obtain a shallower slope at the faint end). The integral of the triple power law of González-López et al. [62] (dark teal line) accounts for a much flatter low faint-end of the number counts, and produces a different result: that even down to 0.01 mJy, the total EBL is not yet fully recovered. More deep blank fields over larger areas will be needed to constrain the faint end of the number counts more robustly and less dependently of clustering and cosmic variance (the ongoing 'ALMA Lensing Cluster Survey' Large Programme (PI: Kohno) will address this). However, the uncertainty on the exact value of the total EBL at approximately 1 mm remains, as discussed above.

Nevertheless, it is clear that with ALMA we are very close to resolving most—if not all—of the sources that contribute to the EBL at current depths; going much deeper is not likely to yield a new population of significant sources. The next step is then to look at the properties of the individual galaxies, including their stellar masses and redshifts, by matching with their optical/near-infrared counterparts. Using the deep ASPECS Large Programme observations, González-López et al. [62] find that the 1 mm number counts are dominated by sources at $1 < z < 3$. They find that there is a continuum in galaxy properties as we move in flux density: the bright number counts ($S_{1.2mm} \gtrsim 1$ mJy) are dominated by massive ($M_* \gtrsim 10^{11} M_\odot$), highly star-forming (SFR $\sim 100$–$1000 M_\odot$yr$^{-1}$), dusty sources ($M_{dust} \gtrsim 10^9 M_\odot$); at intermediate flux densities $0.1 \lesssim S_{1.2\,mm} \lesssim 1$ mJy, we find galaxies with typical $M_* \sim 10^{10}-10^{11} M_\odot$, SFR $\sim 10-100 M_\odot$yr$^{-1}$, and $M_{dust} \sim 10^8-10^9 M_\odot$; at the faintest flux density levels probed by ALMA ($S_{1.2\,mm} < 0.1$ mJy), the number counts are dominated by the very low-mass ($M_* \lesssim 10^9 M_\odot$), low-star

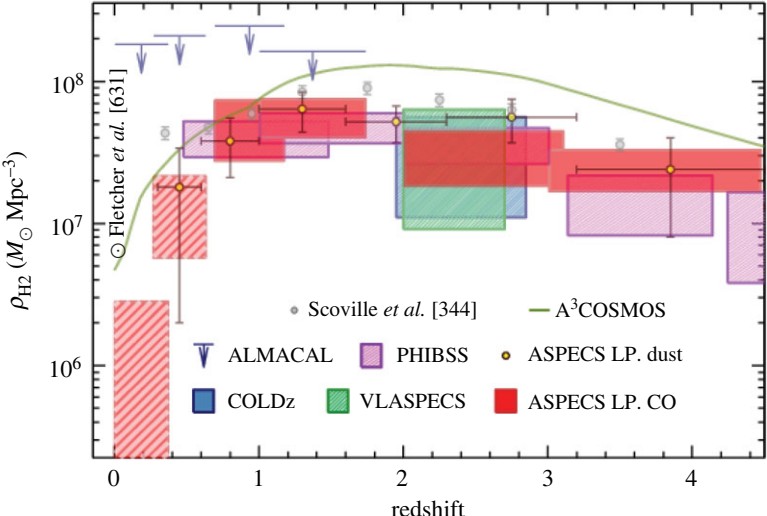

**Figure 27.** The redshift evolution of the cosmic molecular gas mass density, $\rho_{H2}(z)$, obtained by Decarli et al. [629], using CO detections from the ASPECS Large Programme 1 and 3 mm scans (red shaded regions; [78]). Results from another molecular line scan, the VLA COLDz survey [630], are shown in blue. The yellow circles show the evolution of gas density also from the ASPECS-LP, but where the gas masses are inferred from dust continuum observations [418]. Also for comparison, constraints from other targeted and archival surveys are shown: VLASPECS (a VLA follow-up of ASPECS sources; [631]), PHIBSS (targeted CO and dust continuum in massive galaxies; [345]), ALMACAL (via CO absorption towards distant quasars; [304]), A$^3$COSMOS (archival observations; [346]) and the Scoville et al. [344] survey of dust continuum in galaxies. The local measurement from the IRAM xCOLD GAS survey [632] is shown as a black circle. Figure courtesy of Decarli et al. [629].

formation rate (SFR < 10 $M_\odot$ yr$^{-1}$), least dusty ($M_{dust} \sim 10^7$–$10^8$ $M_\odot$) sources, that require the deepest data from *HST/Spitzer* to be detected in the rest-frame UV/optical.

## 4.3. Blind CO detections and the evolution of the cosmic molecular gas density

With its sensitivity and bandwidth, ALMA is a prime instrument to perform deep spectroscopic surveys of CO. However, even with its approximately 8 GHz bandwidth, this is a time-intensive task, and so far only one survey, ASPECS [216], has carried out a blind spectroscopic search for CO across the full range of frequencies (and thus redshifts) allowed by ALMA. ASPECS used the full combined bandwidth of Bands 3 and 6 to search for (low- and mid-*J*) CO emission from sources at redshifts $0 < z < 5$ in a 1 arcmin$^2$ region in the *Hubble* Ultra Deep Field (Pilot survey), and later on an area five times larger (Large Programme). They reached CO line sensitivities of approximately $L'_{CO} \sim 2 \times 10^9$ K km s$^{-1}$ pc$^2$.

The pilot ASPECS survey yielded a total of 11 blind CO detections in the 1 and 3 mm bands [215], i.e. sources that were found purely from searching for lines in the data cube without *a priori* knowledge from other observations. The large programme robustly detected 16 CO sources so far in the 3 mm map ([78,598,626], 1 mm dataset is ongoing at the time of writing). The blind CO detections show a remarkable diversity in their properties [626–628]. These sources are found at $z \sim 1$–4 and have a broad range of stellar masses ($M_* \sim 0.03 - 4 \times 10^{11}$ $M_\odot$), star formation rates (SFR approximately $0 - 300$ $M_\odot$ yr$^{-1}$), and molecular gas masses ($M_{H_2} \sim 5 \times 10^9 - 1.1 \times 10^{11}$ $M_\odot$). Aravena et al. [626] show that they follow the scaling relations between gas content and star formation rate/stellar mass found for stellar mass/SFR-selected samples ([345], §3.1). The CO-detected sources extend the previously established relations in the gas depletion timescales and gas fractions probed, and in some cases significant outliers are found. In particular, the ASPECS blind scan is capable of detecting galaxies below the main sequence that have significant molecular gas reservoirs.

The blind CO detections obtained by ASPECS ([78,215], Decarli et al. [629]) allow for the characterization of CO luminosity functions out to $z \sim 4$. By integrating the luminosity functions (and assuming CO excitation corrections and a CO-to-H$_2$ conversion factor), one can trace the evolution of the cosmic density of molecular gas, $\rho_{H2}(z)$, in an unbiased way out to those redshifts. Overall, the ASPECS survey shows that the CO luminosity functions evolve significantly through cosmic time [78]. As shown in figure 27, the molecular gas density of the Universe peaked at $z \sim 1$–3 (coincident with

the peak of cosmic SFR density; e.g. [2]), when it was between three and seven times higher (depending on the adopted CO-to-molecular gas conversion factor). A complementary effort using over 300 h of VLA time to measure low-$J$ CO transitions over a 60 arcmin$^2$ area by the COLDz team [630] finds a similar result. Figure 27 also shows the evolution of $\rho_{H2}(z)$ obtained using the targeted molecular gas and dust continuum surveys discussed §3.1.2 (PHIBSS, A$^3$COSMOS, and the survey by Scoville *et al.* [344]). All these surveys show a similar qualitative redshift evolution, though the A$^3$COSMOS survey seems to obtain systematically higher gas masses. The differences are probably due to different methodological approaches, e.g. different stellar mass regimes used to integrate the CO/molecular gas functions, different assumptions about the shape and evolution of the star-forming main sequence, and/or different assumptions regarding the CO conversion factor (R. Decarli 2020, private communication). A quantitative analysis of these differences would be extremely beneficial to the field but is outside the scope of this review; for now we focus on the similar qualitative evolution of $\rho_{H2}(z)$ obtained by the different studies. The quantitatively coincident evolution of SFR density and molecular gas density seems to indicate that the SFR density of the Universe since at least $z \sim 3$ is mostly dominated by the available molecular gas supply to form new stars, rather than an evolution of the star formation efficiency in galaxies. Despite consistency with this overall picture, detailed comparisons with cosmological galaxy formation models [183] show that those models struggle to reproduce the redshift evolution of molecular gas density, and the number of gas-rich galaxies. Popping *et al.* [183] show that the tensions between models and observations can be alleviated to some extent by changing the assumed CO excitation and conversion factor, and that they cannot be fully explained by cosmic variance. They argue that the current underestimation of the molecular gas reservoirs in $z > 1$ galaxies in theoretical models—as compared with the ASPECS measurements— could be linked to broader problems in modelling the gas accretion and feedback in galaxies that also make matching the SFRs challenging. That study demonstrates clearly that improved empirical constraints on the full baryonic content of galaxies and their star formation rates, including molecular gas reservoirs, are crucial to test current models, in particular their sub-grid physics.

## 4.4. Continuum and [CII] line searches at high redshift

Deep blind surveys have also attempted to detect galaxies well into the epoch of reionization via their [CII] and dust continuum emission (e.g. [633–635]). The deepest of these surveys so far was the ASPECS Pilot survey [619]. The frequency range of the ASPECS ALMA Band 6 covers [CII] emission at $6 < z < 8$. Aravena *et al.* [636] find 14 [CII] line candidates at greater than $4.5\sigma$ in the ASPECS Pilot region of the UDF, two of which are blind detections, i.e. with no counterparts in the optical/near-infrared *HST* imaging. None of those line candidates are detected in the dust continuum, consistent with the study of Lyman-break galaxies of Capak *et al.* [445]. These observations are a first step toward determining the evolution of the [CII] luminosity functions at high-$z$, as well as testing how local relations between [CII] and infrared luminosity/star formation rate evolve into the epoch of reionization. Using blind detections rather than following up known samples (such as LBGs) helps provide an unbiased census of the [CII] emission at high-$z$. These blind detections suggest that the typical [CII]-to-IR luminosity ratio might be much lower at $6 < z < 8$ than in the local Universe, somewhat in tension with the results of Capak *et al.* [445] discussed in §3.4. However, this particular study acknowledged a relatively high rate of potentially spurious sources (60%; see also [637] for a report of spurious [CII] detections in SSA22), and deeper observations over larger areas are needed to improve on the statistics of the luminosity functions and SFR calibrations. Recent work on the full ASPECS [638] shows that none of the previous [CII] detections of Aravena *et al.* [636] are recovered, highlighting the very high rate of spurious sources. From the theoretical side, models that predict the number counts of [CII] sources (e.g. [576,639]) predicted ASPECS should essentially see less than one galaxy in [CII], consistent with the latest results.

Targeted studies such as the ALMA Large Programme to INvestigate [CII] at Early times (ALPINE) survey (e.g. [584,640–644]) are a more promising way to detect [CII] emission at high-redshift. ALPINE targeted a sample of 118 spectroscopically confirmed star-forming galaxies at $4.4 < z < 5.9$ with a typical beam size of 0.7 arcsec ($\simeq 6$ kpc). They detect 64% of their targets, and show that their detections are diverse in terms of morphology and kinematics, including rotating discs and mergers. ALPINE is only starting to produce results at the time of writing, but it is already demonstrating how ALMA can take advantage of the brightness of [CII] in star-forming galaxies to trace their detailed structure and kinematics [642], and even detect gas outflows and metal enrichment of the circumgalactic medium in the early Universe [584].

# 5. Concluding remarks

In this review, we have described some of the ways in which ALMA is revolutionizing our understanding of high-redshift star formation and galaxy evolution in general. ALMA's unprecedented sensitivity allows us to extend studies beyond the bright, starbursting sources, and toward the more 'normal' star-forming galaxies that contribute the most to both the far-infrared/millimetre extragalactic background light and the cosmic star formation rate over the history of the Universe. The exquisite spatial resolution of ALMA allows us to disentangle different galaxy components and, crucially, observe the processes that shape galaxy evolution down to the relevant physical scales. The frequency coverage has opened up a new realm of line tracers of the multi-phase interstellar medium, allowing studies of the detailed dynamics, chemical composition and physical conditions from dense molecular clouds to the diffuse ionized medium, and in sources from the peak of cosmic star formation rate at $z \simeq 2$ to the epoch of reionization. This review has attempted to capture the most exciting science results enabled by these new capabilities in the less than 10 years since the start of operations; however, many open questions still remain, and some of the new observations have uncovered a new set of puzzles to solve in both the short and longer term. Here we highlight a few.

While ALMA has allowed large samples of single-dish-selected SMGs to be reliably identified for the first time—enabling a plethora of studies on their physical properties—most of the samples studied still lack complete redshift distributions. Even with reliably identified counterparts, optical/near-infrared spectroscopic redshifts are challenging (or impossible) to obtain due to the high levels of dust obscuration in these galaxies. ALMA should be the redshift machine for such sources, where a frequency scan would have the added benefit of yielding multiple lines with which to study the physical conditions and chemistry of the interstellar medium. However, such a survey requires multiple frequency settings across multiple bands, and has so far only been attempted in bright, strongly lensed sources [163,191,237]. A complete, unbiased redshift distribution of SMGs (including those now resolved into multiple distinct sources) is necessary not only for distinguishing between different theoretical models for SMG formation, but also for determining the prevalence of massive, dusty galaxies at early cosmic epochs. This has to be combined with more systematic searches for dusty high-redshift sources using, e.g. surveys at longer wavelengths (e.g. [540]) in order to get the true infrared luminosity function and dust corrections to the high-redshift star formation rate density of the Universe. Such studies will be complemented by next-generation facilities like the SPace Infrared-telescope for Cosmology and Astrophysics (SPICA; [645]) satellite, which will not only detect dust-obscured galaxies and AGN (which can then be followed up efficiently with ALMA) out to high redshifts through wide-area photometric surveys, but will help uniquely characterize the composition of the dust in these galaxies through IR spectroscopy.

Concurrently with progress in identifying the brightest submillimetre sources, the dust content of the very earliest galaxies—close to and even within the epoch of reionization—is now being measured for the first time thanks to input from optically selected samples and the unique sensitivity of ALMA. This is partly thanks to an explosion in studies of the [CII] line, which has delivered on its promise of being a work-horse line in the era of ALMA. A puzzling picture has emerged wherein some low-mass galaxies at $z > 4$ seem to have lower dust contents and potentially different dust attenuation curves than what is usually assumed. However, at the same time, vast amounts of dust (exceeding $10^7 \, M_\odot$) have been detected in other high-redshift low-mass star-forming galaxies and in quasar hosts (e.g. [550,551,553]). These observations challenge our usual assumptions about dust at high redshifts, and they highlight the need for multi-frequency ALMA observations in order to get the most accurate dust temperatures, luminosities and masses. At the same time, *JWST* observations will be crucial to measure metal enrichments, star formation histories, and dust attenuations for these early galaxies, all of which are crucial to piece together the picture of how dust is forming and evolving in these systems. Observational progress in understanding the emergence and evolution of cosmic dust in the earliest galaxies will have to go hand-in-hand with advances in theoretical modelling. A promising avenue is the detailed modelling of the stellar and gas/dust distribution in zoom-in hydrodynamical simulations, including radiative transfer, that can inform on the way observables are affected by different dust intrinsic properties and spatial distributions (e.g. [167,536,539,646]); see also the review by Dayal & Ferrara [83] and references therein. Additionally, models tracing the formation and growth of dust grains from supernova and evolved star envelopes to the dense ISM in the cosmological context are becoming ever more sophisticated (e.g. [208,556,647–650]).

The unmatched spatial resolution achievable with ALMA has allowed studies to resolve the dust continuum and gas distributions in previously unresolved high-redshift galaxies on kpc and even sub-kpc scales, revealing the gas/dust continuum sizes, profiles and sometimes even morphologies

of many high-redshift sources for the first time. While the highest resolution imaging possible with ALMA will necessarily remain limited to select sources due to the inherent trade-off between spatial resolution and surface brightness sensitivity, a handful of strongly lensed sources have even been mapped on scales of approximately 10 s of pc (though we note that such cases are still rare). In some cases, the dust continuum emission seen with ALMA is uncorrelated with the existing stellar populations revealed by deep *HST* images, and there are often spatial offsets between the dust emission and the (rest-frame ultraviolet and optical) stellar emission probed by *HST*. In these cases, we still need to understand how the observed anti-correlations/offsets affect the interpretation and globally derived properties for the high-redshift sources. The highest fidelity sub-kpc mapping of unlensed sources has revealed robust substructure in some galaxies—including structures resembling bars, rings, and spiral arms (e.g. [439])—but currently only for a handful of the brightest SMGs. The evidence for such galactic structures still needs to be assessed in the broader SMG population, as well as the theoretical implications for the mechanisms governing their evolution. Meanwhile, the lack of evidence for such structure in IR-fainter sources needs to be investigated further. Ultimately, a complete understanding of the structure of high-redshift, dusty sources will also require kinematic tracers (using, e.g. [CII] or CO lines) observed at high spatial resolution, and, in the near future, deep high-resolution near- and mid-infrared imaging with the *JWST*, which will pierce through the dust to reveal the underlying stellar populations.

The relationship between star formation and stellar mass (the so-called 'star-forming main sequence' of galaxies) has become a fundamental relation with which astronomers try to understand galactic evolution. Despite this fact, its nature and driving mechanisms are still poorly understood, especially at high-redshifts. ALMA has enabled progress in this area by beating the confusion limit of the *Herschel Space Telescope*, easily detecting the dust emission from non-submillimetre-selected 'main sequence' star-forming galaxies at $z > 3$. This feat has effectively forced the submillimetre community and the general high-redshift community to merge, leading to some confusion in the terminology. More critically, without multi-frequency observations—including high-frequency ALMA bands—there are still large (order of magnitude) uncertainties in the total infrared luminosity (and hence star formation rates) inferred from low-frequency, single-band ALMA measurements. Future multi-band observations with ALMA could address this problem, while rest-frame near-infrared imaging with the *JWST* will further help reduce uncertainties on stellar mass estimates of dusty star-forming galaxies at high redshift, particularly for the classically studied SMGs and other ALMA-detected sources that lack an optical counterpart (*HST*-dark). It is crucial to note, however, that placing galaxies accurately in the star formation–stellar mass parameter space is not sufficient for understanding their nature. ALMA has access to additional information on their molecular gas content and neutral gas kinematics, which can be complemented with integral field unit (IFU) observations of their ionized gas (currently available with KMOS and MUSE, e.g. [627], and in the future with extremely large telescopes (ELTs)), and/or high-resolution imaging with the *JWST*. Synergies with such facilities will be crucial to disentangle the detailed physical processes shaping the so-called main sequence and other scaling relations.

ALMA has revealed the molecular gas reservoirs of unprecedentedly large samples of galaxies at high redshifts, including blindly selected samples from deep fields (e.g. ASPECS; [216]). This is crucial to understand how star formation is fuelled in galaxies, and what determines the star formation history and efficiency in galaxies. However, there are still open questions about what is the best method to measure the molecular gas mass in early galaxies. CO measurements with ALMA at high-redshifts rely heavily on excitation corrections and the CO-to-$H_2$ conversion factor $\alpha_{CO}$, both of which may carry significant systematic uncertainties. One way to overcome the excitation corrections is to target the ground-state CO(1–0) line with the VLA (e.g. COLDz; [630]), or in the future with the planned ALMA low-frequency bands (Band 1 at 35–50 GHz, currently being built, and Band 2 at 65–90 GHz), the proposed next generation VLA (ngVLA), and the Square Kilometre Array (SKA); however, one still must contend with the uncertainty in $\alpha_{CO}$ (as well as the increasing effect of the cosmic microwave background at the highest redshifts). This is where better calibrations of $\alpha_{CO}$ from local studies for a range of metallicities and star-forming environments could be helpful (e.g. [457,651]), in tandem with theoretical modelling (e.g. [484]). Alternatively, more work needs to be done to investigate and calibrate other proposed molecular gas tracers at high-redshift, such as dust continuum, [CI], or even [CII], for a wide range of galaxy properties (e.g. [380,652]).

In addition to the (now) 'typical' molecular and atomic gas tracers studied in high-redshift sources (e.g. CO, [CII]), ALMA's sensitivity and frequency coverage have allowed novel new studies of additional emission/absorption lines in the (sub-)millimetre regime, including new results on (typically higher-*J*) dense gas tracers, CO isotopologues, a variety of atomic fine structure lines, and

molecular absorption lines. Such studies have the potential to open up an entirely new window into the ISM properties of distant star-forming galaxies. However, aside from studies targeting [CII] and other bright fine structure lines—which have exploded, particularly at the highest redshifts—many of these lines are still too faint to detect/resolve in all but the brightest/most strongly lensed sources. This is another area where future radio/millimetre facilities (ALMA Bands 1 and 2, the ngVLA and the SKA) will play a fundamental role through the detection (and imaging) of low-$J$ transitions of dense gas tracers and potentially even radio recombination lines at high-redshift (e.g. [486,653]).

The current capabilities of ALMA have not yet been exploited to their full potential to address the issues described above and other open questions. There are bound to be more exciting results in the coming years. Looking ahead, further improvements in ALMA's capabilities are being proposed as part of the ALMA Development Roadmap [654]. The 'Origins of Galaxies' is one of the three fundamental science drivers for ALMA in the next decades, namely to 'trace the cosmic evolution of key elements from the first galaxies ($z > 10$) through the peak of star formation ($z = 2$–4) by detecting their cooling lines, both atomic ([CII] and [OIII]) and molecular (CO), and dust continuum, at a rate of 1–2 galaxies per hour'. With this in mind, current upgrade priorities are to broaden the receiver instantaneous bandwidth, and to upgrade the associated electronics and correlator. Such an improvement—as recently highlighted by the high-performance wideband correlators deployed on NOEMA (PolyFiX) and the SMA (SWARM; [655])—would enable faster spectral scans (including redshift surveys), and deeper and wider continuum surveys. This would allow for large statistical samples of galaxies at high redshifts, sampling the parameter space down to low luminosities and high-redshifts, and for more efficient spectroscopic studies. These capabilities, combined with the future facilities of the 2020s and 2030s such as the *JWST* and ELTs, hold exciting promise for the future of multi-wavelength studies of galaxy evolution out to the earliest cosmic epochs.

Data accessibility. This article does not contain any additional data.

Authors' contributions. Both authors contributed equally to the structuring and writing of this review.

Competing interests. We declare we have no competing interests.

Funding. E.d.C. gratefully acknowledges the Australian Research Council as the recipient of a Future Fellowship (project no. FT150100079). J.A.H. gratefully acknowledges support of the VIDI research program with project no. 639.042.611, which is (partly) financed by the Netherlands Organisation for Scientific Research (NWO).

Acknowledgements. We are grateful to all our colleagues who took the time to read a preliminary draft of this review and send us their comments, including Fabian Walter, Rob Ivison, Nick Scoville, Gergo Popping, Eva Schinnerer, Catherine Vlahakis, Gabriela Calistro-Rivera, Matus Rybak, Paul van der Werf, Renske Smit, Ian Smail, Justin Spilker and two anonymous referees. We also thank Jorge Gonzalez-Lopez for sharing his compilation of 1 mm number counts, Claudia Lagos and Gergo Popping for sending us their model predictions on the redshift distribution of SMGs, and Cassie Reuter and Joaquin Vieira for sharing their unpublished figure containing the SPT spectra. Finally, we would like thank Rosa and Manuel da Cunha for hosting our Portuguese writing retreat and for the endless supply of delicious food and Port wine. *Muito obrigada a Rosa e Manuel da Cunha por hospedarem o nosso retiro de escrita e pela deliciosa comida e vinho do Porto.*

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
