## [Reviewer comments · Royal Society Open Science]

Review History

RSOS-200556.R0 (Original submission)

Review form: Reviewer 1

Is the manuscript scientifically sound in its present form?

Yes

Are the interpretations and conclusions justified by the results?

Yes

Is the language acceptable?

Yes

Do you have any ethical concerns with this paper?

No

Have you any concerns about statistical analyses in this paper?

No

Recommendation?

Accept with minor revision (please list in comments)

Comments to the Author(s)

Comments to the authors

This review article is a remarkable work, well written and quite complete about high-z star forming galaxies, ALMA has revolutionize the domain, and it is timely to make the census of all new results now. The authors have explained in detail what was the state of the art before ALMA, and described all the progress due to ALMA, keeping also space to SCUBA2 and NOEMA.

The text is well structured, (continuum dust studies, line molecular gas, etc..) and the Figures very well done and relevant.

I have only very minor comments.

Figure 2 computes the sensitivity for a PWV of 3mm, whatever the sites. It might be good to add which percentage of the time this is obtained in each site. Indeed, the ALMA altitude of 5500m and deserty site makes these conditions very common, and even much lower PWV are possible, while they are quite rare in the other sites. This will explain better the amplitude of the ALA revolution.

p.18: comparison between local ULIRGs and SMG, the latter being bluer.

"This difference suggests a lower average dust attenuation, which could be due to the fact that high-redshift SMGs may be more extended than local ULIRGs.". I guess this is not the obvious conclusion, the most obvious being that at high-z galaxies have lower metallicity, and therefore lower dust attenuation, for the same radial extension. Independent observations show that galaxies at high-z are more compact, more clumpy, even if more gaseous. So the stellar component could be bluer, may be also because the stars are in average younger.

In Section 2.3.3 when the GMC and star formation scaling laws are described, and only a few clumps are derived from lensing galaxies, you should cite the recent work on the Snake <https://ui.adsabs.harvard.edu/abs/2019NatAs...3.1115D/abstract> where a lot more clumps have been detected

In Section 3.2.1, when discussing Σ_{SFR} going higher than the Eddington limit, you should caution that the FIR could be excited by a central AGN, and the true SFR could be over-estimated. The fraction of AGN-heated FIR is increasing with mass and L_{FIR} .

Note that in Section 3.2.8, the discussion on the existence or not of the clump could be enlightened by the example of the Snake (Dessauges-Zavadsky et al 2019), that was mentioned earlier.

Section 3.5: influence of the CMB. Indeed, the contrast of the dust/gas emission in the outer parts of a galaxy is lower, as mention in Zhang et al (2016). However, it is true at any redshift, even at $z=0$

"For example, in galaxies where the dust temperature decreases with radius, the cool dust in the outer regions might not be visible against the CMB background, and therefore the size of the dust-emitting region might be underestimated for galaxies at high redshifts." >> at any redshift in fact.

Very complete and impressive work!

The referee

Review form: Reviewer 2

Is the manuscript scientifically sound in its present form?

Yes

Are the interpretations and conclusions justified by the results?

Yes

Is the language acceptable?

Yes

Do you have any ethical concerns with this paper?

No

Have you any concerns about statistical analyses in this paper?

No

Recommendation?

Accept with minor revision (please list in comments)

Comments to the Author(s)

Overall, I found the manuscript a useful and comprehensive review.

I would suggest perhaps that the conclusions section is made more “punchy”, and that the authors consider a list format for some of the most crucial results there, and perhaps a more highlighted set of future recommendations for key work?

A couple of future opportunities seem to be potentially missing,

One is the capacity of very-wide-band correlations, as being highlighted by SmA and NOEMA, and while could update ALMA.

Two, the SkA. While it is too low n frequency for molecular one work, continuum sensitivity, and potential resolution is very impressive, and continuum and radio-recombination lines might make a substantial contribution here. The author’s net rest in CMB effects could provide a route in here?

The most substantial suggestion I have is to include more information about ALMA’s Surface brightness sensitivity being inevitably limited at finest resolution, which I thin should be noted, as integration times for interesting observations can become “challenging”. For non-experts, I suspect this issue might be under appreciated, and the minute-long integrations that can produce impressive results seem strange when compared with the hours required to resolve CO lines.

I wonder whether there’s a way to highlight this near Fig. 2?

Minor comments:

Very end of Section 2.2.2 - isn’t the multiplicity changing with depth already largely clear? It ught help to be more specific about the future approach required to getting samples suitable to understand this better at this point in the manuscript, or to highlight the relevant discussion at the conclusion.

Section 2.2.5. Don’t the substantial SPT samples indicate that a high-flux cutoff must be modest? With decent lending models, they do seem to have some sources substantially brighter than 10mJy? This seems to be revisited in Section 2.3.1, but it might be a good idea to link it here, and discuss this factor.

Section 2.2.6. Median redshift is perhaps not all you’d like to know - there’s also issue of the shape of the whole distribution, and perhaps especially of the high-redshift tail, since detecting the restframe FIR vs UV continues to get easier at higher z? The absence of a long tail I suggest gives a general hint that the first light might be a job for JWST, or points beyond.

Section 3.2.7. Does SKA not have potential to compare with next-generation VLA here? Sensitive low-frequency continuum and radio recombination lines seem to have potential value. If the authors disagree, it would still be worth a gentle note about this large facility that's on its way.

Boundary of P46/47. Can substantial AGN contamination not be ruled out already by spatial information? The AGN emission would have to lurk unresolved? The lenses samples having complex, but enhanced resolution in the source plane might help to address this, even if the ratios are difficult to pin down? Multiplicity itself also suggests AGN can't dominate?

On P49. Young dust with low-density/colloidal/dendritic structure can provide more opacity for lower mass. Early dust may indeed be different, and something that is discussed a little later, but perhaps should appear earlier as a caveat, which could be a positive thing for future investigations, in [CII] vs continuum, Nd perhaps JWST mid-IR spectra?

Overall, I thought it was a comprehensive and informative review. The biggest concern I had was a warning about surface brightness sensitivity at the highest resolution. At present I suspect the presentation doesn't reflect this challenge to ALMA's rate of discovery; without being critical, perhaps more managing expectations.

Decision letter (RSOS-200556.R0)

Dear Dr da Cunha

On behalf of the Editors, I am pleased to inform you that your Manuscript RSOS-200556 entitled "High-redshift star formation in the ALMA era" has been accepted for publication in Royal Society Open Science subject to minor revision in accordance with the referee suggestions. Please find the referees' comments at the end of this email.

The reviewers and handling editors have recommended publication, but also suggest some minor revisions to your manuscript. Therefore, I invite you to respond to the comments and revise your manuscript.

- Ethics statement

- Data accessibility

If you wish to submit your supporting data or code to Dryad (<http://datadryad.org/>), or modify your current submission to dryad, please use the following link:
<http://datadryad.org/submit?journalID=RSOS&manu=RSOS-200556>

- **Competing interests**

- **Authors' contributions**

- **Acknowledgements**

- **Funding statement**

Because the schedule for publication is very tight, it is a condition of publication that you submit the revised version of your manuscript before 16-May-2020. Please note that the revision deadline will expire at 00.00am on this date. If you do not think you will be able to meet this date please let me know immediately.

If your manuscript is newly submitted and subsequently accepted for publication, you will be asked to pay the article processing charge, unless you request a waiver and this is approved by Royal Society Publishing. You can find out more about the charges at <https://royalsocietypublishing.org/rsos/charges>. Should you have any queries, please contact openscience@royalsociety.org.

on behalf of Professor Ian Smail (Associate Editor) and Rob Ivison (Subject Editor)
openscience@royalsociety.org

Associate Editor Comments to Author (Professor Ian Smail):

Associate Editor: 1

Comments to the Author:

Please take note of the comments from the two reviewers in terms of revisions to the manuscript.

Reviewer comments to Author:

Reviewer: 1

Comments to the Author(s)

Comments to the authors

This review article is a remarkable work, well written and quite complete about high-z star forming galaxies, ALMA has revolutionize the domain, and it is timely to make the census of all new results now. The authors have explained in detail what was the state of the art before ALMA, and described all the progress due to ALMA, keeping also space to SCUBA2 and NOEMA.

The text is well structured, (continuum dust studies, line molecular gas, etc..) and the Figures very well done and relevant.

I have only very minor comments.

Figure 2 computes the sensitivity for a PWV of 3mm, whatever the sites. It might be good to add which percentage of the time this is obtained in each site. Indeed, the ALMA altitude of 5500m and deserty site makes these conditions very common, and even much lower PWV are possible, while they are quite rare in the other sites. This will explain better the amplitude of the ALA revolution.

p.18: comparison between local ULIRGs and SMG, the latter being bluer.

"This difference suggests a lower average dust attenuation, which could be due to the fact that high-redshift SMGs may be more extended than local ULIRGs." I guess this is not the obvious conclusion, the most obvious being that at high-z galaxies have lower metallicity, and therefore lower dust attenuation, for the same radial extension. Independent observations show that galaxies at high-z are more compact, more clumpy, even if more gaseous. So the stellar component could be bluer, may be also because the stars are in average younger.

In Section 2.3.3 when the GMC and star formation scaling laws are described, and only a few clumps are derived from lensing galaxies, you should cite the recent work on the Snake <https://ui.adsabs.harvard.edu/abs/2019NatAs...3.1115D/abstract> where a lot more clumps have been detected

In Section 3.2.1, when discussing Σ_{SFR} going higher than the Eddington limit, you should caution that the FIR could be excited by a central AGN, and the true SFR could be over-estimated. The fraction of AGN-heated FIR is increasing with mass and L_{FIR} .

Note that in Section 3.2.8, the discussion on the existence or not of the clump could be enlightened by the example of the Snake (Dessauges-Zavadsky et al 2019), that was mentioned earlier.

Section 3.5: influence of the CMB. Indeed, the contrast of the dust/gas emission in the outer parts of a galaxy is lower, as mention in Zhang et al (2016). However, it is true at any redshift, even at $z=0$

"For example, in galaxies where the dust temperature decreases with radius, the cool dust in the outer regions might not be visible against the CMB background, and therefore the size of the dust-emitting region might be underestimated for galaxies at high redshifts." >> at any redshift in fact.

Very complete and impressive work!

The referee

Reviewer: 2

Comments to the Author(s)

Overall, I found the manuscript a useful and comprehensive review.

I would suggest perhaps that the conclusions section is made more “punchy”, and that the authors consider a list format for some of the most crucial results there, and perhaps a more highlighted set of future recommendations for key work?

A couple of future opportunities seem to be potentially missing,

One is the capacity of very-wide-band correlations, as being highlighted by SmA and NOEMA, and while could update ALMA.

Two, the SkA. While it is too low n frequency for molecular one work, continuum sensitivity, and potential resolution is very impressive, and continuum and radio-recombination lines might make a substantial contribution here. The author’s net rest in CMB effects could provide a route in here?

The most substantial suggestion I have is to include more information about ALMA’s Surface brightness sensitivity being inevitably limited at finest resolution, which I think should be noted, as integration times for interesting observations can become “challenging”. For non-experts, I suspect this issue might be under appreciated, and the minute-long integrations that can produce impressive results seem strange when compared with the hours required to resolve CO lines.

I wonder whether there’s a way to highlight this near Fig. 2?

Minor comments:

Very end of Section 2.2.2 - isn’t the multiplicity changing with depth already largely clear? It ught help to be more specific about the future approach required to getting samples suitable to understand this better at this point in the manuscript, or to highlight the relevant discussion at the conclusion.

Section 2.2.5. Don’t the substantial SPT samples indicate that a high-flux cutoff must be modest? With decent lending models, they do seem to have some sources substantially brighter than 10mJy? This seems to be revisited in Section 2.3.1, but it might be a good idea to link it here, and discuss this factor.

Section 2.2.6. Median redshift is perhaps not all you’d like to know - there’s also issue of the shape of the whole distribution, and perhaps especially of the high-redshift tail, since detecting the restframe FIR vs UV continues to get easier at higher z? The absence of a long tail I suggest gives a general hint that the first light might be a job for JWST, or points beyond.

Section 3.2.7. Does SKA not have potential to compare with next-generation VLA here? Sensitive low-frequency continuum and radio recombination lines seem to have potential value. If the authors disagree, it would still be worth a gentle note about this large facility that’s on its way.

Boundary of P46/47. Can substantial AGN contamination not be ruled out already by spatial information? The AGN emission would have to lurk unresolved? The lenses samples having complex, but enhanced resolution in the source plane might help to address this, even if the ratios are difficult to pin down? Multiplicity itself also suggests AGN can’t dominate?

On P49. Young dust with low-density/colloidal/dendritic structure can provide more opacity for lower mass. Early dust may indeed be different, and something that is discussed a little later, but perhaps should appear earlier as a caveat, which could be a positive thing, for future investigations, in [CII] vs continuum, Nd perhaps JWST mid-IR spectra?

Overall, I thought it was a comprehensive and informative review. The biggest concern I had was a warning about surface brightness sensitivity at the highest resolution. At present I suspect the presentation doesn't reflect this challenge to ALMA's rate of discovery; without being critical, perhaps more managing expectations.

Author's Response to Decision Letter for (RSOS-200556.R0)

See Appendix A.

Decision letter (RSOS-200556.R1)

Dear Elisabete,

It is a pleasure to accept your manuscript entitled "High-redshift star formation in the ALMA era" in its current form for publication in Royal Society Open Science.

on behalf of Professor Ian Smail (Associate Editor) and Rob Ivison (Subject Editor)
openscience@royalsociety.org

Appendix A

Reply to the Referees

We would like to thank both referees for taking the time to read our manuscript and provide valuable feedback which has improved the review. We would also like to apologize for the delay in our response.

Reviewer 1:

This review article is a remarkable work, well written and quite complete about high-z star forming galaxies, ALMA has revolutionize the domain, and it is timely to make the census of all new results now. The authors have explained in detail what was the state of the art before ALMA, and described all the progress due to ALMA, keeping also space to SCUBA2 and NOEMA. The text is well structured, (continuum dust studies, line molecular gas, etc..) and the Figures very well done and relevant.

We thank referee #1 for the extremely positive reaction and specific comments, which we address below.

I have only very minor comments.

Figure 2 computes the sensitivity for a PWV of 3mm, whatever the sites. It might be good to add which percentage of the time this is obtained in each site. Indeed, the ALMA altitude of 5500m and desartic site makes these conditions very common, and even much lower PWV are possible, while they are quite rare in the other sites. This will explain better the amplitude of the ALA revolution.

We thank the referee for this good suggestion, and we have added the following to the caption of Figure 2: "A PWV of ≤ 3 mm occurs $\sim 10\%$, $\sim 35\%$, $\sim 65\%$, and $\sim 80\%$ of the time for CARMA, the PdBI/NOEMA, the SMA, and ALMA, respectively."

p.18: comparison between local ULIRGs and SMG, the latter being bluer.

"This difference suggests a lower average dust attenuation, which could be due to the fact that high-redshift SMGs may be more extended than local ULIRGs." I guess this is not the obvious conclusion, the most obvious being that at high-z galaxies have lower metallicity, and therefore lower dust attenuation, for the same radial extension. Independent observations show that galaxies at high-z are more compact, more clumpy, even if more gaseous. So the stellar component could be bluer, may be also because the stars are in average younger.

We agree with referee #1 that this is true in general. However, if we compare SMGs with ULIRGs, which are a very specific class of IR-bright galaxies, this actually isn't necessarily the case. SMGs are more extended than ULIRGs and there's evidence that they're already quite metal-rich (also from lines of evidence like high dust masses). We have therefore edited this section of the text to read: "This difference suggests a lower average dust attenuation despite similar dust masses (e.g., da Cunha et al. 2010), which could be due to the fact that high-redshift SMGs may be more extended than local ULIRGs and/or the dust and stellar distributions are not co-located (e.g., Hodge et al. 2016)."

In Section 2.3.3 when the GMC and star formation scaling laws are described, and only a few clumps are derived from lensing galaxies, you should cite the recent work on the Snale <https://ui.adsabs.harvard.edu/abs/2019NatAs...3.1115D/abstract> where a lot more clumps have been detected

We thank the referees for pointing us toward this impressive work, which was still embargoed when we wrote this section. We have added a discussion on it in section 2.3.3: "Meanwhile, Dessauges-Zavadsky et al. (2019) used 30-pc ALMA mapping of the CO(4-3) emission in the $z = 1.036$ 'Cosmic Snake' to identify 17 molecular clouds in this Milky Way progenitor. They measured the masses, surface densities and supersonic turbulence implied by these clouds, reporting values 10–100 times higher than present-day analogues, and bringing into question the universality of GMCs. It is important to note that the Cosmic Snake has one of the largest magnification factors known for a giant arc (80 ± 10 ; Ebeling et al. 2009), making this sort of study rare even in the era of ALMA."

In Section 3.2.1, when discussing Σ_{SFR} going higher than the Eddington limit, you should caution that the FIR could be excited by a central AGN, and the true SFR could be over-estimated. The fraction of AGN-heated FIR is increasing with mass and L_{FIR} .

We agree this is a good point. We have edited the relevant paragraph in Section 3.2.1 to read: "Taking the extent of the FIR emission as a proxy for the extent of the dusty star formation, one of the immediate implications of the measured FIR sizes is for the global SFR surface densities (Σ_{SFR}) of high-redshift sources. (We note that the assumption is generally made that the FIR emission is heated primarily by star formation, with negligible AGN contribution; this could be wrong especially for the most massive sources)."

Note that in Section 3.2.8, the discussion on the existence or not of the clump could be enlightened by the example of the Snake (Dessauges-Zavadsky et al 2019), that was mentioned earlier.

Indeed, we have now re-introduced the cosmic snake earlier in that section "For example, Dessauges-Zavadsky et al. (2019) reported the discovery of 17 GMCs in the CO(4-3) emission of the $z = 1.036$ Milky Way progenitor the 'Cosmic Snake'. This is an exceptionally strongly lensed galaxy, providing a source-plane resolution as high as 30 pc, and allowing the GMCs to be studied at a resolution comparable to CO observations of nearby galaxies."

Then, during the discussion on clumps, we have added: "Dessauges-Zavadsky et al. (2019) report comparable mass distributions for their (CO- identified) GMCs and stellar clumps as those seen in (respectively) the gravitationally bound gas clouds and stellar clumps produced by simulations of fragmenting gas-rich disks (Tamburello et al. 2015); however, the GMCs and stellar clumps are still not co-located."

Section 3.5: influence of the CMB. Indeed, the contrast of the dust/gas emission in the outer parts of a galaxy is lower, as mention in Zhang et al (2016). However, it is true at any redshift, even at $z=0$ "For example, in galaxies where the dust temperature decreases with radius, the cool dust in the outer regions might not be visible against the CMB background, and therefore the size of the dust-emitting region might be underestimated for galaxies at high redshifts." >> at any redshift in fact.

Indeed. To this end, and in the context of this review, we have added "(this is true for galaxies at any redshift, but particularly relevant for ALMA observations at high redshifts)."

Very complete and impressive work!

We once again thank the referee for their thorough reading and the good points they raised.

Reviewer 2:

Overall, I found the manuscript a useful and comprehensive review.

We thank referee #2 for the positive report and suggestions, which we address below.

I would suggest perhaps that the conclusions section is made more "punchy", and that the authors consider a list format for some of the most crucial results there, and perhaps a more highlighted set of future recommendations for key work?

A couple of future opportunities seem to be potentially missing,

One is the capacity of very-wide-band correlations, as being highlighted by SMA and NOEMA, and while could update ALMA.

Two, the SKA. While it is too low in frequency for molecular line work, continuum sensitivity, and potential resolution is very impressive, and continuum and radio-recombination lines might make a substantial contribution here. The author's net rest in CMB effects could provide a route in here?

We appreciate the referees suggestions, and we have worked to address them as best as possible. To begin with, although we prefer to keep the paragraph format, we have edited the conclusions to try to better emphasize our main points. (We will not list all of the changes here, but direct the referee to the pdf). Regarding the capacity of very-wide-band correlations, we thank the referee for bringing up this excellent point. Please note that we already discuss the potential update to ALMA's instantaneous BW and the opportunities it would lead to in the final paragraph of conclusions. We have now updated this discussion to explicitly mention the NOEMA and SMA correlators. Finally, regarding the SKA, indeed, there was previously only a limited discussion of the SKA. We have added a new paragraph in the conclusions to focus on non-traditional line studies, including dense gas tracers and radio-recombination lines, and the potential for future SKA/ngVLA/ALMA B1 there (where the SKA is actually not necessarily too low frequency if we consider SKA Band-5). We have avoided a discussion of the continuum impact of SKA as it is beyond the scope of this review.

The most substantial suggestion I have is to include more information about ALMA's Surface brightness sensitivity being inevitably limited at finest resolution, which I think should be noted, as integration times for interesting observations can become "challenging". For non-experts, I suspect this issue might be under appreciated, and the minute-long integrations that can produce impressive results seem strange when compared with the hours required to resolve CO lines.

I wonder whether there's a way to highlight this near Fig. 2?

We understand the referee's point and have made a number of edits throughout the manuscript to address it. In particular:

-We have edited the caption of Figure 2 to add "We caution that for all interferometers (including ALMA), there is an inherent tradeoff between spatial resolution and surface brightness sensitivity, which is not reflected in this figure."

-We have added a caveat in the Introduction (Section 1.2) where we discuss ALMA's improved sensitivity: "We note that, like all interferometers, ALMA is still limited by the unavoidable tradeoff between spatial resolution and surface brightness sensitivity. ALMA offers the ACA to help improve the imaging of extended structures, but this limitation should nevertheless be kept in mind, particularly for observations with the most extended configurations."

-When discussing SDP.81 in Section 2.3.3 on 'kpc and pc-scale studies', we have added this caveat to the first paragraph: "We note that all of the targets for the Long Baseline Campaign were chosen specifically to demonstrate the suitability of the long baseline capability (ALMA Partnership et al. 2015a), and that even despite the relatively compact size of SDP.81's Einstein ring ($\theta_E \sim 1.5''$), a large amount of total observing time was required ($\sim 9\text{--}12$ hours per band) in order to achieve good uv-coverage (ALMA Partnership et al. 2015b). As a result, high-resolution ALMA imaging of this quality is still relatively uncommon."

-In the following paragraph, we have edited the the first sentence (adding part between **) to read: "From the source plane reconstructions, ALMA imaging of strongly lensed sources *with sufficiently good image quality* allows detailed investigations of the dusty star formation and ISM..."

-We have added the caveat to the opening paragraph of Section 3.2 on resolved studies: "We note that due to surface brightness sensitivity limitations,* much of the most detailed/highest-resolution work with ALMA has necessarily still focused on submillimetre-bright sources..."**

-Regarding the referee's point about resolved CO lines, we note that the time required to resolve CO lines isn't due purely to surface brightness sensitivity, but rather due to the intrinsic strength (or weakness) of the lines and the narrower bandwidths of observability. We have clarified this point in Section 3.2.3: "...resolved CO studies are particularly time-intensive due to the brightness of the lines and the more limited bandwidths over which they are observable (compared to the dust continuum), and this remains true even with ALMA."

-We have added a clarifying sentence in the Conclusions in the paragraph on resolved observations: "While the highest resolution imaging possible with ALMA will necessarily remain limited to select sources due to the inherent tradeoff between spatial resolution and surface brightness sensitivity..."

Minor comments:

Very end of Section 2.2.2 - isn't the multiplicity changing with depth already largely clear? It ught help to be more specific about the future approach required to getting samples suitable to understand this better at this point in the manuscript, or to highlight the relevant discussion at the conclusion.

The extent of the multiplicity and how it changes with depth is actually not clear, which is why we raise this point. The multiplicity implied by the blank maps would have repercussions for the submillimeter number counts. We have clarified this point at the end of Section 2.2.2 and also referred ahead to the discussion on the relation of multiples in Section 3.2.3: "The depth reached by the ALMA observations

would sometimes imply a large number ($N > 3$) of blended sources in order for them to be individually undetected (e.g., Hodge et al. 2013b, Stach et al. 2018), *which would have repercussions for the submillimetre number counts. Deeper ALMA observations constraining the source multiplicity as a function of observed flux density will be important for constraining theoretical models for the formation of SMGs (e.g., Cowley et al. 2015, and see Section 2.2.3).*

Section 2.2.5. Don't the substantial SPT samples indicate that a high-flux cutoff must be modest? With decent lensing models, they do seem to have some sources substantially brighter than 10mJy? This seems to be revisited in Section 2.3.1, but it might be a good idea to link it here, and discuss this factor.

We have added a footnote in this section that this hypothesis is regarding the intrinsic number counts, and not the bright-end sources seen in ultra-wide field surveys, which are dominated by lensed sources. The latter can of course be used to come at the problem from the other direction, which we have noted and pointed to Section 2.3.1, where we indeed discuss whether the magnification factors agree with the unlensed source counts: "Note that this hypothesis is regarding the intrinsically bright sources, and not the bright end sources in ultra-wide field surveys which are found to be dominated by lensed sources. The latter, however, can also help inform the debate if the lensing magnification factors are well-constrained (see Section 2.3.1)."

In addition, we have updated the quoted SFR cutoff range to 1000-2000 Msol/yr to correctly reflect the Barger+14 result, and we have corrected the average SPT magnification quoted in Section 2.3.1 to 6.3.

Section 2.2.6. Median redshift is perhaps not all you'd like to know - there's also issue of the shape of the whole distribution, and perhaps especially of the high-redshift tail, since detecting the restframe FIR vs UV continues to get easier at higher z ? The absence of a long tail I suggest gives a general hint that the first light might be a job for JWST, or points beyond.

We agree with the referee that the shape of the distribution would be the ultimate goal, but this is actually quite challenging given the spectroscopic incompleteness in current samples. This is discussed in the 3rd paragraph of Section 2.2.6, and we now clarify this further by emphasizing that spectroscopic completeness rates are $<50\%$ even for the most well-studied extragalactic fields: "More importantly, even with the correct SMG counterpart(s) identified through interferometry, obtaining spectroscopic redshifts in the optical/IR is still very challenging due to the faintness/dust-obscured nature of the galaxies, *resulting in completeness rates for unlensed SMG samples of $<50\%$ for even the most well-studied extragalactic fields (e.g., Danielson et al. 2017, Casey et al. 2017, and to further illustrate the point, note that only 44/707 sources (6%) from the AS2UDS sample of Dudzević et al. 2020 have spectroscopic redshifts).*

We also add a note in Table 2 regarding the reliance of z_{median} estimates on photometric redshifts: "Median redshift estimates for the samples are usually heavily reliant on photometric redshifts for individual sources – see Section 2.2.6."

We agree that the high- z tail is the most interesting aspect, which is already discussed in the 4th paragraph. It is true that the absence of a long tail means JWST will be important, but SMGs aren't the only way to probe the EOR (as discussed later in the review, including in reference to the REBELS ALMA Large program), so we refrain from commenting further on this point here. Lastly, we have added a reference to the updated SPT paper (Reuter 2020) which has since appeared on the ArXiv.

Section 3.2.7. Does SKA not have potential to compare with next-generation VLA here? Sensitive low-frequency continuum and radio recombination lines seem to have potential value. If the authors disagree, it would still be worth a gentle note about this large facility that's on its way.

We agree and have added a reference and relevant citation: "This is therefore an area that is ripe for future work, not just with the current ALMA, but also with the future Band 1/2 receivers, the VLA, the proposed ngVLA, *and the SKA (Wagg et al. 2015)*."

Boundary of P46/47. Can substantial AGN contamination not be ruled out already by spatial information? The AGN emission would have to lurk unresolved? The lenses samples having complex, but enhanced resolution in the source plane might help to address this, even if the ratios are difficult to pin down? Multiplicity itself also suggests AGN can't dominate?

Indeed the implication is that the weak AGN in the model mentioned would be spatially unresolved. Given that most of the galaxies discussed in that section are unresolved or marginally resolved, the current spatial information is not sufficient to resolve AGN. We have clarified in the text that these would be unresolved AGN.

On P49. Young dust with low-density/colloidal/dendritic structure can provide more opacity for lower mass. Early dust may indeed be different, and something that is discussed a little later, but perhaps should appear earlier as a caveat, which could be a positive thing. for future investigations, in [CII] vs continuum, Nd perhaps JWST mid-IR spectra?

We agree that the properties of the dust can change the opacity and that this may be different in the early universe, as already discussed in this section. We have edited it to emphasize this more clearly: "Typically, the dust emissivity properties at high-redshifts are assumed to be similar to those measured in the local Universe, simply because we lack empirical measurements. However, both theoretical models and laboratory studies indicate that different types of dust grains could have widely different emissivity properties (e.g., Köhler et al. 2015). *For example, different chemical compositions and structures of young dust grains can lead to different opacities per unit mass (see, e.g., Galliano et al. (2018), and Appendix B of Inoue et al. (2020))*". It is not far-fetched to consider that dust grains in primordial galaxies could be significantly different in their physical properties than dust grains in the present-day Milky Way..."

Overall, I thought it was a comprehensive and informative review. The biggest concern I had was a warning about surface brightness sensitivity at the highest resolution. At present I suspect the presentation doesn't reflect this challenge to ALMA's rate of discovery; without being critical, perhaps more managing expectations.

We thank the referee for the detailed read and good comments they supplied. We hope our reply above on surface brightness sensitivity adequately addresses their concern.

Additional Changes:

In addition to the changes motivated by the referee reports (discussed above), we have made the following improvements to the manuscript:

- We have added a footnote reference to Gomez-Guijarro+19 when discussing the relation of high-multiplicity Herschel-selected sources in Section 2.2.3.
- We have clarified what the 'S_nu limit' column in Tables 1 and 2 refers to in their respective footnotes, and we have corrected the quoted value in Table 1 for AS2UDS.
- We have added Magnelli+19 to Table 2 and updated panel 3 of Figure 7.
- We have removed a confusing sentence in the caption of Figure 7.
- We have added a reference to Garcia Vergara+20 in Section 2.2.7 in the context of Hierarchical Context of SMGs: "Meanwhile, Garcia-Vergara et al. (2020) used an ALMA-observed sample of SMGs to re-examine their connection to these populations via clustering."
- We have added a reference to Harris+12 in Section 2.3.1 when discussing the strongly lensed sources: "The strongly lensed nature of a small number of these SMGs was confirmed already using imaging with pre-ALMA interferometers (Negrello et al. 2010, Conley et al. 2011, Riechers et al. 2011b, Bussmann et al. 2012, 2013, Wardlow et al. 2013), *or inferred indirectly through comparison with, e.g., the empirical luminosity–linewidth relationship (Harris et al. 2012)*."
- We have added a parenthetical reference to Wilson+17 and Zhang+18 in Section 2.3.2 in the context of previous Herschel studies at shorter wavelengths.
- We have updated the discussion of H₂O studies in Section 2.3.2 to add: "Water line observations of the brightest lensed SMGs started with the PdBI/NOEMA with studies by Omont et al. (2011, 2013), Yang et al. (2016) (see also Riechers et al. (2013), for a multi-observatory detailed study of a maximum-starburst galaxy at $z = 6.34$, where multiple H₂O lines are detected, allowing for a detailed study of its excitation mechanisms). Recently, ALMA has enabled high-resolution observations of this molecule. A high-resolution observation of thermal H₂O in an extragalactic source was achieved with the 0.9'' detection in the strongly lensed source SDP.81 during the ALMA 2014 Long Baseline Campaign (ALMA Partnership et al. 2015b), recently superseded with $\sim 0.4''$ -resolution observations of the strongly-lensed merger G09v1.97 at $z = 3.63$ by Yang et al. (2019)."
- We have updated the discussion of molecular absorption line studies in Section 2.3.2 to add "Herschel/SPIRE enabled a few pre- ALMA detections of OH absorption in some of the brightest lensed high-redshift SMGs (e.g., George et al. 2014, Zhang et al. 2018a)."
- In the same paragraph, we have added: "ALMA has also recently detected the ground-state transitions of OH⁺ and H₂O⁺ in absorption towards two $z \sim 2.3$ lensed SMGs, which can be used to measure the rate of cosmic ray ionization in their extended gaseous halos, and from that infer the ionization rate in dense star-forming regions, closer to the sites of cosmic ray acceleration (Indriolo et al. 2018)."
- We have updated the discussion of SDP.81-related studies in Section 2.3.3 to add: "and Rybak et al. (2020) use photon-dominated region (PDR) models to further constrain the physical conditions of the star-forming gas."
- We have corrected the reference to Liu+19b to read Liu+20
- We have added a note in Section 3.3.2 that optically thin assumptions can also introduce severe biases: "Finally, even the optically-thin assumption can introduce severe systematic biases, with models that consider more general opacity scenarios retrieving typically higher dust temperatures and lower dust masses than optically-thin models (e.g., Cortzen et al. 2020, da Cunha et al., in prep)."
- We have corrected the number of pointings mentioned for the ASAGO survey in Section 4.1 and Table 4.
- We have corrected the reference to the Frontier Fields to Lotz+17

- We have added Umehata+18 in Table 4 and updated the discussion in Section 4.1: "ALMA Deep Field in SSA22 (ADF22). This survey first mosaicked a $\sim 2 \times 3$ arcmin² area with Band 6 at 1.1 mm (ADF22A region, Umehata et al. 2017), targeting the core region of a protocluster at $z = 3.09$, which had been previously identified via overdensities of Lyman-break galaxies (Steidel et al. 1998) and Lyman- α emitters (Hayashino et al. 2004). One of the main goals was to detect and characterize dusty star formation in the protocluster. This survey detected 18 SMGs at $> 5\sigma$, 10 of which are spectroscopically confirmed at the redshift of the protocluster. A follow-up mapped a contiguous area, ADF22B to a similar depth, bringing the combined area of the SSA22 ALMA coverage to 20 arcmin² (71 comoving Mpc² at the protocluster redshift; Umehata et al. 2018). This combined ADF22 area contains a total of 35 SMGs at $> 5\sigma$, with star formation rates $\sim 100 - 1000 M_{\odot} \text{ yr}^{-1}$. This is a clear overdensity of millimetre sources in the protocluster core (by a factor of 3–5 compared to blank-field number counts), suggesting that intense dusty star formation may be enhanced by the large-scale environment, as also found in other studies (e.g., Casey 2016)."
- We have updated Fig 27 and its caption (as well as the relevant section in Section 4.3) to reflect the latest ASPECS results.
- We have added a few other missed references throughout (not bolded due to the typesetting limitations).
- We have thanked the referees in the Acknowledgements.

We once again thank both referees for their time and insight.